# Optimized feature gains explain and predict successes and failures of human selective listening

Ian M. Griffith [1,2,3] ✉, R. Preston Hess [1,2] & Josh H. McDermott [1,2,3,4] ✉

Attention facilitates communication by enabling selective listening to sound sources of interest. However, little is known about why attentional selection succeeds in some conditions but fails in others. While neurophysiology implicates multiplicative feature gains in selective attention, it is unclear whether such gains can explain real-world attention-driven behaviour. Here we optimized an artificial neural network with stimulus-computable feature gains to recognize a cued talker's speech from binaural audio in 'cocktail party' scenarios. Though not trained to mimic humans, the model produced human-like performance across diverse real-world conditions, exhibiting selection based both on voice qualities and on spatial location as well as selection failures in conditions where humans tended to fail. It also predicted novel attentional effects that we confirmed in human experiments, and exhibited signatures of 'late selection' like those seen in human auditory cortex. The results suggest that human-like attentional strategies naturally arise from the optimization of feature gains for selective listening.

Organisms often base behaviour on one of many objects in their environment. This ability typically involves endogenous ('top-down') attention—a change in internal state that allows an organism to wilfully 'select' an object of interest for further processing, memory or a behavioural response. Selective attention has been a central focus of cognitive science and neuroscience since the 1950s[1], and much is known about both attentional abilities and their neural correlates[2–5].

Neurophysiological observations suggest a mechanistic account of attentional selection as multiplicative gains that enhance the perceptual features of an attended object. Specifically, attention to particular features tends to cause the responses of neurons tuned to those features to be scaled upwards, enhancing the representation of objects containing those features[6–9]. Other types of effects on neural tuning[9] can be explained by combining multiplicative gains with normalization[10]. Consistent with such findings, human neuroscience studies show attentional enhancement of attended sound sources[11–15].

However, it remains unclear whether feature-based multiplicative gains are sufficient to account for real-world attention-based abilities, in part because we have lacked working models of attention that can be evaluated in everyday settings. In particular, although computational models of sensory systems have made notable advances in being able to account for some types of human judgements of images and sounds[16–19], they have thus far largely not incorporated attentional mechanisms ('Discussion').

Speech comprehension is one setting in which selective attention is essential in daily life. We routinely must understand what someone is saying despite the presence of other concurrent talkers (the 'cocktail party problem')[20,21]. Human attentional selection of one voice among others has been demonstrated in many settings[22–26] and is known to depend on the features of individual sound sources—both spatial locations and voice qualities such as pitch. However, human attentional selection can be prone to failure in some settings, as when a distractor

[1]Department of Brain and Cognitive Sciences, Massachusetts Institute of Technology, Cambridge, MA, USA. [2]McGovern Institute for Brain Research, Massachusetts Institute of Technology, Cambridge, MA, USA. [3]Program in Speech and Hearing Biosciences and Technology, Harvard University, Cambridge, MA, USA. [4]Center for Brains, Minds, and Machines, Massachusetts Institute of Technology, Cambridge, MA, USA. ✉e-mail: imgriff@mit.edu; jhm@mit.edu

voice shares features with the target voice[23]. The reasons for these failures remain poorly understood. For instance, it has been unclear whether human selection errors represent suboptimal strategies, or whether such errors are largely inevitable given the feature overlap between human voices.

We used the cocktail party problem[20,21] as a setting in which to explore computational accounts of feature-based attention. We tested whether a task-optimized model equipped with multiplicative gains applied to sensory representations would replicate human selective listening behaviour. We assumed gains to be functions of a memory of the attentional target's properties and used a task in which the target properties could be estimated from prior exposure to a target talker's voice. We found that models with such multiplicative gains that were optimized to recognize the words of a cued talker reproduced a wide range of characteristics of human auditory attention, including its sensitivity to both spatial and non-spatial aspects of sound and its occasional failures of selection. They also predicted two previously undocumented traits of human attentional selection that we subsequently confirmed experimentally.

The results indicate that multiplicative gains, when combined with standard filtering, pooling and normalization operations, are sufficient to account for both the successes and failures of human feature-based attention. The framework we propose is general and should be applicable to any behaviour involving feature-based attention.

## Results

### Feature-based attention task

To study feature-based attention, we used a task in which a listener first heard an excerpt of a target voice (the 'cue'), then heard a mixture of the target voice and a distractor voice, and then reported the middle word spoken by the target voice (Fig. 1a). The excerpt of the target voice used for the mixture was different from that used for the cue, such that the cue indicated the target talker's voice and spatial location, but not the words spoken by the target talker in the mixture. To solve the task, the listener had to use properties of the cue (its voice properties and/or spatial position) to select the target voice from the mixture. The task captured key aspects of attentional selection in the wild by requiring selection based on the memory of a sound source, but its stereotyped form enabled training and behavioural benchmarking at a large scale. Depending on the experiment, sounds could be presented over headphones, with the same audio signal to both ears (eliminating any spatial cues), or from different spatial locations (with each ear receiving a different audio signal).

### Model optimization

We used supervised deep learning to build a model that could perform this task. Models were trained to report the middle word in a two-second speech excerpt spoken by a cued talker within a mixture of talkers. Both cue and mixture were presented as binaural audio. During training, audio was spatially rendered at locations in simulated reverberant rooms using head-related transfer functions to reproduce the spatial cues available to humans. To emulate the variety of real-world scenarios encountered by humans, each mixture in the training data varied in the number of distractor sources, whether distractors were speech or non-speech, the relative intensities of each source, their spatial configuration and the dimensions of the room they were heard in. The cue was always an excerpt of the target voice in isolation, presented at the same location as the target in the corresponding mixture. Models classified the middle word in the target excerpt (out of 800 possible word classes).

### Model architecture

Our main model architectures (referred to as feature-gain models) incorporated learnable attentional gain functions into a deep neural network model of the auditory system (Fig. 1b). The neural network took simulated cochlear representations of an audio waveform as input. Feature-based attention was implemented with parameterized sigmoidal functions (green box in Fig. 1c) that mapped the model's internal representation of the cue to scalar multiplicative gains. These gains were then applied multiplicatively to the mixture representation.

The model architecture instantiated the hypothesis that attention is mediated by multiplicative gains that are deterministic functions of the features in a mental representation of the target of attention. To compute attentional gains, the cue stimulus for a trial was first passed through the auditory system model (blue model stages in Fig. 1b). The activations of each model stage were then time-averaged and stored to obtain stage-dependent memory representations of the cue (orange stages in Fig. 1b,c). Gains were computed as a sigmoidal function of the corresponding memory representations (features with high activations in the cue yielded high gains), which were then multiplied with the activations of the mixture representation during the forward cascade of the model (black stages in Fig. 1b). Intuitively, features that are present in the cue should result in high gains, passing those features through the model when they occur in the mixture. The sigmoidal functions should enable models to learn how strongly to modulate the features at each model stage to select the target from the mixture. We used a single set of shared sigmoid parameters per model stage (we found this was sufficient to yield good performance). Gain function parameters and model features were jointly optimized, such that models should learn to prioritize information that enables selection of the target voice content.

We note that the application of feature gains happened immediately prior to each convolutional 'block', which always began with a normalization operation. The model thus instantiated the key components of the normalization model of single-unit attentional effects[10], but embedded in a hierarchical system of learnable gains and features.

To ensure that the results generalized across the details of the model architecture, we trained ten different feature-gain models with different architectures (Extended Data Table 1). The results for the feature-gain models are the average over these ten architectures. In a later section, we test the extent to which the feature-gain architectural assumption was important to the results.

### Human–model behavioural comparisons to assess attentional strategies

Because the models receive binaural audio signals, they should be able to learn to use both spatial cues and voice timbre cues if they are useful in solving the task. However, because the models were optimized only for word recognition, they were not explicitly constrained to reproduce human-like strategies and were free to learn any solution that yielded good performance. To assess whether the model reproduced the properties of human attention, we conducted a set of experiments to characterize the model's behaviour and compare it to that of humans.

### The model replicates human cocktail party performance in monaural conditions

We first ran human participants in a behavioural benchmark (Experiment 1) containing a set of diotic listening conditions (that is, in which the same audio waveform was presented to the left and right ears, eliminating binaural spatial cues). We then simulated the same experiment on the model. The evaluation stimuli were new to the model (and to the participants). As shown in Fig. 2, the model approximately replicated both the overall performance of human listeners and the dependence on signal-to-noise ratio (SNR) and distractor type (see Extended Data Fig. 1 for results for the ten individual model architectures and Supplementary Fig. 1 for plots of data distributions).

Previous models of word recognition have replicated patterns of intelligibility across different types of noise[19,27]. However, because they lacked the ability to use a cue to selectively report one of several talkers, they were unable to perform at human levels in the presence of

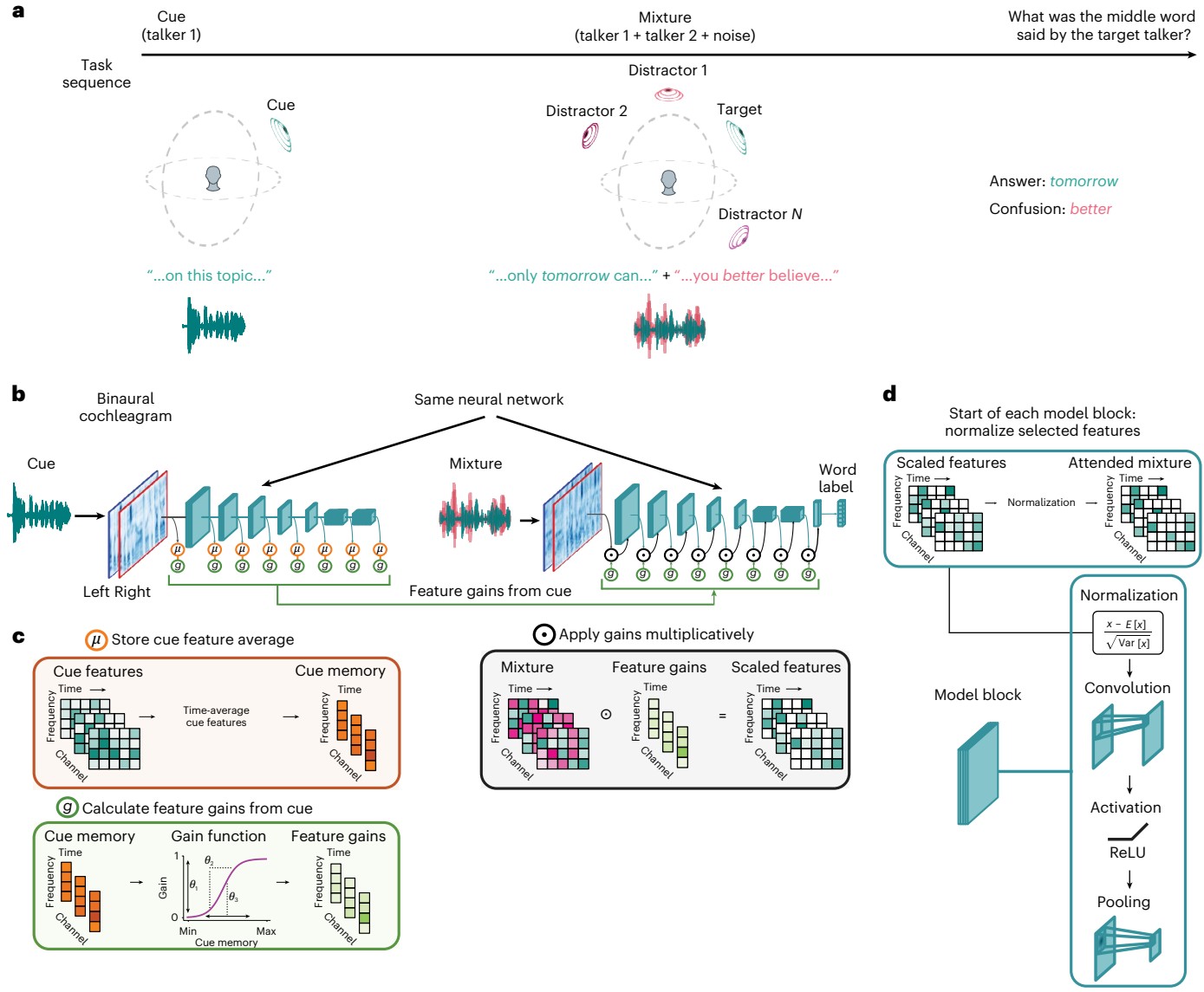

**Fig. 1 | Task and model architecture used to study attention. a**, Selective listening task. Listeners and models first heard an excerpt of a target talker's voice (the cue), then heard a mixture of a different excerpt of the target talker's speech superimposed with other sounds (the mixture) and then reported the word uttered by the target talker at the midpoint of the excerpt. **b**, Computational framework for modelling attention. A feedforward model of the auditory system (blue), comprising a simulation of the cochlea followed by a neural network, took binaural audio as input and produced word labels as output. Each convolutional block of the model consisted of layer normalization, convolution, rectified linear activation and pooling operations. When processing the mixture, the activations of the auditory model were multiplied by scalar gains (green). The gains were

determined by sigmoidal functions that operated on the average activations of a cue stimulus (orange), obtained by passing the cue stimulus through the same model of the auditory system. **c**, Example application of multiplicative gains in the model. Intuitively, the sigmoidal gain functions should enable gains to be high for features that have high activations in the cue, allowing these features to be passed through the auditory system, enhancing the representation of the target talker. **d**, Operations within a convolutional block of the model. Feature gains were applied immediately prior to the beginning of the subsequent model block. Each block began with a normalization operation, such that multiplicative gains were followed by normalization. ReLU, rectified linear unit activation function.

a single distractor talker, because without a cue, the task is ill-defined. As a result, it was not clear a priori that a model would be able to match human behaviour in this regime. However, the feature-gain model performed similarly to humans on trials including only the target and a single distractor talker (one-distractor condition in Fig. 2a). The similarity to human performance in this condition indicates that the model succeeds at attentional selection on par with human listeners. Human and model performance was robust to variation in cue duration (Extended Data Fig. 2), demonstrating some generality in the model's behaviour despite the stereotyped structure of the training examples.

**The model replicates effects of distractor language and sex**

To assess whether the model was sensitive to some of the same aspects of speech that influence human attentional selection, we reanalysed the one-distractor condition of Experiment 1. Human attentional selection is known to improve when target and distractor talkers differ in sex, in part because of the acoustic differences this entails[23]. Selection is also better when distractor talkers speak a language unfamiliar to the listener[28]. Native-English-speaking human participants in Experiment 1 replicated both of these effects, showing higher word recognition performance with different-sex distractors (Fig. 2b) and Mandarin language (Fig. 2c). The model also exhibited

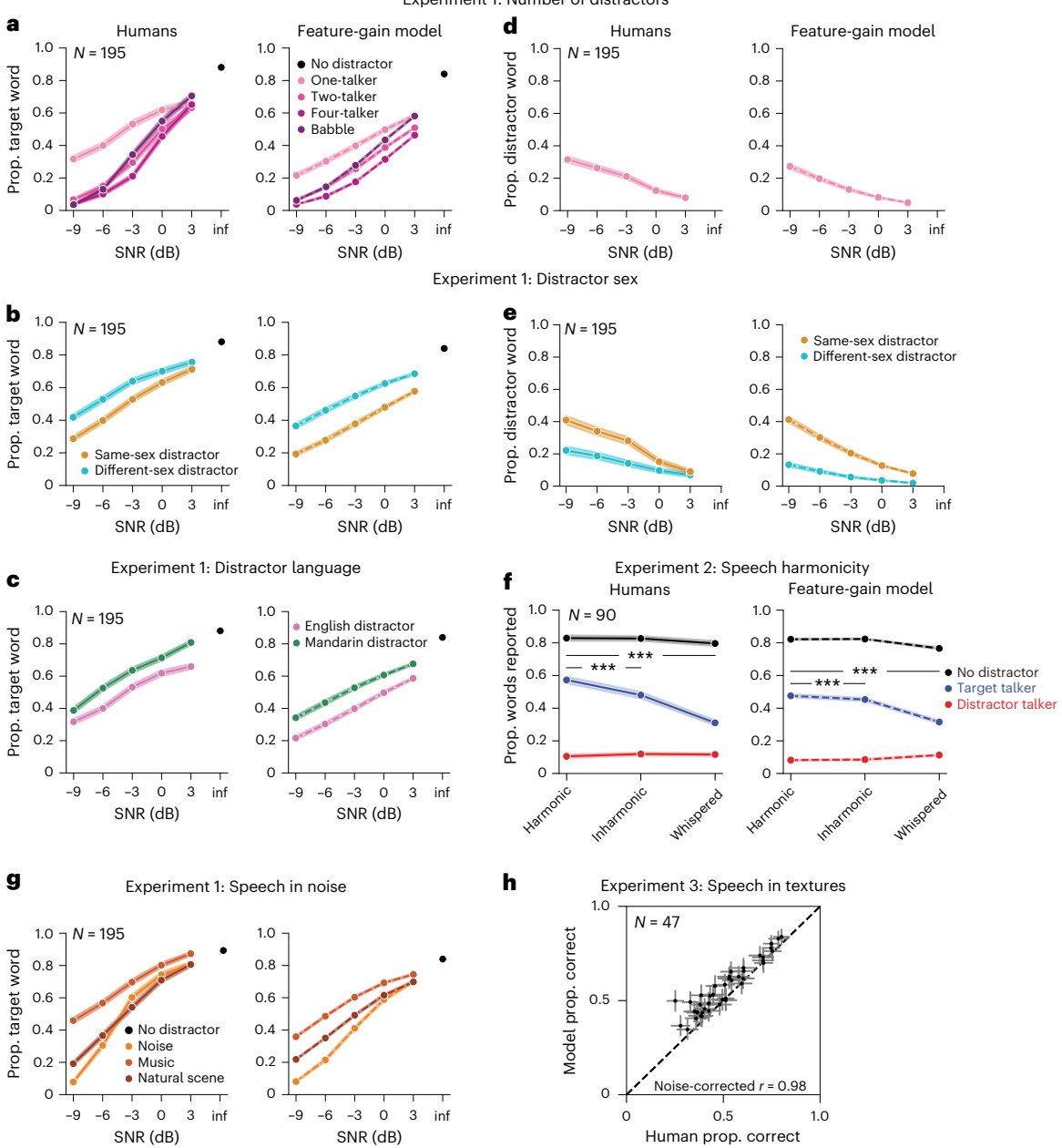

**Fig. 2 | Comparison of human and model attentional selections in monaural conditions. a**, Human and model performance versus SNR for speech distractor stimuli (Experiment 1). Here and in panels **b**–**g**, the shaded areas plot s.e.m. In some cases the shaded areas are smaller than the line width. The sample size is the number of human participants. Note that there is only one data point for the infinite (inf) SNR condition, because the distractor is zeroed out in this condition, making the different distractor conditions equivalent. Prop., proportion. **b**, Effect of same- versus different-sex distractor talkers (Experiment 1). **c**, Effect of foreign-language distractor talkers (Experiment 1). **d**, Human and model confusions, plotted versus SNR (Experiment 1). Confusions occurred when the

participant erroneously reported the word uttered by the distractor talker. **e**, Human and model confusions plotted separately for same- and different-sex distractor talkers (Experiment 1). **f**, Effect of talker harmonicity (Experiment 2). Note that the rate of confusions (reports of words uttered by the distractor talker) is low because the SNR was always 0 dB, at which there are few confusions in Experiment 1 as well. ***$P < 0.001$; two-sided $t$-tests, not corrected for multiple comparisons; the largest $P$ value was 0.0003. **g**, Effect of non-speech distractor sounds (Experiment 1). **h**, Human and model performance across a large set of distractor noises (Experiment 3). Error bars plot s.e.m.

both effects, indicating the model learned to rely on some of the same cues as humans.

## The model exhibits human-like failures of attention

Human attention is also known to demonstrate systematic failures. Even when attempting to attend to a target talker, humans sometimes mistakenly report what was said by a distractor talker instead. The root causes of these failures are not clear.

To quantify selection failures, we reanalysed the one-distractor condition of Experiment 1, measuring how often listeners reported words in the distractor utterance ('confusions' of the target and distractor). The overall confusion rate was low but increased at lower SNRs ($F_{4,776} = 51.71$; $P < 0.0001$; $\eta^2_{\text{partial}} = 0.21$; 95% confidence interval (CI), (0.18, 0.25)) and when the target and distractor were the same sex ($F_{1,194} = 89.11$; $P < 0.0001$; $\eta^2_{\text{partial}} = 0.32$; 95% CI, (0.25, 0.39); Fig. 2d,e). The model exhibited quantitatively similar effects (Fig. 2d,e). This

result suggests that some selection failures are an inevitable consequence of target–distractor feature similarity. We note that humans had slightly higher rates of confusions than the model; this could also reflect lapses or failures of executive function, which undoubtedly contribute to human errors in some settings.

### The model replicates effects of speech harmonicity

Human attentional selection has also been shown to depend on whether the constituent frequency components of speech signals are harmonically related[29]. We ran an additional experiment to measure this effect using our task (Experiment 2) and simulated the same experiment on the model. Target, distractor and cue signals were resynthesized to be harmonic, inharmonic or whispered (always of the same type; all harmonic, all inharmonic or all whispered). Without a concurrent distractor talker, human word recognition was similar for resynthesized harmonic, inharmonic and whispered speech (Fig. 2f; see Supplementary Fig. 2 for plots of data distributions). But with a concurrent distractor, human recognition was modestly impaired for inharmonic speech and substantially impaired for whispered speech, producing an interaction between the effect of speech harmonicity and that of single versus mixtures of talkers ($F_{2,178} = 61.42$; $P < 0.0001$; $\eta^2_{partial} = 0.41$; 95% CI, (0.34, 0.49)), as in prior work (Fig. 2f). The model qualitatively reproduced these effects, also showing an interaction between speech harmonicity and single versus mixtures of talkers ($F_{2,18} = 260.73$; $P < 0.0001$; $\eta^2_{partial} = 0.97$; 95% CI, (0.95, 0.99)), driven by worse performance for inharmonic ($t_9 = 5.75$; $P = 0.0003$; Cohen's $d = 1.82$; 95% CI, (1.21, 3.37)) and whispered ($t_9 = 15.14$; $P < 0.0001$; Cohen's $d = 4.79$; 95% CI, (3.80, 8.50)) speech than for harmonic speech in concurrent distractor conditions.

### The model replicates human speech-in-noise performance

The model also reproduced human-like patterns of performance across different types of noise distractors, like previous models[19,27]. In Experiment 1, both humans and the model demonstrated better performance with noise distractors than with speech distractors, and for some types of noise than for others (Fig. 2g). Figure 2h shows an additional experiment (Experiment 3) comparing the model and human listeners across a large set of different types of (non-speech) noise[19], further demonstrating the model's human-like dependence on noise type.

### The models replicate signatures of human spatial attention

Humans also benefit from spatial separation between sources, an effect often termed 'spatial release from masking'. This benefit is thought to be partly attentional in origin, as it depends in part on knowing where to listen[24]. Because our model was trained on binaural audio rendered from sounds at different locations, it could in principle learn to use spatial information to aid selection. However, because the models were only optimized to recognize words, it was not obvious whether spatial attention would emerge as a listening strategy.

To test whether the models exhibited human-like effects of spatial separation, we evaluated performance as a function of target–distractor separation in azimuth. We replicated a previously published experiment in human listeners[30] in which performance was measured with distractors placed symmetrically in azimuth (Fig. 3a). This design has the advantage of preserving the SNR at both ears as the distractors are moved away from the target talker, such that an advantage of spatial separation cannot be explained merely by increased SNR at one of the ears. As in the human experiment, we summarized performance at each spatial separation with a threshold (the SNR granting 50% of ceiling performance). The model displayed thresholds that varied with spatial separation on par with humans (main effect of offset for the model, $F_{4,36} = 426.57$; $P < 0.0001$; $\eta^2_{partial} = 0.98$; 95% CI, (0.97, 1.0); Fig. 3b), suggesting that it also made use of spatial location for selection.

### The model replicates human advantage with illusory spatial separation

Another compelling demonstration of spatial attention in humans comes from the fact that human listeners benefit from illusory separation of sources. Such illusory separation can be mediated by the "precedence effect"[31,32], in which a sound played in rapid succession from two locations is heard to come from the first location, plausibly a consequence of a strategy for robust localization in the presence of reflections[18]. Specifically, a distractor signal colocated with a target signal (Fig. 3c, top right) impairs speech recognition relative to when it is spatially separated (Fig. 3c, top left). But if a second, spatially displaced copy of the distractor signal is played along with the colocated signal, human performance improves, provided the displaced copy begins shortly before the colocated copy. The presumptive explanation is that the colocated copy is interpreted by the brain as a reflection, such that the distractor is represented as coming from the displaced location and can be filtered out with spatial attention. To test whether the model similarly benefited from illusory spatial separation, we simulated a previously published experiment[33] contrasting the benefit of illusory and true separation (Fig. 3c, Experiment 5). As shown in Fig. 3d, model performance benefited from illusory separation in a qualitatively similar way as in humans (main effect of illusory separation versus colocated, $F_{1,9} = 1,224.92$; $P < 0.0001$; $\eta^2_{partial} = 0.99$; 95% CI, (0.990, 0.996)). As in models optimized for sound localization in realistic environments[18], the model presumably learns representations of sound location that are adapted to the presence of reflections (potentially by suppressing the localization cues of late-arriving sound, akin to the biological auditory system[34]), allowing it to benefit from the illusory spatial separation in this experimental setting.

### Model predictions of human behaviour

There are many demonstrations of auditory spatial attention in humans, but they have mostly been restricted to a fairly consistent set of spatial configurations. One benefit of a stimulus-computable model is that experiments on the model are inexpensive, such that the model can be used to screen large numbers of experimental conditions to search for interesting effects. Such effects can then be tested in humans, serving both to further characterize human perception and provide strong tests of the model. With these goals in mind, we ran the model on all possible combinations of target–distractor locations, summarizing the effect on performance in Fig. 4a (see Supplementary Figs. 3 and 4 for the full set of results). Two effects stood out from this exhaustive screen, which we subsequently probed for in humans.

### The model predicts horizontal/vertical asymmetry in spatial selection

When we exhaustively tested target–distractor configurations on the model, target word recognition increased as a function of the distractor spatial offset, as expected. However, offsets in the vertical direction produced much less benefit of spatial separation than the same offsets in the horizontal direction (Fig. 4a, left vs right). Such differences are plausibly due to the different cues involved in the two types of offset—horizontal offsets are mostly signalled by differences between the ears, whereas vertical offsets are mostly signalled by monaural cues. It seemed plausible that monaural cues might be less robust to the presence of concurrent sources (Extended Data Fig. 3). This issue has been addressed in a handful of previous studies[35–39], but with mixed results, and to our knowledge, the effect of horizontal and vertical offsets had never been compared in the same setting.

To test for analogous effects in humans, we ran an experiment with human participants measuring speech reception thresholds (that is, the SNR permitting average performance of 50% correct) for different target–distractor offsets in azimuth and elevation (Fig. 4b; Experiment 6; see Supplementary Fig. 5 for plots of data distributions). As expected, thresholds decreased as azimuthal distractor offsets

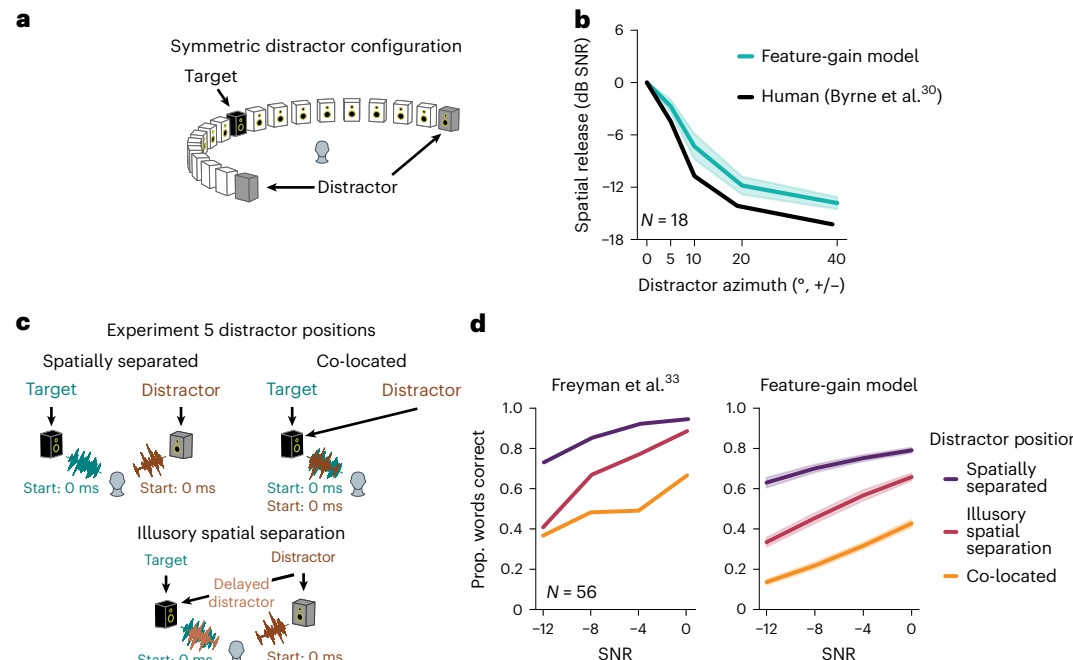

**Fig. 3 | Comparison of human and model spatial attention. a**, Stimulus set-up for Experiment 4. The target talker was positioned in front of the listener, with distractor talkers positioned symmetrically on either side. Word recognition performance was measured as a function of SNR for different target–distractor spatial offsets. **b**, Results of Experiment 4. The plot shows spatial release from masking, measured as the decrease in speech reception thresholds from the colocated condition where distractor talkers were also positioned directly in front of the listener. Human data were scanned in from the original publication[30] and replotted. The shaded areas indicate 2 s.e.m. for the feature-gain model (error bars for humans were not available in the original publication). The sample size is the number of human participants. **c**, Stimulus set-up for Experiment 5. Speech signals were played from two speakers. The target and distractor could be spatially separated, colocated or given illusory spatial separation by virtue of the precedence effect. In this latter condition, the target and distractor talker were both played from the left speaker, and a second copy of the distractor

speaker was played from the right speaker. The colocated copy of the distractor was slightly delayed relative to the separated copy, inducing the illusion that the distractor came from only the right speaker. This effect is commonly thought to reflect a localization strategy adapted to cope with environmental reflections. Reflections typically arrive from directions distinct from the true direction of the source from a listener but are delayed due to the increased path length compared with the sound that arrives directly from the source. The human auditory system appears to suppress localization cues from delayed sound components that are likely to be due to reflections, presumably to achieve more robust localization in real-world conditions. **d**, Results of Experiment 5. The plots show word recognition versus SNR for the three different spatial configurations. The shaded areas indicate 2 s.e.m. for the feature-gain model (error bars for humans were not available in the original publication[33]). The sample size is the number of human participants.

increased, replicating known effects of spatial release from masking. However, the benefit was much weaker in elevation (Fig. 4c; significant interaction between the direction of offset and the extent of offset, $P < 0.0001$, using a non-parametric test for interaction; Methods), confirming the model prediction.

**The model predicts central/peripheral differences in the width of a spatial 'spotlight'**

A second effect evident in Fig. 4a is that the spatial separation between target and distractor needed to obtain a benefit is much less for targets at the midline (0°) than for targets in the periphery (±90°). See Extended Data Fig. 4 for further analysis of this effect for both speech and noise distractors. This effect is plausibly due to the higher acuity of localization at the midline[40,41], which in turn is thought to relate to the derivative of binaural cues with spatial position (Extended Data Fig. 3). However, it is not obvious that spatial acuity for isolated sources should directly translate to the acuity with which speech can be selected from other sources. To our knowledge, this issue had not been previously examined in humans.

We ran an experiment with human participants measuring the benefit of spatial separation as a function of distractor offset, for both centrally and peripherally positioned targets (Fig. 4d; Experiment 7). The model was run on a simulation of the same experiment. As shown in Fig. 4e, the qualitative difference between central and peripheral

targets seen in models was also evident in humans: the distractor had to be spatially offset by a much larger amount for peripheral targets to yield the same benefit seen at smaller offsets for centrally positioned targets, producing a significant interaction between the effect of target–distractor offset and target position ($F_{3,81} = 6.73$; $P = 0.0004$; $\eta^2_{partial} = 0.20$; 95% CI, (0.11 to 0.35)).

We note that in both Experiments 6 and 7, human performance was overall somewhat higher than model performance. An analysis of human performance over the course of both experiments showed evidence of a practice effect in humans. Specifically, human performance was similar to model performance at the start of the experiment but got better over time (Fig. 4f). A plausible explanation is that human participants adapted to the room in which the experiment was conducted[42], boosting their performance (the model had no such ability as its weights were frozen following training and did not change over the course of the simulated experiments).

**The models exhibit signatures of late selection**

A signature of human auditory attention to speech is the enhancement of the neural representation of a target source at relatively late stages of the presumptive auditory hierarchy, such as non-primary auditory cortex[14,15,43–45]. However, it remains unclear why attentional enhancement might be limited to particular processing stages. In principle, the stage at which enhancement occurs could be determined by anatomical

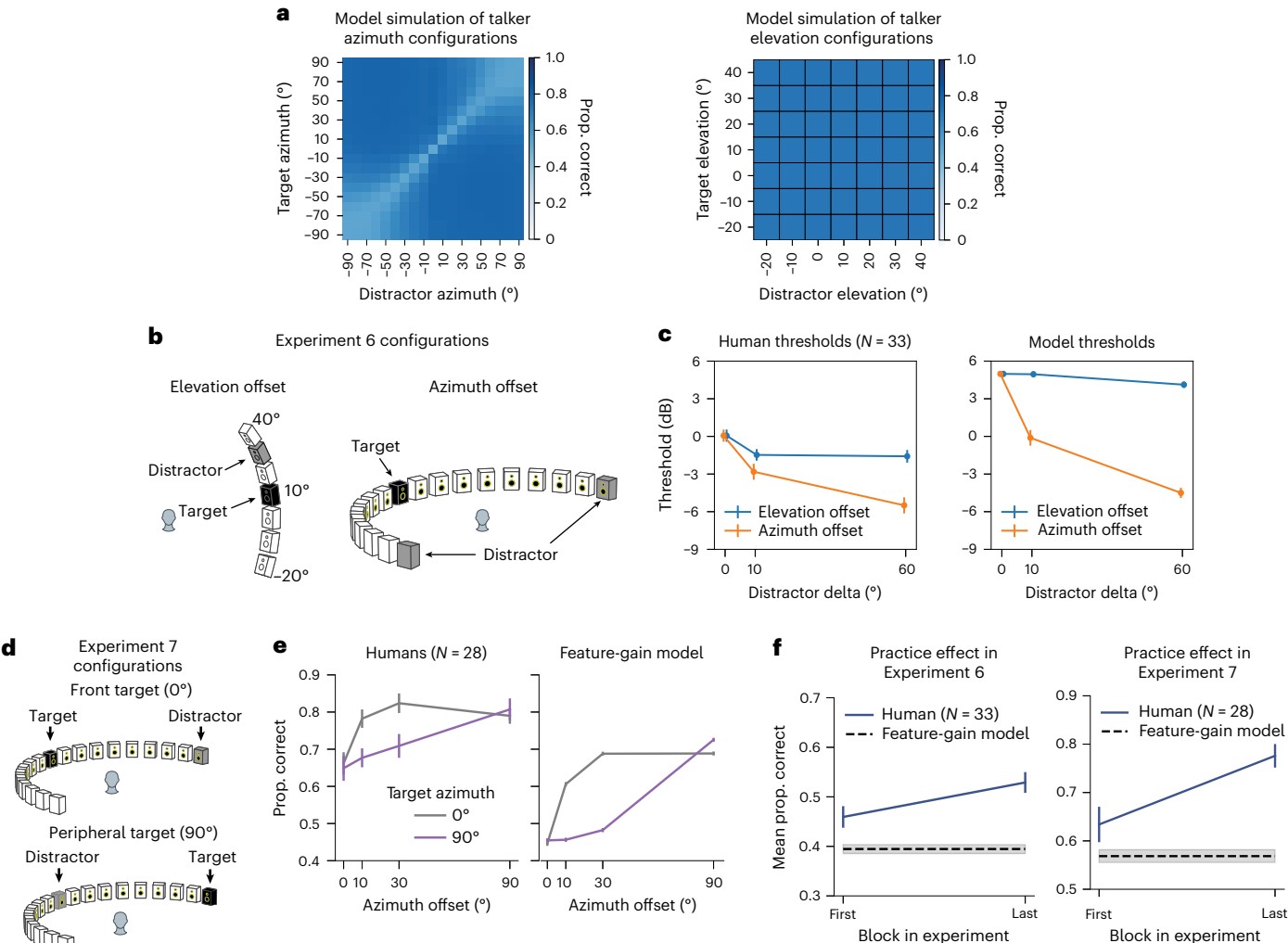

**Fig. 4 | Model predictions of human spatial selection. a**, Model word recognition performance across all possible combinations of target and distractor locations. The results are averaged over elevation and the front and back hemifields (left) or azimuth (right) for ease of visualization. The model results suggest larger benefits from azimuthal separation than from elevation separation (note that there was almost no effect of separation in elevation, such that the right panel is uniform) and a 'spotlight' that varies in width, being narrow for targets at the midline and wide for targets at peripheral locations. **b**, Spatial configurations tested in Experiment 6. To eliminate the possibility that an advantage from azimuth offset might be due to changes in SNR in the input to an ear, the azimuth offset condition presented two distractor talkers, symmetrically located about the target talker position. The elevation offset condition presented two distractor talkers at one elevation (constrained by the locations that were possible given the speaker array). **c**, Results of Experiment 6. Humans and models showed a larger benefit of spatial separation in azimuth than in elevation. The error bars indicate the s.e.m. The sample size is the number of human participants. **d**, Spatial configurations tested in Experiment 7. Target talkers were positioned at either 0 or 90 degrees, with distractors at one of four offsets. **e**, Results of Experiment 7. The error bars indicate the s.e.m. The sample size is the number of human participants. **f**, Practice effect evident in human performance in Experiments 6 and 7. The plots show human performance in the first and last block of each experiment, compared to overall model performance. The error bars indicate the s.e.m. The sample sizes are the numbers of human participants.

constraints on top-down connections. But the stage at which enhancement occurs might also simply reflect the stage at which features are most useful for selection. Task-optimized models provide one way to gain insight into this issue, because they reveal an optimized solution that can be compared to that of the brain.

To assess the locus of attentional enhancement, at each model stage we measured correlations between the activations of target–distractor mixtures and the activations of either the target or distractor alone[12] (Fig. 5a). Attentional enhancement of the target should result in target–mixture correlations being higher than distractor–mixture correlations. Differences between target–mixture and distractor–mixture correlations were largest for configurations with spatially separated target and distractor signals, but in all cases became pronounced only at later model stages (Fig. 5b), indicating that enhancement occurs relatively late in the model. See Extended Data Fig. 5 for the results

plotted separately for individual model architectures. The same analysis performed on a model with randomly initialized weights did not show evidence for stage-specific enhancement (Fig. 5c), indicating that this result is not an inevitable consequence of the model architecture. This result is qualitatively consistent with human neuroscience evidence[14,15,43–45] and suggests that the solution arrived at in the brain could reflect the representational locus at which enhancement is effective in enabling speech recognition.

To examine how selection relates to feature tuning, we measured model activations for a set of speech excerpts rendered at different spatial locations and analysed the proportion of units at each model stage that were selective for location, pitch, talker and the middle word of the excerpt (Extended Data Fig. 6). This analysis revealed that tuning to location and talker—the two dimensions used for selection in the task—was prevalent in early model stages, with tuning to words being

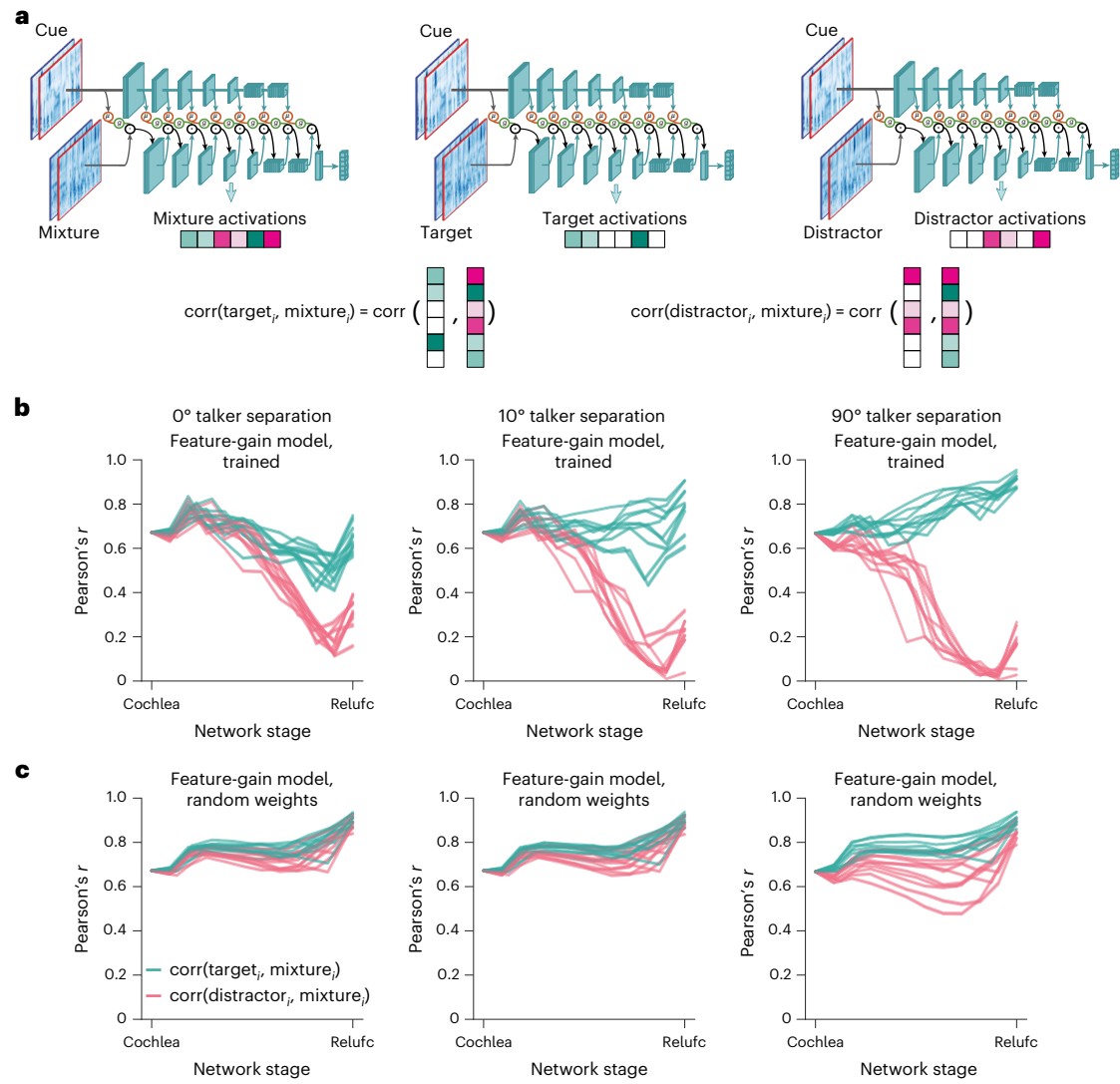

**Fig. 5 | Stage of attentional selection. a**, Explanation of stage-of-selection analysis. Target, distractor and mixture excerpts were passed through the auditory model, with feature gains derived from a cue matching the target. At each model stage, the activations of the target and distractor were correlated with the activations of the mixture. **b**, Task-optimized feature-gain models showed signatures of target selection in late stages. Each plot shows the results for a different spatial configuration of the target and the distractor. Each line shows the results for one of ten model architectures. The $x$ axis indicates model stage (normalized for depth) from early (Cochlea) to late (Relufc). Relufc denotes

the ReLU activation function following the final fully-connected layer. In all architectures and configurations, the late-stage representation of the mixture is more similar to that of the target than that of the distractor. In **b** and **c**, error bars on results for individual models are omitted for clarity. **c**, Same as **b**, but for models with random weights. Note that the modest difference between the target– and distractor–mixture correlations with 90 degrees of separation probably reflects the large difference in SNR that occurs in this condition within an ear, such that random features that by chance have greater input from one ear than the other are sufficient to achieve a modest degree of selection.

weak in early stages and stronger later. Overall, this tuning is consistent with the results shown in Fig. 5, in that tuning to the dimensions used for selection is present prior to the stages where selection is evident, such that feature gains could be used there to help select the target.

## Models lacking architecturally constrained feature gains are less human-like

The inclusion of gain functions in the model instantiates an inductive bias on how attention could work. To investigate the importance of this inductive bias, we trained several alternative versions of the model (Fig. 6a). A 'baseline' model removed the constraint of explicit gain functions entirely, instead receiving the mixture and cue as separate input channels, with model weights again optimized for recognition of the target word in the mixture. To test the importance of the application of gains followed by normalization, we also trained a feature-gain model in which this ordering was reversed, with gains applied after

normalization ('gain-after-normalization'). We also ran alternative architectures in which attention was constrained to operate only on particular stages. An 'early-only' model had gains applied only at the cochleagram stage, whereas a 'late-only' model had gains applied only at the final fully connected model stage. Because each model stage had a single shared sigmoid gain function defined by only three parameters, the early-only, late-only and feature-gain models had nearly the same number of parameters (differing by 21 parameters out of 62.6 million). The baseline model had slightly more parameters (just over 69 million) due to the extra input channels, but the models were overall similar in complexity.

After optimizing each of these alternative models, we ran them on each of the previously described experiments, measuring the proportion of target words correctly reported in each condition as well as the proportion of confusions in single-distractor conditions. We compared each model's pattern of performance across conditions to that of

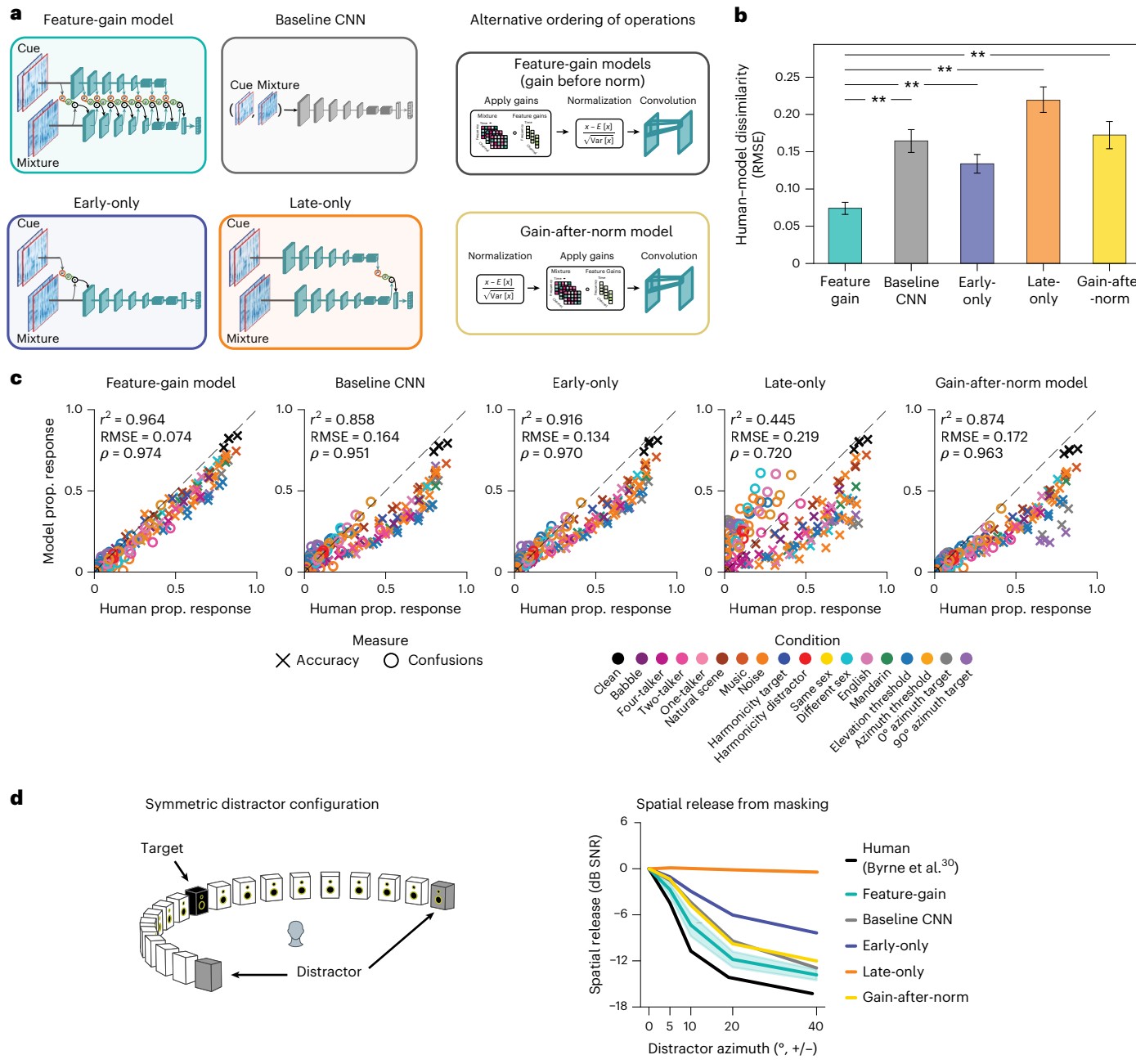

**Fig. 6 | Dependence of attentional selection on model architecture.**
**a**, Alternative model architectures: (1) the main feature-gain architecture; (2) a baseline architecture in which the cue was supplied to the model as additional input channels, without explicit feature gains; (3) a gain-after-normalization architecture; (4) an early-only architecture in which feature gains were applied only at the cochleagram; and (5) a late-only model in which feature gains were applied only at the last convolutional stage. CNN, convolutional neural network.
**b**, Aggregate measure of human–model behavioural dissimilarity for each model architecture. The plot shows RMSE between human and model performance in all experimental conditions. Because this is an aggregate measure derived for a single model across all 200 experimental conditions, there is no underlying data distribution to depict. The error bars show 95% CIs obtained via bootstrap of human–model similarity (resampling experimental conditions). **P < 0.01, via two-tailed sign test, not corrected for multiple comparisons; exact P = 0.002

in each case. **c**, Scatter plots of model versus human performance in each of the experimental conditions from Figs. 2–4. Each of the alternative architectures produces a worse match to human performance in some conditions.
**d**, Performance of alternative model architectures (feature-gain, baseline, early-only, late-only and gain-after-normalization models) as a function of target–distractor separation in azimuth for symmetrically positioned distractors (Experiment 4). The models are compared to human data from Byrne et al.[30]. The y axis plots spatial release from masking (the decrease in speech reception thresholds compared with the colocated distractor condition) at each tested azimuthal offset. Each of the alternative models displayed less human-like spatial release from masking than the feature-gain model. The late-only model showed no spatial masking release (orange), and the others exhibited degraded masking release compared with the feature-gain model.

humans, using both root-mean-squared error (RMSE) and correlation metrics. These two metrics are complementary: the RMSE metric captures differences in both absolute and relative performance, whereas the correlation metric isolates differences in relative performance.

A good model should match both absolute performance and relative performance differences between conditions and should match humans as well as or better than other models on all metrics. We also used a mutual information metric (Extended Data Fig. 7), which yielded

similar conclusions. As an overall summary measure, we jointly analysed both the proportion correct in each condition and the proportion of confusions in each single-talker–distractor condition.

Overall, the feature-gain model explained much of the variance in human performance across all experiments, and the explanatory power depended on the architectural constraints provided by the feature gains (Fig. 6b; see Extended Data Fig. 8 for the results for each individual model architecture). Each alternative architecture showed significantly lower human–model similarity (RMSE: feature-gain versus baseline sign test ($N = 10$), 10 negatives, $P = 0.002$; difference of means, −0.070; 95% CI, (−0.078, −0.063); feature-gain versus gain-after-normalization sign test ($N = 10$), 10 negatives, $P = 0.002$; difference of means, −0.078; 95% CI, (−0.086, −0.071); feature-gain versus early-only sign test ($N = 10$), 10 negatives, $P = 0.002$; difference of means, −0.040; 95% CI, (−0.048, −0.032); feature-gain versus late-only sign test ($N = 10$), 10 negatives, $P = 0.002$; difference of means, −0.125; 95% CI, (−0.133, −0.118); Pearson's $r^2$: feature-gain versus baseline sign test ($N = 10$), 10 positives, $P = 0.002$; difference of means, 0.093; 95% CI, (0.085, 0.099); feature-gain versus gain-after-normalization sign test ($N = 10$), 10 positives, $P = 0.002$; difference of means, 0.077; 95% CI, (0.070, 0.082); feature-gain versus early-only sign test ($N = 10$), 10 positives, $P = 0.002$; difference of means, 0.036; 95% CI, (0.028, 0.041); feature-gain versus late-only sign test ($N = 10$), 10 positives, $P = 0.002$; difference of means, 0.507; 95% CI, (0.498, 0.513)). Inspection of the pattern of proportion correct revealed that the alternative models tended to perform worse than the feature-gain model and were less correlated with humans across conditions (Fig. 6c). In particular, there was weaker spatial release from masking for each alternative model (Fig. 6d). See Extended Data Fig. 9 for an explicit comparison of model performance.

These results confirm that the architectural bias of multiplicative gains helps reproduce human-like attentional behaviour. We emphasize that the results do not preclude the (likely) possibility that a sufficiently large baseline model, trained on enough data, could reproduce the performance of the feature-gain model (potentially by learning to replicate the effect of the gains). We view the feature gains as analogous to other architectural motifs (for example, convolution) that in principle could also be learned by a less structured model. We also emphasize that the alternative model architectures were independently optimized and were free to learn an entirely different feature hierarchy that allowed them to maximize their performance given the constraint of gains in particular locations (or no explicit gains at all). It is thus perhaps not surprising that these models were able to perform well above chance. The success of the architectural constraint in replicating human behaviour nonetheless provides support for its role in biological attention.

## Discussion

We investigated feature-based attention using stimulus-computable models optimized to solve the cocktail party problem. We augmented standard feedforward neural network architectures with memory-driven multiplicative feature gains and optimized the models to report the word spoken by a cued talker given only binaural audio. The resulting models replicated the phenotype of human auditory attention, correctly reporting a cued talker's speech at comparable levels to humans and exhibiting performance variation across conditions like that of humans. In particular, the models exhibited human-like advantages for different-sex and different-language distractors, harmonic voices, and both real and illusory spatial offsets between target and distractor talkers. We used the models to sample target–distractor spatial arrangements more exhaustively than has been possible in human listeners and saw two notable effects that we then tested in humans. Both of these model predictions were borne out in human listeners. The model also made errors in the same settings as humans, and to around the same degree. However, model performance and human–model similarity were dependent on the architectural motif of

multiplicative gains, being worse in models without explicit gain functions or with gains restricted to either early or late model stages alone. These results provide support for the idea that feature-based attention can be explained by multiplicative gains and suggest that both human successes and failures of attention reflect an optimized solution to the problem of selecting a sound source via its features. Lastly, inspection of the model representations showed evidence of late selection, providing a normative perspective on effects seen neurophysiologically.

### Relation to prior work

Although the conceptual ingredients of our model have a long history in attention research, there has been little prior work incorporating them into working models of sensory systems. Previous auditory models of the cocktail party problem have largely focused on the problem of inferring the distinct sources underlying an auditory scene, without a means to direct attention to one or more of the sources[46–50]. Previous computational work on attention has tended to either model effects of attention on neural responses[10] (rather than behaviour) or consider behavioural effects in small-scale models and simple tasks[51–54]. The main previous attempt to test the effect of multiplicative gains in a working model tested vision models on an object detection task in which four different images of objects were concatenated[55]. This study found that the application of gains proportional to a unit's selectivity for a particular object category increased the likelihood that the model would report that category, as might be expected to occur behaviourally in humans. A related recent vision model used feedback connections as a way to emphasize particular object categories, again showing that this aided a model's object detection in concatenations or superpositions of object images[56]. Our work builds on these efforts by (1) optimizing attentional mechanisms for task performance, (2) showing that a single computational framework can account for many of the known attentional phenomena in the domain we consider (attention to speech) and (3) using a model to make predictions about human attentional selection (and then validating these predictions).

Other work has used models to explore other aspects of attention. For example, neural network models can be trained to guide simulated eye movements during visual search using priority maps computed from a target image and to reproduce aspects of human visual search behaviour[57,58]. 'Bottom-up' exogenous attentional cueing effects can also in some cases emerge naturally in task-optimized models[59]; such effects are widely believed to tap into distinct mechanisms from the endogenous effects we studied. We note that the word 'attention' is also used to refer to a computational motif within transformer architectures popular in current machine learning[60] but that this motif differs from biological selective attention in not being directed to a particular target object or sound source.

Our work was inspired by a large body of neuroscience experiments documenting neural correlates of attentional selection[2,3,12–15,43–45] and suggesting that such correlates can be explained by multiplicative effects on neuronal responses[7,8]. However, such experimental findings leave it unclear whether the documented neurophysiological effects are sufficient to account for behavioural effects. Our results indicate that multiplicative gains applied at the appropriate stage of processing, with the right features, are sufficient to enable human-like auditory attentional behaviour. The model results leave open the neurobiological implementation of the cue memory and feature gains but are compatible with previous proposals for sources of feature-based attention in prefrontal cortex[61] and for gain changes implemented via combinations of excitation and inhibition[62].

One of the main debates surrounding attention involves the stage at which attentional selection occurs[44,45,63,64]. These debates originally concerned whether attention acted before or after 'semantic' processing, but with the discovery of attentional effects within sensory systems, interest shifted to differences in attentional modulation between stages of sensory hierarchies[65], and it is that setting in which we

contrast 'early' and 'late' selection. Our results show how relatively late selection can emerge as an optimized solution for attentional selection of speech. This optimized solution could also depend on the system architecture or on other biological constraints we did not model. Our results thus do not provide a definitive explanation of the locus of selection found in the brain, but they illustrate a computational approach to understanding the issue. It also remains possible that the optimal stage of selection depends on the task[66], which could be investigated by applying our framework to multiple tasks.

Our work builds on a large prior literature that has documented human performance in selective listening tasks with speech[22–25,28–30,33]. The model clarified these previously documented phenomena in several ways. First, the similarity of human and model performance (in particular, the similar extent and pattern of selection errors) suggests that human selection failures may partly be an inevitable consequence of feature overlap between target and distractor talkers (some are surely also due to lapses, distraction and other factors we did not model). A priori it was not obvious that human errors would be so closely replicated by an optimized system. Second, we used the model as an engine to screen a large set of spatial configurations, not all of which had been tested thoroughly in humans. This process yielded predictions of effects that we then confirmed in human experiments. Such cycles of human and model experiments illustrate the value of having stimulus-computable models of behaviour.

### Limitations and future directions

Our modelling framework used a stereotyped task setting in which there is a cue stimulus from which attentional gains can be derived, followed by a stimulus to which the attentional gains are applied. This setting facilitated large-scale training and testing of attention but does not fully capture the variety of ways in which real-world attention arises. Some of the additional complexity found in real-world attention probably reflects flexible executive control. For instance, humans can direct auditory attention to a verbally instructed location or to a familiar voice that comes to mind even in the absence of a prior cue stimulus. We envision that the same basic attentional gain mechanisms could work in these settings, but with the gains derived from an internal representation rather than the cue stimulus. Humans can also adjust the strength of attention using executive control, perhaps on the basis of the perceived difficulty of a task and potentially relating to the feeling of effort. Extending the modelling framework to allow the strength of attentional gains to vary could help us understand effort in computational terms.

Another complexity that we neglected is that attentional gains are probably refined over time, as the experience of attending to a particular sound source plausibly causes the representation of the source's properties to become more precise, which could be profitably incorporated into the attentional 'filter'. Evidence for an evolving attentional filter comes from findings that humans can use attention to track sources whose features change over time[67]. Accounting for these abilities would require a more complicated way to set attentional gains according to the selected source. The ability to refine attention over time may be particularly important when attention is based on an abstract memory of a class of target stimuli (for example, the sound of a motor vehicle or of a woman's voice), rather than a specific recently heard stimulus. The abstract memory may serve as an initialization of attentional gains that are then updated on the basis of the experience of listening to the actual sound source encountered by the listener.

As instantiated in the model we presented here, attention depends only on the cue stimulus. In some settings, attentional gains might additionally be shaped by the properties of stimuli to be ignored. Models whose gains are also a function of the mixture could be used to investigate this idea. Models like ours could also provide hypotheses for the representation of stimuli outside the focus of attention[1,13], for

instance by measuring what can be decoded about unattended stimuli from different model stages.

The models we built here are composed of simple operations that are loosely inspired by neuroscience but deviate in many ways from biological sensory systems, making them inappropriate as models of some neural phenomena related to attention[68]. However, as it becomes possible to train models that are more biologically realistic[69], the general framework we propose here should remain applicable. Lastly, our framework is modality-independent, and extensions to other modalities could help reveal whether general principles govern feature-based attention in vision and hearing.

## Methods

### Informed consent

All participants provided informed consent in accordance with the Massachusetts Institute of Technology Committee on the Use of Humans as Experimental Subjects. All participants received compensation for their involvement. No experiments were preregistered.

### Training data generation

The training dataset consisted of 3,973,192 labelled exemplars, with each exemplar consisting of a cue signal and a multi-source mixture with a corresponding word label. Cue signals were two-second binaural audio clips of a single target talker spatialized to a location (defined by an azimuth and an elevation) relative to a simulated listener position. Mixture signals were another two-second binaural audio clip: a superposition of a different excerpt of the target talker (rendered at the cued location) and other spatialized audio signals (excerpts of other talkers and/or non-speech sound sources, rendered at locations that could be distinct from that of the target). Word labels corresponded to the word spoken by the target talker that overlapped the middle (that is, the one-second mark) of the mixture.

**Speech corpora.** The speech excerpts used to train the models were sourced from the English-language training split of the ninth release of the Common Voice speech corpora[70]. We screened the corpus to obtain a curated list of speech excerpts that would support our attentional word recognition task. The purpose of the screening was to balance the occurrence of single-word classes, the number of utterances per talker, and talker sex for both targets and distractors across examples contained in the training set.

First, word boundaries were extracted from the recordings and transcripts by using the Wav2Vec2 model to perform forced alignment[71]. We then removed examples that did not have a 44,100-kHz sampling rate, helping ensure that excerpts natively contained the frequency content implicated in supporting human sound localization[72]. Words spelled with fewer than five characters or spanning longer than two seconds were excluded. From the remaining set of word excerpts, we took at most 5,000 examples per word to limit word class imbalance, randomly sampling from the available examples for word classes exceeding this limit. The top 800 most numerous words were taken to be our training vocabulary. Word class balance was then obtained by resampling each word class to have 5,000 exemplars, drawing from the screened set with replacement. This returned 3,994,484 total single-word examples, with 2,041,825 unique utterances before up-sampling word classes.

Of the 15,735 unique talkers that remained in our training set after the prior screening steps, 3,407 were female talkers and 12,325 were male talkers. To avoid overrepresenting male talkers when generating training scenes, our final screening step was to filter the 3,994,484 excerpts to obtain a sex balance at the excerpt level. We did this by taking all available examples of female talkers in the word-balanced set (496,649 of 3,994,484) and then sampling the same number of male-talker examples from the remaining set. The resulting screened set holds 993,298 total single-word examples (496,649 female, 496,649

male). Cue, target and distractor speech excerpts were sampled from this screened set of single-word examples when we constructed the final training set.

Validation set examples were generated using the same screening procedure described above, sourcing materials from the corresponding validation split of Common Voice. For the validation set, we sampled up to 250 examples per word class instead of 5,000 as in training, resulting in 196,968 single-word excerpts. Gender balancing was not imposed for the validation set.

**Noise corpora.** Natural sound clips were sourced from AudioSet[73]. We screened the entirety of AudioSet to select sound excerpts that were appropriate for our scene generation procedure. The main goals were (1) to avoid speech content, as we wanted all speech in the dataset to come from the speech corpora described above, and (2) to ensure that the sampling rate was high enough to be appropriate for spatialization. We screened AudioSet examples to find a curated list of 'parent' clips and then excerpted individual clips from these parent clips. We first removed AudioSet examples labelled with "Music", "Speech", "Singing", "Vocal music", "Whispering", "Shout" or "Silence". As with the Common Voice excerpts, we then removed examples that did not have a 44,100-kHz native sampling rate. Parent clips were also screened to be at least 9 seconds long, enabling many 2.5-second excerpts per example. We performed the screening separately on the original training and validation split assignments of AudioSet. This returned 661,877 parent clips for training and 14,570 parent clips for validation. Individual natural sound clips were taken from these parent clips as part of the scene generation procedure. To obtain each natural sound clip in a scene, we first sampled a parent clip (uniformly from the screened training clips) and then sampled a 2.5-second excerpt from the parent clip via a uniform random crop.

**Virtual room simulation.** To spatialize scenes, we used the same room simulator and a similar set of room simulations as Saddler and McDermott[19]. The description of the simulator is reproduced from their paper apart from minor edits indicating where our parameters differ from theirs. The simulator used the image-source method, incorporating KEMAR HRTFs, to render sets of binaural room impulse responses for 2,000 different shoebox-style rooms. Room dimensions were sampled log-uniformly between 3 and 30 m for length and width, and between 2.2 and 10 m for height. One listener head position was sampled for each room (uniformly), under the constraint that the head was at least 1.45 m from every wall and no higher than 2 m from the floor. Binaural room impulse responses for 1,584 source locations (2 distances from the listener × 72 azimuths × 11 elevations) were rendered for each room and listener position. One of the distances was always 1.4 m. The other distance was independently sampled for each azimuth–elevation combination (drawn uniformly between 1 m and 0.1 m less than the distance from the listener to a wall). We included 1,800 of the 2,000 rooms in the training set and used the remaining 200 for validation. The final training and validation datasets consisted of 2,851,200 and 316,800 simulated positions, respectively. Our simulated spatialization differed from that in Saddler and McDermott[19] only by the inclusion of negative elevations: we used 11 total elevations compared with the 7 in Saddler and McDermott, giving 1,584 source locations per listener environment with 2,851,000 total positions (versus 1,008 source locations per listener environment and 1,814,400 total positions in Saddler and McDermott).

**Training data scene generation.** Auditory scenes were created by combining talker clips from the curated set of Common Voice examples with natural sound clips from the curated set of AudioSet examples, which were then spatialized using the room simulator. Scene parameters were varied across training exemplars to sample a wide range of conditions. To generate a training exemplar, we used the following

procedure. First, we sampled a target excerpt from the curated list of Common Voice examples. Second, we sampled a cue excerpt from the set of clips produced by the target talker, restricted to be centred on a different word than the target clip. Third, we sampled the total number of distractor sources (uniformly from one to six sources). Fourth, we sampled the number of these distractor sources that were talkers (uniformly from zero to $n$, with $n$ being the sampled number of distractors for the scene). Fifth, we sampled the distractor talker clips from the Common Voice examples not produced by the target talker. Sixth, we sampled the remaining distractor sources from the curated set of AudioSet excerpts. Individual natural sound clips were taken from an AudioSet excerpt by first uniformly sampling a parent clip and then randomly excerpting a 2.5-second crop from that parent clip. Seventh, the sound level of each distractor source was uniformly sampled from a 20-dB range.

In each scene, the target and cue sources were localized to the same azimuth and elevation relative to the sampled listener position. To prevent the model from exclusively exploiting localization cues, half of the training scenes had distractor sources at the same location as the target source. For the remaining half of the training examples, the azimuth and elevation of each distractor source were uniformly sampled from the possible source locations relative to the listener position. After the distractor clip levels were set, the distractor clips were spatialized to their selected locations and then combined (summed in each channel).

To increase variability in the training data, each target clip in the curated set of Common Voice clips was sampled four times, each time being part of a different scene. The final composition of the training set contained 3,973,192 unique exemplars (1,988,317 female, 1,984,875 male). Validation scenes were generated via the same procedure, but with clips sampled from the validation set source materials, resulting in 196,968 exemplars.

**Signal augmentations.** Cue clips, target clips and distractor scenes were pre-sampled and stored separately, enabling augmentations to be applied to the target clips and distractor scenes. To prevent the models from conflating the characteristics of the recording conditions for an individual talker with properties of their voice, bandpass filters were applied as augmentations to either the cue or target excerpt for 50% of the training examples. Filtering was performed using digital Butterworth filters, uniformly sampling low-frequency cut-offs from 40 Hz to 400 Hz, high-frequency cut-offs from 4 kHz to 16 kHz and the filter order from 1 to 4. To avoid having models overfit to the onset times of labelled words, target clips were randomly shifted in time (either forward or backward with equal likelihood). Time shifts were uniformly sampled from 0 to 50% of the labelled word's duration, constrained so the labelled word still overlapped the middle (one-second mark) of the target clip after shifting. Mixture clips were obtained online during training by superimposing the target and distractor scenes at an SNR uniformly sampled from −10 dB SNR to 10 dB SNR. Cue and mixture clips were then RMS-normalized to 0.02. Finally, to allow the models to learn how to report words said in isolated speech excerpts without being cued, 10% of training examples featured single-talker target signals with silence as a cue. This was done by setting the mixture equal to the target and using silence (an array of zeros) instead of the paired cue signal for these examples.

**Boundary handling.** To avoid signal onset/offset artefacts, all cue, target and mixture signals were extracted to be 2.5 seconds long. Once augmentations were applied, signals were passed through the cochlear model, after which the middle two seconds were excerpted as the final stimulus.

## Model implementation
Models were implemented in the PyTorch deep learning library using the PyTorch Lightning framework for efficient training in

distributed settings. All model training and analysis used the computing resources of MIT's OpenMind computing cluster, running Python version 3.11.5, Pytorch version 2.1.1 and Pytorch Lighting version 2.1.1. All Python dependencies and package versions are available in the code repository.

## Cochlear model stage

The first stage of our models was a fixed simulation of the cochlea and auditory nerve, providing an ear-by-time-by-frequency representation intended to replicate the auditory cues provided by the human auditory periphery. This initial stage took the sound waveform as input. The input was passed through a finite-impulse-response approximation of a gammatone filter bank (with impulse responses truncated to 25 ms to reduce memory consumption), the output of which was half-wave rectified, passed through a compressive nonlinearity and then downsampled. These operations were performed separately on the left and right audio channels.

First, each of the two 44.1-kHz stereo audio waveforms was convolved in the time domain with the above-mentioned finite-impulse-response approximation of a 40-channel gammatone filter bank (1,102 taps per filter) with centre frequencies spaced uniformly on an equivalent-rectangular-bandwidth-numbered scale between 40 Hz and 20 kHz. Second, the resulting subbands were half-wave rectified. Third, the half-wave-rectified subbands were raised to the power of 0.3 to simulate the compressive response mediated by outer hair cells. Fourth, the compressed, rectified subbands were low-pass filtered with a 4-kHz cut-off and downsampled to 10 kHz, to both impose the upper limit on phase locking of inner hair cells and reduce the dimensionality of the neural network inputs. Low-pass filtering and downsampling were performed via 1D convolution with a Kaiser-windowed sinc filter, with a filter width of 64, a roll-off of 0.94759 and a beta of 14.76965, implemented with the Torchaudio transforms resample method. Finally, to avoid signal onset/offset artefacts, the middle 2 seconds were excerpted from the full 2.5-second input signal duration as described above. We refer to the resulting representation as a cochleagram. The left and right cochleagrams were supplied as two input channels, yielding a two-channel-by-40-frequency-by-20,000-time-step input to the neural networks.

## Neural network model stages

Models of feature-based attention were built using a convolutional neural network backbone equipped with feature-based gains as parameterized sigmoid functions. The neural network backbone consisted of a series of convolutional blocks, each consisting of convolution, pooling and normalization operations. The final convolutional block of each architecture was followed by a sequence of four operations: a single fully connected layer, a point-wise nonlinearity, dropout and a softmax classifier. Such architectures have yielded strong task performance and close matches to human behaviour on other auditory tasks in previous work from our lab[17–19,27,74], as well as state-of-the-art predictions of brain responses from human auditory cortex[27,75].

## Feature-based gain implementation

Feature-based attention was implemented with sigmoidal functions with learnable parameters at each neural network model stage:

$$g^{(l)} = \sigma\left(m_{cue}^{(l)}; \theta_g^{(l)}\right) = \theta_1 + \frac{(1 - \theta_1)}{1 + e^{-\theta_2\left(m_{cue}^{(l)} - \theta_3\right)}}$$

where $(l)$ indicates the network stage; $\theta_g = [\theta_1, \theta_2, \theta_3]$ are learned parameters for the bias, slope and threshold of the function; $m_{cue}^{(l)} \in \mathbb{R}^{C \times F}$ is the time-averaged representation of the cue at layer $(l)$ (of size $C$ channels by $F$ frequencies); and $g^{(l)} \in \mathbb{R}^{C \times F}$ are the feature gains for each of the $C \times F$ features. Gain functions operated at the cochleagram and each convolutional block in the neural network.

## Forward pass through the feature-gain models

The forward pass for our feature-gain models augmented the standard operations of a feedforward neural network with stage-specific application of gains derived from the cue representation at the same stage. Given a cue $x_{cue}$ and a mixture $x_{mix}$, the forward pass of the model runs as follows. First, we obtain a representation of the cue for each model stage

$$x_{cue}^{(l)} = f^{(l)}\left(x_{cue}^{(l-1)}; \theta_a^{(l)}\right)$$

where $f^{(l)}(\cdot; \theta_a^{(l)})$ is the $l$th model stage (that is, convolutional block), $\theta_a^{(l)}$ are the parameters for that model stage and $x_{cue}^{(l-1)}$ is the cue representation from the previous model stage. Second, a memory representation, $m_{cue}^{(l)} \in \mathbb{R}^{C \times F}$, of the cue, $x_{cue}^{(l)} \in \mathbb{R}^{C \times F \times T}$, is obtained by averaging over the time dimension of the cue representation at the same model stage:

$$m_{cue}^{(l)} = \frac{1}{T}\sum_{t=1}^{T} x_{cue}^{(l)(t)}$$

Third, gains are then obtained from the memory representation

$$g^{(l)} = \sigma\left(m_{cue}^{(l)}; \theta_g^{(l)}\right)$$

where $\theta_g^{(l)}$ are parameters for the gain function at stage $l$. Fourth, the mixture representation is obtained:

$$x_{mix}^{(l)} = f^{(l)}\left(x_{mix}^{(l-1)}; \theta_a^{(l)}\right) \odot g^{(l)}$$

where $x_{mix}^{(l-1)}$ is the mixture representation from the previous stage, $f^{(l)}(\cdot; \theta_a^{(l)})$ is the output of the $l$th convolutional block, $g^{(l)}$ are the feature gains obtained from the cue and $\odot$ is the element-wise multiplication operator. The same set of feature gains were applied to all time points of the corresponding mixture representation. After the final convolutional stage, the mixture representation was passed through a fully connected layer followed by the 800-dimension linear output stage for the word recognition task.

During this forward pass, the weights of the convolutional blocks, $\theta_a$, were shared when obtaining cue and mixture representations for each training example.

## Artificial neural network constituent operations

**Layer normalization.** Normalizing activations between neural network stages improves the efficiency of artificial neural network training by helping stabilize gradient updates: the magnitude of a parameter update in one stage is less likely to be amplified in following stages if they are separately normalized. Normalization-like operations are also common in biological sensory systems and have been proposed to interact with multiplicative gains to produce attentional effects observed neurophysiologically[10]. Layer normalization[76], a common choice for normalization operations in artificial neural networks, point-wise normalizes input examples individually using the mean and variance over feature dimensions, with high output reflecting high activity relative to the features of that input example. We used layer normalization (as opposed to batch normalization[77], the other common choice for artificial neural network normalization operations) because it is more similar to the normalization found in sensory systems in normalizing responses by a function of the current stimulus[78] rather than by a function of the training distribution. Layer normalization also does not require scaling the test examples to match the statistics of the training examples (as is necessary with batch normalization), which might aid generalization at inference. The layer normalization operation is defined as

$$x_{normalized} = \frac{x - E[x]}{\sqrt{\text{Var}[x] + \epsilon}} \times \gamma + \beta$$

where $x \in \mathbb{R}^{C \times F \times T}$ is the input tensor, $E[x]$ and $\mathrm{Var}[x]$ are the mean and variance over all feature dimensions $(C, F, T)$ of $x$, $\epsilon = 0.00001$ to prevent division by zero, and $\gamma \in \mathbb{R}^{C \times F \times T}$ and $\beta \in \mathbb{R}^{C \times F \times T}$ are tensors of learnable parameters.

**Convolution.** Each convolutional layer consisted of a bank of learnable filter kernels, $W \in \mathbb{R}^{C_{out} \times C_{in} \times n_f \times n_t}$, with $C_{out}$ different kernels, $C_{in}$ input channels and kernel dimensions of $n_f$ by $n_t$ in frequency and time, respectively. Inputs to each convolutional layer, $x \in \mathbb{R}^{C_{in} \times F \times T}$, were three-dimensional tensors with $C_{in}$ input channels, $F$ features and $T$ time samples. For the first convolutional layer, $F = 40$ and $T = 20{,}000$ (the frequency and time dimensions of the cochleagram), while $C_{in} = 2$ (the left and right audio channels).

Boundary handling for convolution operations was identical to that in Francl and McDermott[18]. Specifically, 'valid' convolution was used in the time dimension (that is, no zero-padding was applied), and 'same' convolution was used in the frequency dimension. The rationale for this choice was to avoid temporal edge artefacts that would otherwise result from zero-padding in the time dimension. Edge effects in the frequency dimension are less clearly inconsistent with biology because the cochlea has upper and lower frequency limits, and these effects were considered preferable to the rapid loss of dimensionality along the frequency axis that would occur with 'valid' convolution given the small number of input frequency channels.

For an input tensor $x$, the output of a convolutional layer is a tensor $y \in \mathbb{R}^{C_{out} \times F \times (T - n_t + 1)}$ given by:

$$y_{k_{out}, i, j} = \sum_{n=1, \, m=1, \, C_{in}=1}^{n_f, \, n_t, \, C_{in}} W_{C_{out}, C_{in}, n, m} \times x_{C_{in}, i+n, j+m}$$

The output from each convolution thus had dimension $(C_{out} \times F \times T - n_t + 1)$. Because convolutional layers were preceded by layer normalization, bias vectors were omitted (to minimize redundant parameters and reduce memory consumption). All convolutional layers used a stride length of 1.

**Point-wise nonlinearity.** We used rectified linear units as the point-wise nonlinearity in all architectures. This operation is defined as

$$\mathrm{ReLU}(x) = max(0, x)$$

**Weighted-average pooling.** Pooling layers downsample their inputs by aggregating information across neighbouring frequency and time points. Weighted average pooling with Hanning windows was used to reduce aliasing in our networks (which would occur if downsampling was not preceded by low-pass filtering)[79]. Pooling was performed by convolving input tensors $x \in \mathbb{R}^{C \times F \times T}$ with two-dimensional (frequency by time) Hanning window kernels, $H \in \mathbb{R}^{h_f \times h_t}$:

$$y_k = H_{s_f, s_t} * x_k$$

where $*$ is the convolution operation, $k$ indexes the channel dimension, and $s_f$ and $s_t$ are the stride length in frequency and time, respectively. As in Saddler et al.[17], the Hanning kernel had a stride-dependent shape, where

$$h_\cdot = \begin{cases} 1 & s_\cdot = 1 \\ 4s_\cdot & s_\cdot > 1 \end{cases}$$

for $h_f$ and $h_t$ depending on $s_f$ and $s_t$, respectively. For an input $x \in \mathbb{R}^{C \times F \times T}$, the corresponding output is $y \in \mathbb{R}^{C \times \frac{F}{s_f} \times \frac{T}{s_t}}$. If either $s_f$ or $s_t$ equals 1, no pooling was performed on the corresponding dimension.

**Fully connected layer.** A fully connected (also sometimes called linear or dense) layer applies an affine transformation to an input vector

$\mathbf{x} \in \mathbb{R}^D$. In our networks, where $x \in \mathbb{R}^{C \times F \times T}$ is the output of a convolutional block, $x$ was first flattened to a vector $\mathbf{x} \in \mathbb{R}^D$ where $D = C \times F \times T$. Then, an affine transform was applied

$$y_i = \sum_{d=1}^{D} W_{i,d} \cdot \mathbf{x_d} + b_i$$

producing an output vector $\mathbf{y} \in \mathbb{R}^N$, where $W \in \mathbb{R}^{N \times D}$ are learned weights and $\mathbf{b} \in \mathbb{R}^N$ is a learned bias vector.

**Dropout regularization.** Dropout is a form of regularization applied to input tensors $x \in \mathbb{R}^N$ during training, intended to minimize co-adaptation of units. On each forward pass, units are randomly set to zero with probability $p$, chosen independently by sampling from a Bernoulli distribution. The remaining fraction of units are scaled by $1/(1 - p)$, so the expected sum over all outputs is the same as the expected sum over $x$. During inference, the operation is replaced by the identity function. Dropout was applied during training to the activations of the penultimate fully connected layer preceding the softmax classifier in our networks, with $p = 0.5$.

**Softmax classifier.** A softmax classifier was the final stage in all models. First, input vectors $\mathbf{x} \in \mathbb{R}^D$ were passed through a fully connected layer $h(\cdot, W)$, with weights $W \in \mathbb{R}^{V \times D}$ mapping from $D$ features to $V$ word classes. The operation returned un-normalized activations for each word class (often called logits):

$$\mathbf{x}_{logits} = h(\mathbf{x}, W)$$

These logits were then scaled by the softmax function to produce an output vector, $\mathbf{y} \in \mathbb{R}^V$:

$$\mathbf{y}^{(v)} = \frac{\exp\left(\mathbf{x}_{logits}^{(v)}\right)}{\sum_{i=1}^{V} \exp\left(\mathbf{x}_{logits}^{(i)}\right)}$$

where $v$ indexes the word class. Because the values of $\mathbf{y}$ are greater than zero and sum to one, $\mathbf{y}$ can be interpreted as a probability distribution over word classes for the given input mixture.

### Architecture selection
Artificial neural network performance depends both on the network weights learned during gradient descent and on the 'hyperparameters' that define an architecture (that is, the number of layers, the number of channels per layer and the operations in each layer). We drew 13 architectures from successful models of word recognition and sound localization identified by prior work[17–19,27,74]. First, we piloted all training and model experiments using one of these architectures (the top architecture in Saddler and McDermott[19]), which yielded strong matches to human behaviour. We then trained the remaining 12 and selected the top 10 architectures (out of the full set of 13) on the basis of their validation set performance. Model results for each experiment are reported as the average across these 10 best network architectures, enabling us to report measures of uncertainty and marginalize across the eccentricities of any single network architecture. The hyperparameters of these 10 architectures are provided in Extended Data Table 1.

### Models with alternative architectural constraints
To test whether explicit feature gains were necessary for attentional selection, we instantiated models that altered how gains were included in the model architecture. We tested three control architectures that each informed a particular hypothesis. Each control architecture used the backbone convolutional neural network of the best-performing feature-gain model ("arch_v00" in Extended Data Table 1). Control models were trained using the same training set and optimization hyperparameters as the feature-gain architectures.

For the baseline model without explicit feature-based gains, cue and mixture signals were concatenated along the channel dimension and passed as a single input, $x \in \mathbb{R}^{C \times F \times T}$, with dimensions $C = 4$ channels (the stereo channels for cue and mixture), $F = 40$ and $T = 20{,}000$ (the frequency and time dimensions of each cochleagram). The model architecture was therefore augmented to accept four input channels (compared with two as in the feature-gain models), using the forward pass of a traditional convolutional neural network.

For the model with feature-based gains only at early stages, inputs to the model, the model forward pass and application of gains were identical to the unaltered feature-gain models, except that gains were only applied to the cochlear representation of the mixture before the first convolutional stage.

For the model with feature-based gains only at late stages, model inputs, the model forward pass and application of gains were identical to the full feature-gain models, except that gains were only applied to the outputs of the final convolutional stage.

For the model with alternative location of feature-based gains, the architecture was identical to the main feature-gain architecture except that the gains were applied after the normalization operation rather than before it.

## Model optimization

All models described above were trained via stochastic gradient descent, using the AdamW optimizer (with a learning rate of 0.00005 and a batch size of 288). The models were trained using a distributed data parallel strategy run on four NVIDIA A100 GPUs, where each GPU ran a unique subset of a training batch through a copy of the model parameters (which were updated synchronously with respect to the whole-batch loss), enabling larger batch sizes in training. A total of 16 CPUs (4 per GPU) and a total of 100 Gb of memory were used to execute data reading and online signal augmentations during training for each model. All models were trained until performance on the validation set converged (approximately ten training epochs of the training set).

## Human behavioural experiments

Human experiments were conducted both online and in person, depending on the type of experiment. All participants in both online and in-person experiments self-reported as native English speakers having no known hearing loss.

All diotic listening experiments (Experiments 1–3) were conducted online to facilitate large sample sizes (to increase the reliability and reproducibility of the results). Extensive research conducted in our laboratory has consistently demonstrated that online data can match the quality of data collected in traditional laboratory settings[27,80–85], provided measures are taken to ensure standardized sound presentation and participant compliance. Online participants were recruited using the Prolific platform. Participants were prescreened to have at least a 95% submission approval rate and to have not completed any online experiment hosted by the authors' lab in the past six months (to avoid familiarity with our experimental stimuli and task). All online participants were instructed to complete the experiment in a quiet environment and completed a headphone check experiment[86] prior to the main experiment to help maximize sound presentation quality. Participants who did not pass the headphone check did not complete the rest of the experiment. The participants then adjusted the volume of a calibration sound to a comfortable level at the start of the experimental session, and all stimuli were scaled relative to this maximum level. The participants also completed 12 catch trials in each experiment. These were intended to make sure the participants were paying attention to the experiment and served as an additional screening metric. Catch trials presented isolated words (spoken by one of the authors) in silence using the same clip for cue and mixture and were randomly intermixed with the experimental trials. Participants scoring less than 11 correct catch trials (91.6% accuracy) were excluded from our analysis.

Experiments 6 and 7, measuring the effects of spatial separation on attentional selection, were run in person over a speaker array (detailed below) to ensure the accuracy of stimulus playback location. Because the participants were monitored for the duration of these experiments, catch trials were not included.

## Experiment 1: effect of distractors on attentional selection in monaural conditions

Experiment 1 measured cued word recognition as a function of SNR and distractor type. Talkers and speech materials were not included in model training to enable fair comparisons between models and humans. Nine distractor types were used: one-talker same-sex, one-talker different-sex, two-talker, four-talker, eight-talker babble, one-talker Mandarin speech, stationary noise, recorded auditory scenes and instrumental music.

**Stimuli.** Cue, target and English-speech distractor signals were sourced from the portion of Spoken Wikipedia Corpora[87] material screened in Feather et al.[79]. Feather et al. removed Spoken Wikipedia articles due to potentially offensive content for human listening experiments, missing data or having bad audio quality (for example, due to computer-generated voices of speakers reading the article or the talker changing midway through the clip). We applied further screening to this set to identify clips centred on words in the model vocabulary (316,748 clips), so that the model could also be run on the experiments.

Target clips were additionally constrained to be sex-balanced. The in-vocabulary clips were further screened to find words with at least one example produced by both a male and a female talker, yielding 488 target words. One male-talker and one female-talker example was uniformly sampled for each target word, giving 976 total target clips. Cue clips were drawn from target excerpts centred on words not included in the model vocabulary.

English speech clips for the distractor signals of the one-, two- and four-talker distractor conditions were sampled from the remaining clips in the screened set of Spoken Wikipedia excerpts. This made it possible to measure the rate of distractor word reports without needing to reuse target clips as distractors in a given experiment. So that the effect of talker-sex similarity could be analysed in the one-talker distractor condition, one male and one female distractor clip were sampled for each target clip. Two two-talker distractor clips were sampled for each target clip by summing either two male or two female clips. The four-talker distractor clips were created by summing the two male and two female distractor clips. The talker identity and middle word of the distractor clip(s) were constrained to differ from those of the target clip. The one-talker Mandarin distractor clips were sourced from the Mandarin validation (termed "dev" in the dataset documentation) portion of Common Voice (version 9). One male and one female Mandarin speech clip were sampled for each target clip to be included in the talker-sex analysis. Speech-shaped noise was synthesized for each target clip by imposing the power spectrum of the target clip on white noise. Instrumental music, auditory scenes and eight-talker babble were sampled from the MUSDB18, IEEE AASP CASA Challenge and Common Voice test clips used in Saddler and McDermott[19] (450, 400 and 400 total clips, respectively).

Target clips were combined with distractor clips from each condition at six SNRs (infinite (that is, no distractor) and −9, −6, −3, 0 and +3 dB), producing 44,896 possible mixture stimuli (976 target clips with no distractor + 976 target clips × 9 distractor types × 5 SNRs). Cue, target and distractor signals were two seconds long, sampled at 44.1 kHz. Mixtures were obtained by summing the target and distractor clips at the desired SNR. All cue and mixture clips were normalized to an RMS of 0.02 for the model experiments and were presented at the same participant-determined level for the human experiments (described in the 'Procedure' section).

**Procedure.** Individual participants completed 12 catch trials and 184 experimental trials (4 trials × 5 SNRs × 9 distractor conditions + 4 no-distractor trials). Each participant heard a random sample of 184 of the 488 target words, with the talker sex randomly determined for each word. The words were then randomly assigned to distractor and SNR conditions for each participant, constrained to yield both a sex balance over target talkers and target–distractor sex conditions, and unique cue and mixture clips. Target–distractor pairings were constrained as described above. Catch trials and experimental trials were randomly intermixed.

On every trial, the participants first heard the 2-second cue, then a 0.5-second delay and then the 2-second mixture. This audio sequence was initiated by a button click. The participants reported the word said by the target voice in the middle of the mixture clip (defined as the word overlapping the one-second mark of the mixture). To equate the task for humans and models, the participants were asked to select words from a list of the 800 words in the model vocabulary. The participants typed responses into a text box, and, while they typed, the web page displayed a list of 800 words that was continuously updated to only include matches to the string they typed. Only words in the word list could be entered, analogous to how the models could only report one of these 800 words. The participants received feedback after each trial, displaying the trial outcome ('correct' or 'incorrect'), the number of remaining trials and their running accuracy. On incorrect trials, the correct word was also displayed to help participants learn to perform the task.

**Participants.** A total of 195 participants (98 female, 92 male, 4 non-binary, 1 no-report) met the inclusion criteria. The participants' ages were between 18 and 71 (median 33) years.

**Model experiment.** The models were tested on all combinations of the 976 target clips (both male and female examples of each target word) with the five SNRs, nine distractor conditions and the no-distractor condition (44,896 total stimuli) used in the human experiments. Because listeners used headphones rather than free-field speakers, the audio signals given as input to the model were not run through the room simulator. Instead, each mono clip was presented diotically (that is, the mono stimulus waveform was input for both the left and right channel of the model).

**Analysis.** To account for the possibility that participants correctly recognized the target speech signal but inadvertently reported a word that was adjacent to the trial's target word rather than the target word itself, word recognition performance was measured by contrasting a participant's response against all in-vocabulary words contained in a trial clip. In all conditions, a response was counted as correct if it matched any of the in-vocabulary words that were present in the target utterance. In the one-distractor condition, confusions were analogously measured as reports of words contained in the transcript of the distractor utterance. This scoring was applied to both participant and model reports.

To analyse the effects of target–distractor sex similarity and language familiarity, we reanalysed both the English and Mandarin one-distractor conditions for both participants and models. To analyse target–distractor sex similarity, trials from both distractor language conditions were first pooled, and then the mean for each distractor sex condition was obtained for each participant (or model architecture). To analyse the effect of distractor language, trials from both distractor sex conditions were pooled within each language, and then the mean for each distractor language condition was obtained for each participant (or model architecture).

**Experiment 1b: effect of cue duration**
It seemed possible that the fixed duration of the cue signal in our main task could limit the ecological validity of our results

(Extended Data Fig. 2). To assess whether human and model behaviour was robust to variation in the cue duration, we ran an additional experiment using the materials and procedure of Experiment 1, but with cues that varied in duration. The same set of cue, target and single-distractor excerpts used in Experiment 1 were used, but the cue signals were centre-cropped to be 0.5, 1 or 2 seconds long. All target–distractor mixtures were presented at 0 dB SNR.

**Procedure.** The procedure was identical to that of Experiment 1, with the following exceptions. The participants each completed 12 catch trials and 30 experimental trials (10 trials × 3 cue duration conditions). Each participant heard a random sample of 30 of the 488 target words, with the cue duration for each example randomly assigned for each participant. The 12 catch trials were randomly intermixed with the 30 experimental trials.

**Participants.** A total of 84 participants (43 female, 38 male, 3 non-binary) met our inclusion criteria. Participant ages were between 20 and 40 (median 32) years.

**Model experiment.** The models were tested on all combinations of the 976 target clips (both male and female examples of each of the 488 target words) with the three cue duration conditions and both distractor sex conditions (5,856 total stimuli) used in the human experiments. As in Experiment 1, the mono audio clips were not run through a room simulator and were instead presented to the model diotically.

**Experiment 2: effect of harmonicity on attentional selection in monaural conditions**
Experiment 2 measured how violations of harmonicity (the tendency of frequency components to be integer multiples of a fundamental frequency) impact cued word recognition. The participants were presented with a cue, followed by either the target excerpt alone or the target excerpt mixed with a distractor excerpt. Harmonic, inharmonic and whispered stimulus variants were generated as in Popham et al.[29] (summarized below), using the stimuli from the English one-distractor condition of Experiment 1. Twelve target–distractor harmonicity conditions were used (3 target harmonicities (harmonic, inharmonic and whispered) × 3 distractor harmonicities + 3 target harmonicities with no distractor). We only analysed the conditions in which the target and distractor were of the same type (harmonic/harmonic, inharmonic/inharmonic and whispered/whispered), to be consistent with the original experiment run by Popham et al. Participant and model responses were analysed identically to Experiment 1.

**Stimuli.** Experiment 2 used the set of cue, target and distractor pairings from the one-talker same-sex and one-talker different-sex conditions of Experiment 1. All stimuli were resynthesized from the original speech excerpts of Experiment 1 using the STRAIGHT algorithm[88]. Inharmonic speech examples were synthesized by shifting the frequency of individual harmonics above the fundamental frequency by an amount randomly sampled from a uniform distribution between −30% and 30% (which maximally reduced intelligibility in Popham et al.[29]). Jitter values were sampled independently for each harmonic's frequency of each original speech clip but were constrained (via rejection sampling) such that adjacent harmonics were always separated by at least 30 Hz. To generate the stimuli for an inharmonic trial, one jitter pattern was sampled for the target excerpt, and this same jitter pattern was applied to the cue and distractor utterances. Harmonic stimuli were synthesized by running stimuli through STRAIGHT without changing any of the constituent harmonic frequencies. Whispered speech was created as in Popham et al.[29], and the description of the synthesis from that paper is reproduced here with minor edits for clarity. Whispered stimuli were generated by omitting the sinusoidal excitation used for harmonic/inharmonic synthesis and high-pass-filtering the noise

excitation to simulate breath noise in whispered speech. The filter was a second-order high-pass Butterworth filter with a (3 dB) cut-off at 1,200 Hz whose zeros were moved toward the origin (in the $z$-plane) by 5%. The resulting filter produced noise that was 3 dB down at 1,600 Hz, 10 dB down at 1,000 Hz and 40 dB down at 100 Hz, which to the authors sounded like a good approximation of whispering. Without the zero adjustment, the filter removed too much energy at the very bottom of the spectrum. The noise excitation was combined with the time-varying spectral envelope using the same procedure employed for harmonic and inharmonic speech. The noise-excited stimuli were thus generated from the same spectrotemporal envelope used for harmonic and inharmonic speech, just with a different excitation signal.

Experiment 2 used the same combinations of cue, target and distractor source clips as Experiment 1. Harmonic, inharmonic and whispered versions of each target–distractor pairing were synthesized as described above. To produce the nine target–distractor harmonicity combinations, each version of a target clip was crossed with each version of the paired distractor clip. Cue clips always used the harmonicity of the target clip. Signal sampling rates and presentation levels matched those of Experiment 1.

**Procedure.** The procedure of Experiment 2 was identical to that of Experiment 1, with the following exceptions. The participants each completed 12 catch trials and 180 experimental trials (15 trials × 12 harmonicity conditions). Each participant heard a random sample of 180 of the 488 target words, with the harmonicity condition for each word randomly assigned for each participant. The 12 catch trials were randomly intermixed with the 180 experimental trials. We note that this experiment used a different type of cueing than was used in the original experiment of Popham et al. (the Popham et al. experiment showed the participants the first two words of the sentence they were supposed to listen to, prior to the start of the audio stimulus). This task difference may be responsible for the lower rate of confusions we observed than that in the Popham et al. experiment.

**Participants.** A total of 90 participants (43 female, 47 male) met our inclusion criteria. Participant ages were between 19 and 64 (median 34.5) years.

**Model experiment.** The models were tested on all combinations of the 976 target clips (both male and female examples of each of the 488 target words) with the nine harmonicity conditions and three no-distractor conditions (23,424 total stimuli) used in the human experiments. As in Experiment 1, the mono audio clips were not run through a room simulator and were instead presented to the model diotically.

### Experiment 3: word recognition in naturalistic auditory textures

To obtain a stronger test of the extent to which the models captured human speech-in-noise perception, we ran our models on an experiment from Saddler and McDermott[19], which probed human speech-in-noise recognition across a large set of naturalistic distractor signals. The description of the human experiment is reproduced from the original paper with minor edits for brevity.

Human word recognition accuracy was measured in 43 different naturalistic auditory textures. A total of 376 speech excerpts from the evaluation portion of the Word-Speaker-Noise dataset[79] were embedded in 376 unique exemplars of each auditory texture. The 2-s texture exemplars were previously generated[18] to match the statistics of 43 recorded real-world textures[89]. Speech excerpts were randomly assigned to one of the 43 texture conditions with the SNR fixed at −3 dB (each participant heard a different random assignment). The experiment was run online.

**Participants.** A total of 47 participants (24 female, 23 male) met the inclusion criteria (passing a headphone check[86], completing at least 100 trials and responding correctly to at least 85% of the catch trials). Participant ages were between 23 and 59 (median 39) years. We replotted the human data from the figure in the original paper.

**Model experiment.** We simulated a version of the original human experiment, modified to make the task compatible with the model task. For each target excerpt, a corresponding cue was obtained by sourcing a different excerpt of the target talker from the Word-Speaker-Noise dataset. We then measured model word recognition accuracy for speech embedded in each of 43 auditory textures, using the same stimuli as in the human experiment. The models were evaluated on the full stimulus set (16,168 stimuli = 376 speech excerpts × 43 auditory textures).

### Experiment 4: spatial tuning for masked speech

We simulated experiment 1 of Byrne et al.[30], which measured spatial tuning for the release of masking of target speech. The original experiment used a closed-set masked speech identification task, in which listeners were presented with five-word sentences, in English with fixed syntactic structure, spoken by one of 12 female talkers. Listeners were cued to the target stream by being told the starting word of the target sentence, which was fixed throughout the experiment, and that the target would occur directly in front of their position. Three distractor conditions were used: (1) two competing female talkers drawn from the 11 remaining talkers, (2) time-reversed speech made from the same sources used in condition 1 and (3) speech-shaped speech-envelope-modulated noise derived from the speech distractors of condition 1. The target source was randomly selected per trial and was always located at 0° elevation and 0° azimuth. Two distractor sources were positioned symmetrically in azimuth around the target talker at angles of ±0°, ±5°, ±10°, ±20° and ±40° azimuth and 0° elevation. For each distractor × angle condition, speech reception thresholds were measured using a one-down–one-up adaptive procedure, varying the SNR between the target and summed distractor signals to estimate the 50% correct point on the psychometric function. The distractor level was held constant such that the sum of the two distractors was 60 dB, and the target level was adjusted by 3-dB increments depending on participant accuracy. Trials were evaluated as correct, and the target level was reduced, if participants correctly identified the remaining four words in the target sentence. We reproduced the first of the three conditions (the one with two competing talkers) in our model experiment.

**Participants.** Data were collected for 18 normal-hearing, native-English-speaking listeners (11 female, 6 male, 1 non-specified; aged 18–40 years). We scanned the results in from the original paper figure.

**Model experiment.** We simulated a version of the original human experiment using a virtual anechoic room and the same-sex distractor speech materials from Experiment 7. The model simulation differed in two respects from the human experiment. First, the target voice was designated by a cue that was a different excerpt of the target talker (as in the other model experiments in this paper) rather than by one of the words spoken by the target talker. Second, in lieu of the adaptive procedure, we ran the model on nine SNRs (−18 dB to 6 dB SNR in 3-dB steps) at each spatial location. All combinations of 976 stimuli (cue, target and one-talker same-sex-distractor pairs), five loudspeaker presentation conditions (±0°, ±5°, ±10°, ±20° and ±40° azimuth) and nine SNRs were simulated (43,920 total stimuli). Model thresholds for each loudspeaker condition were estimated as follows. First, word recognition performance was measured at each SNR and presentation condition. Second, thresholds at each presentation condition were estimated by fitting a second-order polynomial to the performance-by-SNR curve and calculating the SNR granting 50% of the model's maximum performance.

## Experiment 5: the precedence effect with concurrent speech signals

We simulated experiment 1 of Freyman et al.[33], which measured the benefit of perceived spatial separation on speech intelligibility, using the precedence effect to induce illusory spatial separation between concurrent talkers. The 'precedence effect' traditionally refers to a perceptual phenomenon that occurs when two sounds played in quick succession from different locations seem to originate from the first location[31,32]. The effect has been hypothesized to reflect a strategy for localizing accurately in the presence of reflections. Because reflections typically arrive from a direction that is different from the direction of the source from the listener, they provide erroneous localization cues. However, they are also delayed relative to the direct sound from the source (which has cues that are faithful to the source location), because the path length traversed by the reflection is longer than that for the direct sound. Suppressing location cues from delayed components of a sound could thus make localization more robust to reflections. Consistent with this explanation, models that are optimized for localization in environments with reflections exhibit the precedence effect, whereas those optimized in anechoic environments do not[18]. Models plausibly learn to accentuate localization cues from early parts of a sound and suppress cues that are likely to originate from reflections, akin to effects seen in the early auditory system[34].

In the experiment, a target talker was superimposed with a distractor talker. The experiment included a condition where the target and distractor were colocated and a condition where they were spatially separated, as well as conditions where the locations were illusorily shifted by adding a second copy of one of the signals at a different location, with a small temporal offset.

In the original experiment, the participants listened to recordings of speech, with one target talker and one distractor, and reported the keywords spoken by the target talker. Speech materials were of syntactically correct sentences that had no meaning. The target talker was always the same female voice, while the distractor was always the same alternative female voice. The experiment took place in an anechoic chamber, with loudspeakers positioned in a semicircular arc at 0° and 60° to the right relative to the listener position, at a distance of 1.9 metres from the listener. Six conditions were used for the target and distractor locations (the numbers reflect the ordering in the original paper): (1) both the target and the distractor presented at 0°, (2) the target presented at 0° with the distractor presented at 60°, (3) the target presented at 0° with the distractor illusorily presented at 0° via the precedence effect, (4) the target presented at 0° with the distractor illusorily presented at 60° via the precedence effect, (5) both the target and the distractor illusorily presented at 0° via the precedence effect, and (6) the target illusorily presented at 0° and the distractor illusorily presented at 60°. Illusory separation in conditions 3–6 was enabled by presenting signals from both positions and starting playback at the 'perceived' location 4 ms before starting playback at the 'lag' location (taking advantage of the precedence effect). Four SNRs were presented per loudspeaker condition: −12 dB, −8 dB, −4 dB and 0 dB SNR. We reproduced conditions 1, 2 and 4.

**Participants.** A total of 56 participants (sexes not reported) completed the original experiment. We scanned the results in from the original paper figure.

**Model experiment.** We simulated the original human experiment using a virtual anechoic room and the speech materials from Experiment 7. All combinations of 1,952 stimuli (cue, target and single-distractor pairs), three loudspeaker presentation conditions (conditions 1, 2 and 4 described above) and the four original SNRs (−12 dB, −8 dB, −4 dB and 0 dB SNR) were simulated (23,436 total stimuli). To simulate the precedence effect, we first spatialized the distractor to the lead channel. Next, we applied the 4-ms onset delay (by zero-padding) to the

distractor and then spatialized it to the lag location. Finally, the two binaural waveforms were summed in each channel.

## Experiment 6: effect of spatial separation in azimuth and elevation

The benefit of target–distractor spatial separation was measured for separation in azimuth and elevation. Experiment 6 was conducted in person, and the stimuli were presented over a loudspeaker array (described below). Participant and model responses were analysed as in Experiment 1.

**Stimuli.** Speech excerpts for Experiments 6 and 7 were sourced from speech clips used in the online monaural experiments (Experiments 1 and 2). Cue, target and distractor stimuli were screened to include excerpts with frequency content beyond 16 kHz to help enable sound localization in elevation. Target excerpts were further screened under the constraint that each word was spoken by both a male and a female talker. These screening steps reduced the number of unique target words to 488 from 800 (976 total target excerpts, 488 per talker sex). This vocabulary size was large enough to allow for one unique word per trial in the in-person experiment. Cue excerpts were selected for each target excerpt by finding clips of the target talker centred on words not in the target vocabulary. To allow for analysis of talker–distractor sex similarity effects (which we did not present in this paper), each target excerpt was paired with both a same-sex and a different-sex distractor. To enable the use of symmetrically positioned distractors in the azimuth conditions, two distractors per sex condition were sampled from the screened set of distractors. This screening resulted in 1,952 unique combinations of cue, target and two-talker distractors that were sex-balanced across both target talkers and target–distractor sex pairings.

**Speaker array.** All experiments measuring the effects of spatial separation were run using a 19-by-7 array of loudspeakers arranged on a hemisphere (two-metre radius). The participants were seated at the centre. Relative to the participant's head, the array spanned 180° in azimuth (frontal hemifield) and −20° to 40° in elevation, with speakers spaced every 10° in both azimuth and elevation. Speaker locations were coded using labels that were attached to the bottom of each loudspeaker. Participant responses were collected on an Apple iPad held by the participant during the experiment. The participants were instructed to direct their head towards the loudspeaker directly in front of them for the duration of the stimulus. Once the stimulus ended, the participants could look at the iPad to enter their response. The participants were instructed to redirect their head towards the front loudspeaker before the start of the next trial was triggered. Compliance with these instructions was confirmed by experimenter observation.

**Procedure.** The trial procedure and task were identical to those of Experiment 1 (which was run online), except that the participants heard sounds from loudspeakers rather than headphones/earphones and that the cue/target/distractor varied in spatial position. First, a 2-second cue excerpt would play from a specific location, allowing the participants to hear both the target talker's voice and location. After a 0.5-second delay, a 2-second target excerpt was played from the same location, while two 2-second distractor excerpts were played simultaneously at locations that varied with respect to the target location. The target and distractor were separated in either azimuth or elevation, depending on the condition.

Two distractors were used in every trial. When the distractors varied in azimuth, the two (symmetrically placed) distractors served to preserve the SNR at each ear across different target–distractor separations[30]. When the distractors varied in elevation, the use of two distractors at the same elevation served to match the distractor properties to those of the azimuth conditions. The participants reported the

middle word spoken by the target talker (that is, the word overlapping the one-second mark). The participants entered responses by typing on an iPad, connected to a local server running the same experimental interface used in the online experiments.

A common set of target positions and distractor offsets were used in azimuth and elevation conditions to compare the effect of spatial separation in each dimension. Distractors were offset relative to target positions by 0° (that is, colocated), 10° or 60° in either azimuth or elevation. These offsets were chosen as those likely to reveal the extent of spatial tuning subject to the limits of our speaker array: 10° was the smallest offset between speakers, while 60° was the largest span available in elevation (minimum elevation of −20°, maximum of 40°). Cue and target excerpts were always presented at 0° azimuth and at either −20° or 40° elevation (so that 60° target–distractor offsets could be used). To equate the uncertainty of the target location in azimuth and elevation, one of the two possible target elevations was sampled randomly for each participant and was used for all trials (so that for any single participant the target was always at the same location in azimuth and elevation).

In the conditions in which the target and distractors were separated in azimuth, the cue, target and distractor signals shared a common elevation. The two distractor excerpts were presented symmetrically around the target at the specified offset. In the conditions in which the target and distractors were separated in elevation, the cue and target excerpts were always presented at 0° azimuth. The two distractor excerpts were presented from the same position, at 0° azimuth and an elevation offset from the target elevation by the specified amount. When the target was at −20°, the distractors were offset above the target location; when the target was at 40°, the distractors were offset below the target location.

The experiment used the method of constant stimuli. At each target–distractor position, word recognition was measured at a set of SNRs (−9, −6, −3, 0, 3 and 6 dB). In trials with colocated distractors, the two distractor waveforms were first summed, and then their combined signal was normalized to 65 decibels sound pressure level (dB SPL). In trials with symmetrically positioned distractors, each distractor was individually set to 61.99 dB SPL, such that the level of the two distractors was 65 dB SPL on average when summed at the ear. The level of the target excerpt was then adjusted to obtain the desired SNR as 65 dB + SNR. The cue signal was always presented at 65 dB SPL, so that the cue level would not be informative for the task.

The participants completed 16 trials for each combination of target position, distractor offset and SNR, totalling 480 trials (1 target position × 5 distractor offsets × 6 SNRs × 16 trials). The trials were randomly ordered and then grouped into six blocks of 80 trials, with the participants encouraged to take self-timed breaks after each block. The entire experiment took approximately two hours to complete. The selection of stimuli and assignment of excerpt to condition were randomized independently for each participant.

**Participants.** A total of 33 participants (22 female, 11 male) completed the experiment. Participant ages were between 18 and 40 years (median age, 23).

**Analysis.** Speech reception thresholds were estimated for each distractor offset in azimuth or elevation using the following procedure. First, word recognition performance as a function of SNR was measured for each participant as in Experiment 1, at each distractor offset. Second, thresholds at each distractor offset and offset direction were estimated by fitting a second-order polynomial to the mean of these values across participants and calculating the SNR granting 50% performance from this curve. The uncertainty in these thresholds was estimated via bootstrap (thresholds were measured from 10,000 bootstrap samples of 33 participants; the graphs of results plot the mean and standard deviation of these thresholds).

**Model experiment.** The models were tested on all combinations of the 1,952 stimuli (all cue, target and two-distractor pairings) with the five distractor offsets and six SNRs (58,560 total combinations). The stimuli were spatialized using a virtual rendering of the loudspeaker array room that the participants were tested in. The sound pressure levels in the human experiment were translated to RMS levels for the model experiment, with 65 dB SPL mapping to an RMS level of 0.036. To approximately equate the model and human testing conditions, pink noise was added to the model stimuli 35 dB lower than the speech to match the background noise level present in the loudspeaker array room used to test the participants. Model speech reception thresholds for each distractor offset and direction were estimated by bootstrapping over the ten model architectures (10,000 samples).

### Experiment 7: measuring the width of the selection 'spotlight'
Experiment 7 measured how spatial release from masking in azimuth depends on the azimuthal position of the attended (target) talker relative to the listener. Eight spatial configurations were used: 2 target azimuths (0° or 90°) × 4 distractor azimuthal offsets (0°, 10°, 30° or 90°). Experiment 7 used the same cue, target and distractor clips as Experiment 6, except that only one distractor talker was included on each trial. All stimuli were presented at 65 dB SPL, with the target and distractor SNR set to 0 dB. Participant and model responses were analysed as in Experiment 1.

**Procedure.** Experiment 7 used the same procedure as Experiment 6, except for the following changes. The participants completed 160 total trials (2 target positions × 4 distractor offsets × 20 trials). Each trial had a different target word. The 160 target words for a participant were randomly sampled from the full set of 488 target words, and their assignment to spatial locations was randomized per participant. The trials were arranged into four blocks of 40 trials. All the trials in a block had the same target position, and the block included ten trials of each distractor offset. The block order and the order of trials within blocks were randomized per participant.

Because the speaker array did not enable left–right symmetric offsets at the 90° azimuth positions (the array did not extend beyond 90° on either side), we used the following procedure to better equate the uncertainty over source positions between the 0° and 90° target conditions. First, distractors were offset in a consistent direction relative to the target (that is, the distractors occurred either to the left or to the right of the target for the entire experiment). The distractor offset direction was alternated across participants. Second, the 0° and 90° target conditions were achieved by rotating the participants to face either the centre or the side of the array, while maintaining the source playback locations. In all trials, the cue and target were presented at the centre of the array, with distractors occurring 0°, 10°, 30° or 90° away from the target in azimuth (to the participant's left or right depending on the offset direction). In blocks testing the 0° target condition, the participants faced the centre speaker. In blocks testing the 90° target condition, the participants were rotated left or right so that the centre speaker was at 90° relative to the participant, and such that the distractor stimuli were played from the same speakers as in the 0° target condition, in the frontal hemifield of the participant. All stimuli were presented at 0° elevation.

**Participants.** A total of 28 participants (16 female, 12 male) completed the experiment. Participant ages were between 19 and 39 years (median age, 26).

**Model experiment.** The models were tested on all combinations of the 1,952 stimuli (cue, target and single-distractor pairs) with the two target positions and four distractor offsets (15,616 total combinations). All model stimuli were RMS-normalized to 0.02. The room simulation and background noise were the same as in Experiment 6.

## Analysis of model locus of attention

To analyse the locus of attention in models, we compared the model's representation of a single-talker target clip with its representation of a mixture composed of the target and a single-talker distractor clip, as a function of whether the model was cued to the target talker or the distractor talker (Fig. 5). The logic for this analysis was that the representations of a single source and a mixture containing that source would be similar if attentional selection for the single source's talker was applied to the mixture. We analysed the similarity between targets and mixtures containing one distractor talker, presented signals diotically to the model, and quantified similarity using Pearson's correlation coefficient. The cue, target and single-distractor speech excerpts curated for Experiment 1 were used as stimuli. To measure any effect of spatial separation on this analysis, we included three azimuthal separation conditions: 0°, 10° (with the target at +5° and the distractor at −5° or vice versa) and 90° (with the target at +45° and the distractor at −45° or vice versa) separation. Activations for all excerpt pairs in each azimuthal separation condition were measured. The following procedure was performed for each cue, target, distractor and mixture in each separation condition. First, representations of the isolated target clip, $x_{\text{target}}^{(l)}$, distractor clip, $x_{\text{distractor}}^{(l)}$, and their mixture, $x_{\text{mixture}}^{(l)}$, were obtained separately from each stage, $l$, using the same cue to designate the target talker for each. Second, we measured Pearson's correlation coefficient between target and mixture as $\text{corr}(x_{\text{target}}^{(l)}, x_{\text{mixture}}^{(l)})$ and between distractor and mixture as $\text{corr}(x_{\text{distractor}}^{(l)}, x_{\text{mixture}}^{(l)})$ where $\text{corr}(x, y)$ is the Pearson correlation:

$$\text{corr}(x, y) = \frac{\sum_{i=1}^{N}(x_i - \bar{x})(y_i - \bar{y})}{\sqrt{\sum_{i=1}^{N}(x_i - \bar{x})^2}\sqrt{\sum_{i=1}^{N}(y_i - \bar{y})^2}}$$

and $i$ indexes the flattened feature dimension of size $N$ = Channels × Frequency × Time. We used 1,000 cue, target and single-distractor pairs to measure correlations (500 same-sex and 500 different-sex distractors). The targets and distractors were presented at 0 dB SNR, and all signals were RMS-normalized to 0.02.

## Aggregate measures of human–model similarity

Human–model behavioural similarity was quantified by comparing human and model performance in each experimental condition, separately for each model (Fig. 6b,c). To ensure that our conclusions were robust to the choice of similarity metric, the following analysis was performed using Pearson's correlation coefficient, RMSE and mutual information as similarity metrics. For each model, we measured the similarity between the mean human behaviour (averaged across experiment participants) and the mean model behaviour, using means computed per experimental condition (for example, a particular distractor type × SNR). For the feature-gain model, the mean model behaviour was taken as the average across the ten network architectures. To calculate mutual information, we generated histograms of human and model performance. To ensure robustness to histogram bin size, mutual information was computed using bin counts of 5, 10, 15 and 20 bins. Error bars on the mutual information for each bin size were obtained via bootstrap (1,000 samples).

The statistical significance of the effect that each alternative architectural constraint (baseline, early-only, late-only or gain-after-normalization architectures) had on overall human–model similarity was assessed by comparing the human–model similarity scores against the distribution of the feature-gain model's human–model similarity scores via sign test. Two-tailed $P$ values are reported, and effect sizes were quantified by measuring the difference between similarity scores over the mean score of the feature-gain model.

## Analysis of spatial cues

The methods for the analysis of spatial cues are described in the caption for Extended Data Fig. 3.

## Analysis of model feature tuning

The methods for the analysis of model feature tuning are described in the caption for Extended Data Fig. 6.

## Sample sizes

For Experiment 1, we ran a pilot experiment with 95 participants, calculated the split-half reliability of the confusion rates across conditions (as this seemed likely to be the least reliable measure from the experiment) and estimated that a sample size of about 200 participants would yield a split-half reliability of 0.9. We performed a similar analysis for Experiment 2 but using a subset of three conditions from the same pilot experiment and estimated that a sample size of about 80 participants would yield a split-half reliability of 0.9. For Experiment 6, we ran a pilot experiment with nine participants and estimated CIs about the threshold for the colocated condition in samples of different sizes. We extrapolated that a sample size of about 30 participants would yield a CI of 1 dB with 90% power. For Experiment 7, we sought to be able to detect a difference in performance for the 10° offset conditions with targets at either 0° or 90°. We assumed a large effect size ($d = 0.8$), yielding a target sample size of 19 to achieve 90% power. Experiments 3–5 used human data from previously published experiments.

## Statistics

Interactions were assessed using a repeated-measures analysis of variance, implemented using the statsmodels[90] (v0.14.2) and Pingouin[91] (v0.5.5) packages for Python (v3.11.5). The significance of the effect of model architectural constrains on human–model similarity was determined using sign tests implemented via the statsmodels and scipy.stats packages. Paired two-tailed $t$-tests used to measure the effect of distractor harmonicity on model performance were implemented using the scipy.stats[92] (v1.11.3) package.

Experiment 6 did not permit a conventional test for an interaction because thresholds were difficult to estimate reliably in individual participants. Instead, we assessed the statistical significance of the interaction between the direction of offset (azimuth or elevation) and the offset magnitude (0°, 10° or 60°) using a permutation test. First, the interaction between conditions was measured by subtracting the marginal contribution of each independent variable:

$$\text{interaction}_{ij} = \text{threshold}_{ij} - \mu_{\text{direction}}^{(i)} - \mu_{\text{offset}}^{(j)} + \mu_{\text{total}}$$

where $i$ indexes the direction of offset (azimuth or elevation), $j$ indexes the offset magnitude, $\mu_{\text{direction}}^{(i)}$ is the average threshold over offsets for direction $i$, $\mu_{\text{offset}}^{(j)}$ is the average threshold over directions for offset $j$ and $\mu_{\text{total}}$ is the average over both direction and offset. An overall interaction effect was then obtained as the sum of squares of these interaction terms:

$$\text{overall interaction} = \sum_{i,j} \text{interaction}_{ij}^2$$

The overall interaction was recomputed 10,000 times with direction of offset and offset magnitude labels permuted at the participant level, obtaining a null distribution used to calculate a $P$ value for the actual overall interaction.

Except where otherwise noted in the figure captions, the error bars in the figures indicate ±1 standard error of the mean across experiment participant results (human results) or across the ten network architectures (model results).

## Reporting summary

Further information on research design is available in the Nature Portfolio Reporting Summary linked to this article.

## Data availability

The processed, anonymized human data and simulation data from this study are available via OSF[93]. There are no restrictions on data availability, and all relevant files are provided in CSV format. No data with mandated deposition are included in this study.

## Code availability

The code used for modelling and data analysis in this study is available via GitHub at https://github.com/mcdermottLab/auditory_attention and is linked from the project OSF repository[93].

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

## Acknowledgements

This research was supported by National Institutes of Health grant R01 DC017970. The funders had no role in study design, data collection and analysis, decision to publish or preparation of the manuscript. We thank A. Thomas for running the human participants in Experiments 6 and 7.

## Author contributions

I.M.G. and J.H.M. conceived the project and wrote the first draft of the paper. I.M.G. and R.P.H. trained the models, designed the experiments and ran the experiments. I.M.G. made the figures. All authors edited the paper.

## Competing interests

The authors declare no competing interests.

## Additional information

**Extended data** is available for this paper at https://doi.org/10.1038/s41562-026-02414-7.

**Correspondence and requests for materials** should be addressed to Ian M. Griffith or Josh H. McDermott.

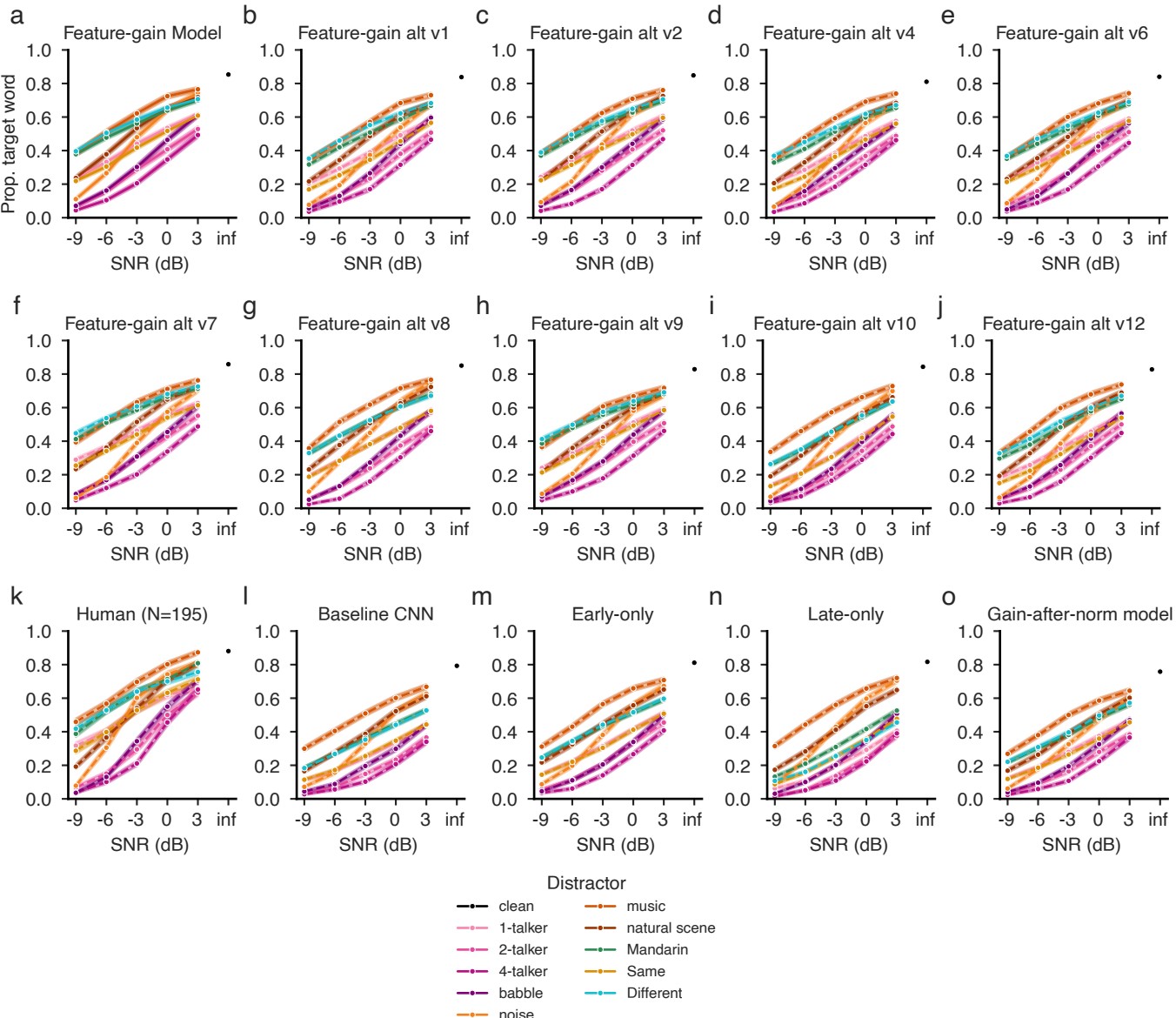

**Extended Data Fig. 1 | Performance of all models on Experiment 1.** Word recognition as a function of signal-to-noise ratio and distractor type for each model architecture. Panels **a-j** show results for each individual feature-gain model architecture (main figures show results averaged across these ten architectures, with different conditions plotted separately to elucidate particular effects of interest). Panel **k** shows results for human participants, replotted from Fig. 2. Panels **l-o** show the baseline, early-only, late-only, and gain-after-normalization architectures which each perform worse with speech-on-speech examples than both humans and the feature-gain models, particularly with one distractor (light-pink, gold, green, and blue lines).

**Extended Data Fig. 2 | Human and model performance for different cue durations.** Results for Experiment 1b, which measured recognition of words spoken by a cued target voice superimposed on a single distractor talker at 0 dB SNR. Error bars plot standard error.

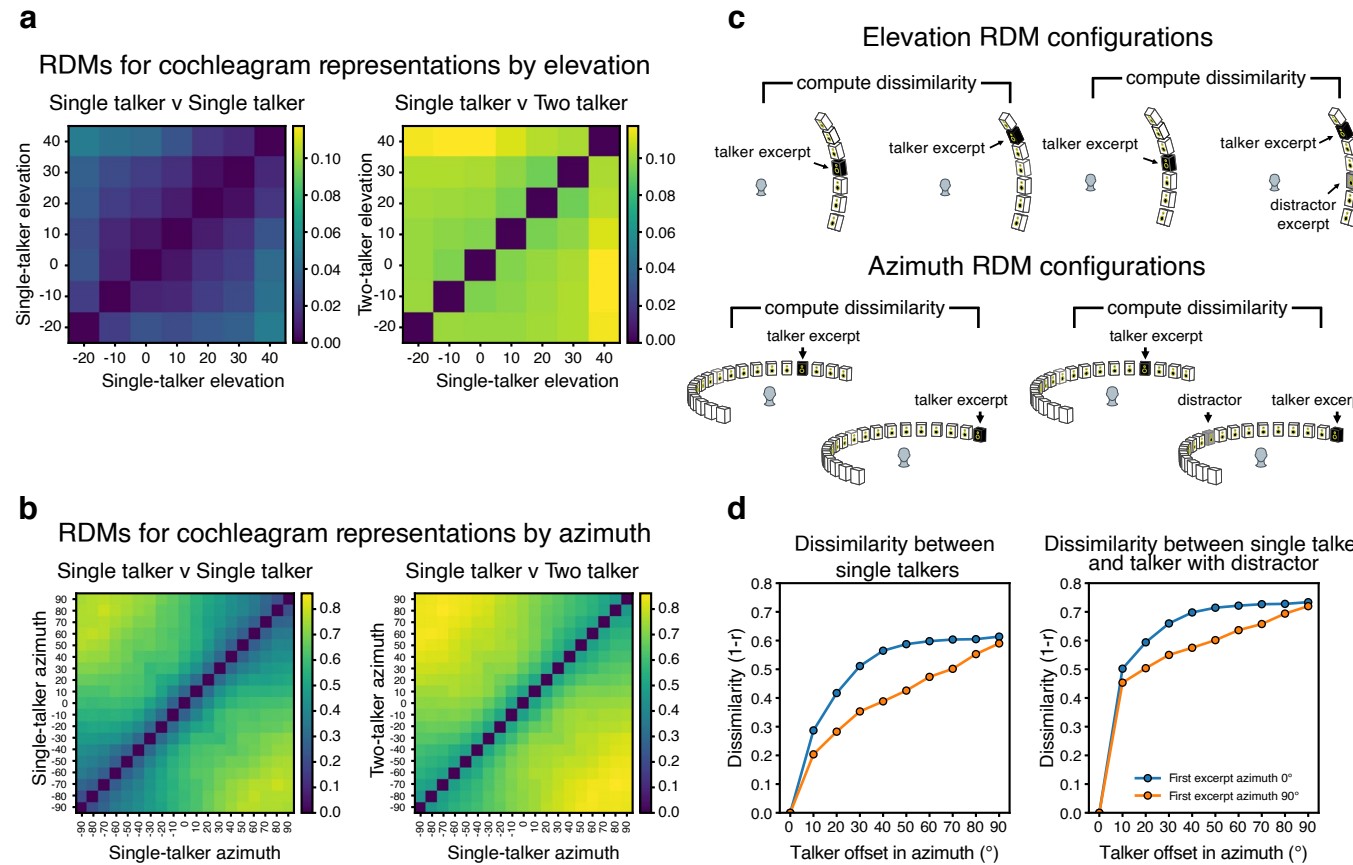

**Extended Data Fig. 3 | Effect of concurrent talker on spatial cues.** We quantified acoustic spatial cues to azimuth and elevation by calculating the dissimilarity between the binaural cochleagram representations of different stimuli as a function of differences in the spatial positions of speech signals spatially rendered in the stimuli. We first measured this dissimilarity for the same speech signal rendered at different positions. We used the same speech signal so that interaural time differences would be reflected in the cochleagram correlation (if the speech signals had not been identical, their fine structure would have been uncorrelated irrespective of any spatial cue from the interaural time difference). We then measured the dissimilarity with a distractor talker superimposed on the second signal. The distractor was positioned in front of the listener (at zero elevation). We expected to see a gradient of dissimilarity as the spatial offset was increased, and that this might be reduced by the distractor talker. Dissimilarity was computed as $1 - r_{xy}$, where $r_{xy}$ is Pearson's correlation between the binaural cochleagrams of signals $x$ and $y$. In each RDM, the $ij_{th}$ entry was computed as the average dissimilarity between representations of signals whose sources were rendered at location $i$ and location $j$. Speech excerpts were drawn from the set of 500 target and same-sex distractors used in the stage of selection analysis, originally selected for Experiment 1. **a.** Results for elevation offset. A gradient of dissimilarity is evident as talker offset is increased. This gradient is less evident in the presence of a distractor talker. To measure dissimilarity in elevation, target

sources were spatially rendered from -20° to 40° elevation (in 10° increments) and 0° azimuth. Dissimilarity was measured using the time-averaged cochleagram, to best reveal spectral cues to elevation. **b.** Results for azimuth offset. A gradient of dissimilarity is evident even in the presence of a distractor talker, indicating that azimuth cues are robust to the presence of a concurrent talker. To measure dissimilarity in azimuth, target sources were spatially rendered from 90° to -90° azimuth (in 10° increments) and 0° elevation. **c.** Schematic of the elevation (top row) and azimuth (bottom row) configurations included in the dissimilarity analysis. Dissimilarity was computed either between representations of single talkers (left column), or between representations of a single talker and representations of a talker with a concurrent distractor (right column). **d.** Comparison of dissimilarity gradient for a target talker at the midline (0 degrees) vs. at the periphery (90 degrees). The dissimilarity increases rapidly for the talker at the midline, and more gradually for the talker at the periphery, consistent with the idea that binaural spatial cues permit a tighter spatial attentional focus at the midline compared to the periphery. We emphasize that these analyses do not reveal anything about whether location-related differences in cochlear representations could actually be used for attentional selection (for this you need a model of behavior); they are nonetheless useful in providing some intuitive understanding.

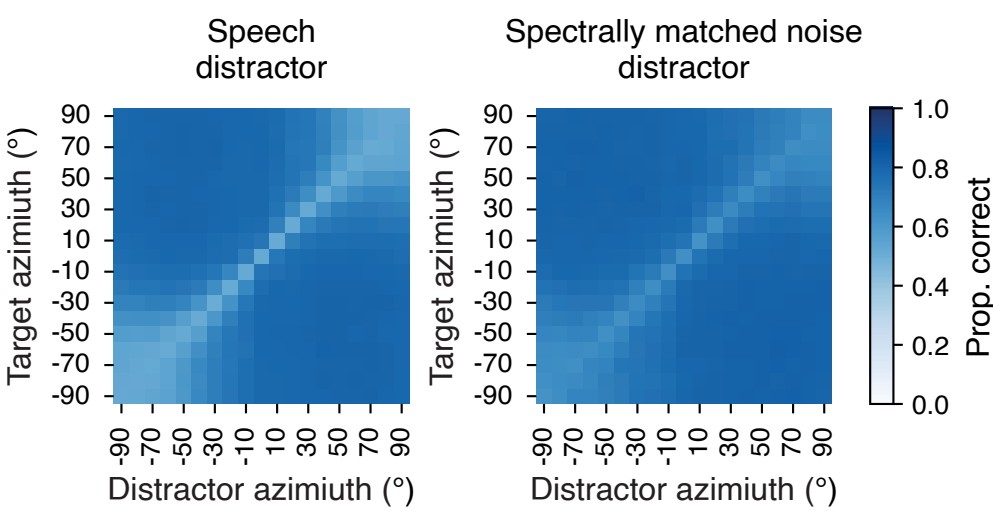

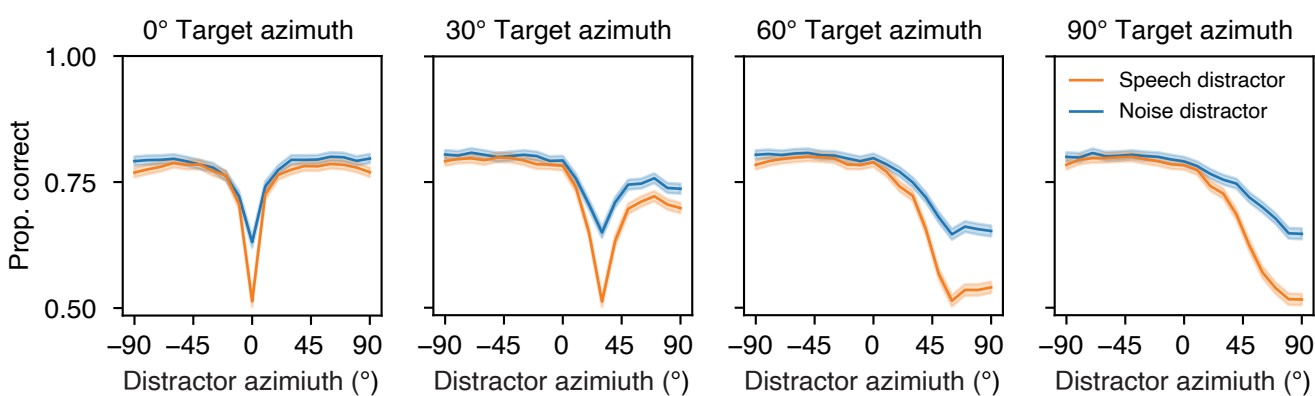

**Extended Data Fig. 4 | Effect of target and distractor spatial position on model performance for speech and noise distractors. a.** Word recognition performance for the main feature-gain model (arch_v00 in Extended Data Table 1) is plotted as a function of target and distractor positions in azimuth, for both single-talker distractors and speech-shaped noise distractors. As in Fig. 4, results are averaged across front and back positions, and elevations, for ease of visualization. **b.** Data from (a) plotted as line graphs for four example target positions, to enable direct comparison of performance for speech and noise distractors. Shaded regions plot confidence intervals (95%) obtained via bootstrap. Performance with noise distractors varies less with spatial position than performance with speech distractors, as in humans[26]. The presumptive explanation is that voice features alone are sufficient to select speech from noise, such that there is less benefit from spatial attention. There is nonetheless some spatial benefit on performance, consistent with prior results in human listeners[26]. This spatial dependence could be explained by a combination of changes in signal-to-noise ratio at the ear (which improves at the ear closer to the target as the distractor is moved away) and binaural effects related to masking[41]. However, the larger effect for speech distractors suggests a contribution of spatial attention in this setting. The similar spatial dependence for speech and noise distractors presumably reflects the spatial dependence of the underlying binaural cues, which also cause localization acuity to be worse in the periphery.

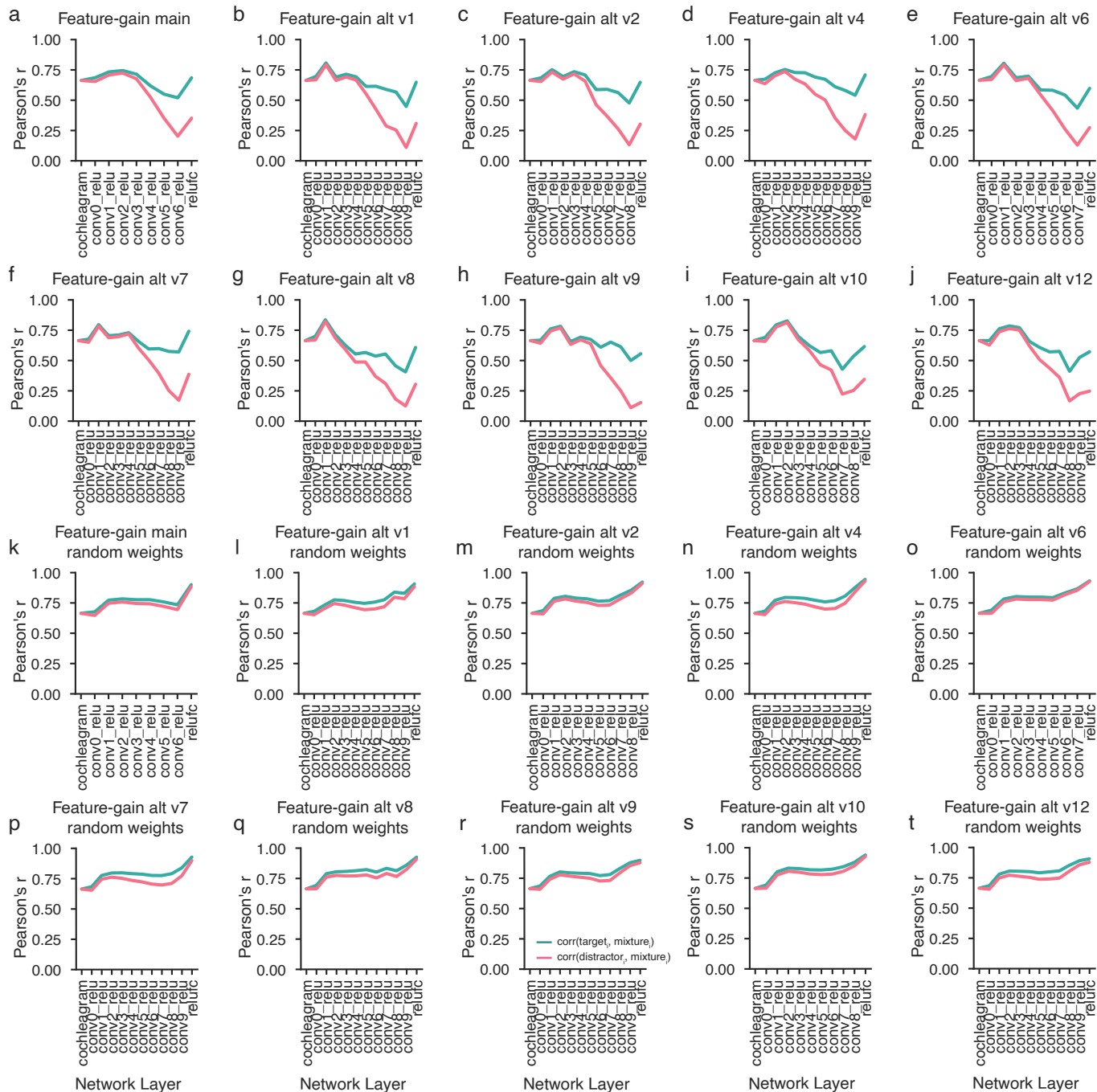

**Extended Data Fig. 5 | Stage-of-selection analysis for each model architecture.** Analysis of Fig. 5, plotted separately for each model architecture for the configuration where target and distractor are co-located. All feature-gain architectures displayed fairly similar patterns of target enhancement at late model stages (panels **a-j**). Each architecture initialized with random weights did not show the same trend (panels **k-t**).

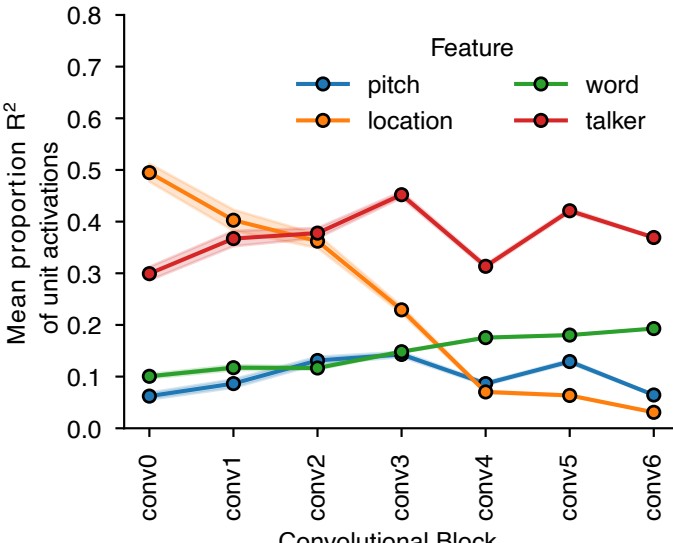

**Extended Data Fig. 6 | Feature tuning across model stages.** We measured model unit activations from the best-performing feature-gain architecture for a set of speech excerpts rendered at different spatial locations. To assess whether unit responses could be explained by the pitch, location, word, or talker of a speech excerpt, we ran four one-factor ANOVAs on each unit's responses, one for each of the four variables, and calculated the variance explained by each variable from the sums of squares. This analysis was performed using the ordinary least squares and linear model ANOVA methods of the statsmodels[90] python package. The graph plots the mean variance explained for each feature at each model stage. The two dimensions that are useful for selection (location and talker) account for considerable response variance in early model stages. Tuning to words is weak in early stages and increases towards the end of the model. Tuning to pitch (specifically, the mean f0 of the voiced segments of the speech excerpt), which is one of many features defining the talker's voice, is evident throughout the model.

Location tuning was less prevalent in deep model stages, whereas talker tuning persisted, potentially because talker cues are extracted over a larger number of hierarchical stages. Details of the analysis: we used 5,000 speech excerpts from the Word-Speaker-Noise dataset[74,79] as this dataset had a similar vocabulary to our training task, but was constructed to balance talkers across word classes. We spatialized speech excerpts to 0°, ±10°, ±45°, and 90° azimuth and 0° elevation for 7 total locations. The subset of the Word-Speaker-Noise dataset that we used contained 431 talkers and 776 words. All of the 5,000 speech excerpts were spatialized to each of the 7 locations. We used the average F0 of the voiced segments of an excerpt (estimated using pyin[94]) to capture pitch. Pitch was discretized into 54 bins, determined by calculating the appropriate bin width using the Freedman-Diaconis method: bin width $= 2 \times \frac{IQR}{n^{\left(\frac{1}{3}\right)}}$ where IQR is the interquartile range over average F0 and $n$ is the number of excerpts.

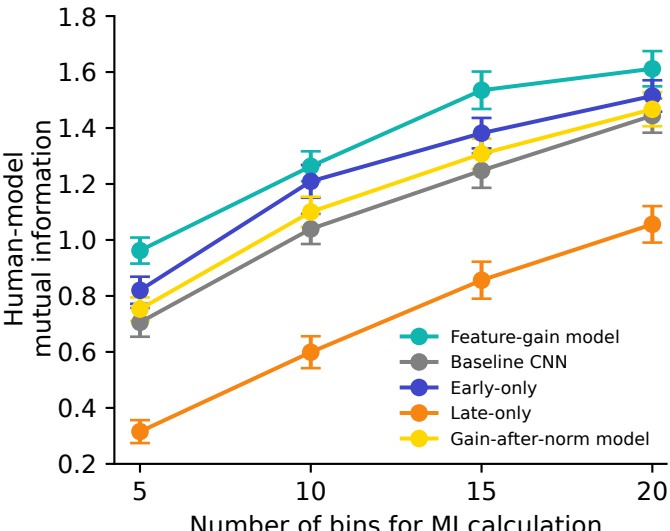

**Extended Data Fig. 7 | Analysis of human-model similarity using mutual information.** Mutual information was calculated using histograms of human and model proportion correct in different experimental conditions. We performed the analysis for different numbers of histogram bins to ensure that the results were robust to the bin width. Error bars plot standard error, derived via bootstrap.

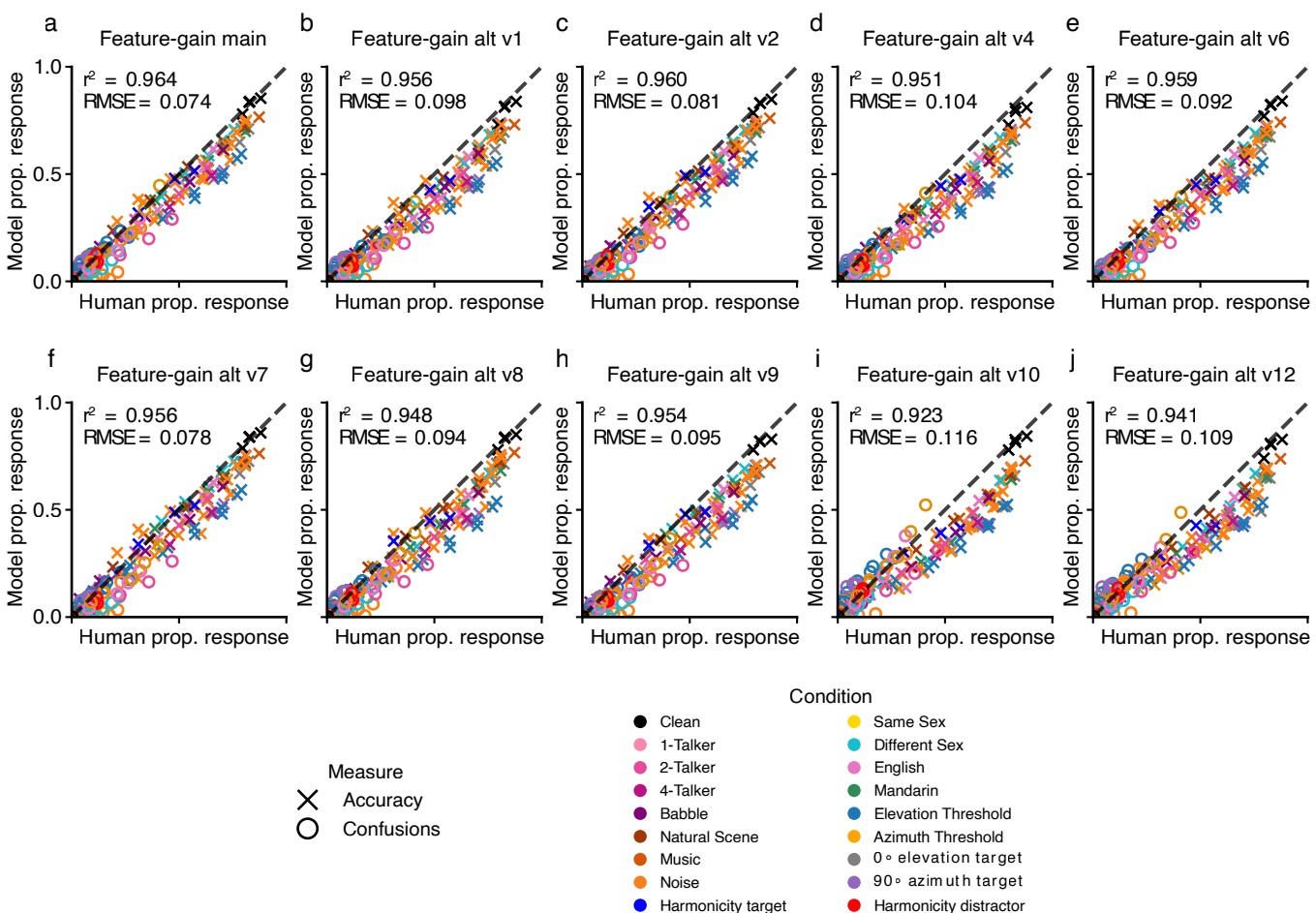

**Extended Data Fig. 8 | Comparison of human and model performance for each model architecture.** Panels **a-j** show scatter plots of model vs. human performance for each individual feature-gain architecture.

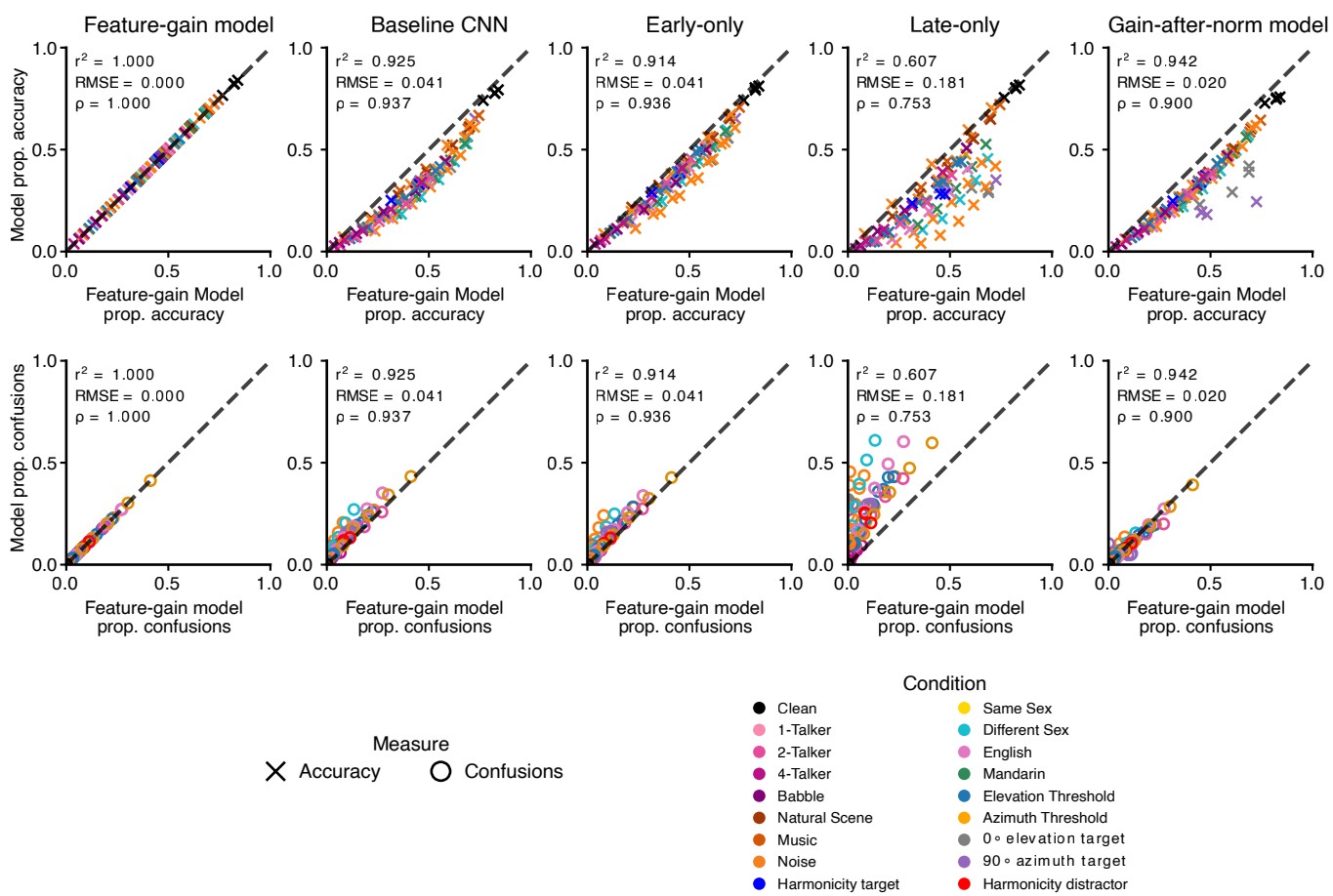

**Extended Data Fig. 9 | Comparison of performance of different model architectures.** Scatter plots of alternative model performance vs. that of the feature-gain model.

## Extended Data Table 1 | Neural network architectures for feature-gain models

| Architecture | arch_v00 | arch_v01 | arch_v02 | arch_v04 | arch_v06 | arch_v07 | arch_v08 | arch_v09 | arch_v10 | arch_v12 |
|---|---|---|---|---|---|---|---|---|---|---|
| **Operation** | Input [40,20000,2] | Input [40,20000,2] | Input[40,20000,2] | Input [40,20000,2] | Input [40,20000,2] | Input [40,20000,2] | Input [40,20000,2] | Input [40,20000,2] | Input [40,20000,2] | Input [40,20000,2] |
| 1 | Coch Lnorm | Coch Lnorm | Coch Lnorm | Coch Lnorm | Coch Lnorm | Coch Lnorm | Coch Lnorm | Coch Lnorm | Coch Lnorm | Coch Lnorm |
| 2 | Feature-gains 0 | Feature-gains 0 | Feature-gains 0 | Feature-gains 0 | Feature-gains 0 | Feature-gains 0 | Feature-gains 0 | Feature-gains 0 | Feature-gains 0 | Feature-gains 0 |
| 3 | Lnorm0 | Lnorm0 | Lnorm0 | Lnorm0 | Lnorm0 | Lnorm0 | Lnorm0 | Lnorm0 | Lnorm0 | Lnorm0 |
| 4 | conv0 [2, 34, 32] | conv0 [1, 59, 32] | conv0 [3, 55, 64] | conv0 [3, 55, 64] | conv0 [1, 22, 32] | conv0 [2, 71, 32] | conv0 [1, 32, 32] | conv0 [2, 11, 32] | conv0 [1, 16, 32] | conv0 [3, 75, 32] |
| 5 | Relu0 | Relu0 | Relu0 | Relu0 | Relu0 | Relu0 | Relu0 | Relu0 | Relu0 | Relu0 |
| 6 | HannPool0 [2,4] | HannPool0 [2,1] | HannPool0 [2,5] | HannPool0 [2,3] | HannPool0 [2,3] | HannPool0 [2,5] | HannPool0 [2,6] | HannPool0 [2,2] | HannPool0 [2,3] | HannPool0 [2,2] |
| 7 | Feature-gains 1 | Feature-gains 1 | Feature-gains 1 | Feature-gains 1 | Feature-gains 1 | Feature-gains 1 | Feature-gains 1 | Feature-gains 1 | Feature-gains 1 | Feature-gains 1 |
| 8 | Lnorm1 | Lnorm1 | Lnorm1 | Lnorm1 | Lnorm1 | Lnorm1 | Lnorm1 | Lnorm1 | Lnorm1 | Lnorm1 |
| 9 | conv1 [2, 14, 64] | conv1 [2, 72, 64] | conv1 [2, 74, 64] | conv1 [2, 32,128] | conv1 [1, 34, 64] | conv1 [1, 33, 64] | conv1 [3, 47, 64] | conv1 [3, 60, 64] | conv1 [2, 79, 64] | conv1 [2, 44, 64] |
| 10 | Relu1 | Relu1 | Relu1 | Relu1 | Relu1 | Relu1 | Relu1 | Relu1 | Relu1 | Relu1 |
| 11 | HannPool1 [2,4] | HannPool1 [2,5] | HannPool1 [2,6] | HannPool1 [2,4] | HannPool1 [2,5] | HannPool1 [2,1] | HannPool1 [2,4] | HannPool1 [2,6] | HannPool1 [2,5] | HannPool1 [2,6] |
| 12 | Feature-gains 2 | Feature-gains 2 | Feature-gains 2 | Feature-gains 2 | Feature-gains 2 | Feature-gains 2 | Feature-gains 2 | Feature-gains 2 | Feature-gains 2 | Feature-gains 2 |
| 13 | Lnorm2 | Lnorm2 | Lnorm2 | Lnorm2 | Lnorm2 | Lnorm2 | Lnorm2 | Lnorm2 | Lnorm2 | Lnorm2 |
| 14 | conv2 [5, 5, 256] | conv2 [4, 3, 128] | conv2 [5, 4, 128] | conv2 [5, 3, 256] | conv2 [3, 5, 128] | conv2 [3, 6, 128] | conv2 [4, 4, 128] | conv2 [4, 5, 128] | conv2 [5, 4, 128] | conv2 [5, 4, 128] |
| 15 | Relu2 | Relu2 | Relu2 | Relu2 | Relu2 | Relu2 | Relu2 | Relu2 | Relu2 | Relu2 |
| 16 | HannPool2 [1,5] | HannPool2 [1,3] | HannPool2 [1,2] | HannPool2 [1,3] | HannPool2 [1,3] | HannPool2 [1,2] | HannPool2 [1,3] | HannPool2 [1,1] | HannPool2 [1,2] | HannPool2 [1,2] |
| 17 | Feature-gains 3 | Feature-gains 3 | Feature-gains 3 | Feature-gains 3 | Feature-gains 3 | Feature-gains 3 | Feature-gains 3 | Feature-gains 3 | Feature-gains 3 | Feature-gains 3 |
| 18 | Lnorm3 | Lnorm3 | Lnorm3 | Lnorm3 | Lnorm3 | Lnorm3 | Lnorm3 | Lnorm3 | Lnorm3 | Lnorm3 |
| 19 | conv3 [5, 5, 512] | conv3 [6, 3, 256] | conv3 [6, 5, 256] | conv3 [5, 4, 512] | conv3 [6, 4, 256] | conv3 [5, 3, 256] | conv3 [4, 6, 256] | conv3 [4, 4, 256] | conv3 [6, 6, 256] | conv3 [4, 6, 256] |
| 20 | Relu3 | Relu3 | Relu3 | Relu3 | Relu3 | Relu3 | Relu3 | Relu3 | Relu3 | Relu3 |
| 21 | HannPool3 [1,4] | HannPool3 [1,3] | HannPool3 [1,2] | HannPool3 [1,1] | HannPool3 [1,3] | HannPool3 [1,2] | HannPool3 [1,3] | HannPool3 [1,3] | HannPool3 [1,2] | HannPool3 [1,3] |
| 22 | Feature-gains 4 | Feature-gains 4 | Feature-gains 4 | Feature-gains 4 | Feature-gains 4 | Feature-gains 4 | Feature-gains 4 | Feature-gains 4 | Feature-gains 4 | Feature-gains 4 |
| 23 | Lnorm4 | Lnorm4 | Lnorm4 | Lnorm4 | Lnorm4 | Lnorm4 | Lnorm4 | Lnorm4 | Lnorm4 | Lnorm4 |
| 24 | conv4 [6, 6, 512] | conv4 [4, 4, 512] | conv4 [3, 3, 512] | conv4 [6, 6, 512] | conv4 [3, 4, 512] | conv4 [6, 6, 512] | conv4 [3, 6, 512] | conv4 [3, 4, 512] | conv4 [6, 4, 512] | conv4 [5, 6, 512] |
| 25 | Relu4 | Relu4 | Relu4 | Relu4 | Relu4 | Relu4 | Relu4 | Relu4 | Relu4 | Relu4 |
| 26 | HannPool4 [1,1] | HannPool4 [1,2] | HannPool4 [1,1] | HannPool4 [1,1] | HannPool4 [1,1] | HannPool4 [1,3] | HannPool4 [1,2] | HannPool4 [1,3] | HannPool4 [1,3] | HannPool4 [1,1] |
| 27 | Feature-gains 5 | Feature-gains 5 | Feature-gains 5 | Feature-gains 5 | Feature-gains 5 | Feature-gains 5 | Feature-gains 5 | Feature-gains 5 | Feature-gains 5 | Feature-gains 5 |
| 28 | Lnorm5 | Lnorm5 | Lnorm5 | Lnorm5 | Lnorm5 | Lnorm5 | Lnorm5 | Lnorm5 | Lnorm5 | Lnorm5 |
| 29 | conv5 [5, 5, 512] | conv5 [5, 3, 512] | conv5 [4, 5, 512] | conv5 [4, 5, 512] | conv5 [6, 4, 512] | conv5 [3, 5, 512] | conv5 [4, 3, 512] | conv5 [4, 3, 512] | conv5 [3, 5, 512] | conv5 [5, 5, 512] |
| 30 | Relu5 | Relu5 | Relu5 | Relu5 | Relu5 | Relu5 | Relu5 | Relu5 | Relu5 | Relu5 |
| 31 | HannPool5 [1,1] | HannPool5 [1,3] | HannPool5 [1,2] | HannPool5 [1,1] | HannPool5 [1,3] | HannPool5 [1,1] | HannPool5 [1,1] | HannPool5 [1,1] | HannPool5 [1,2] | HannPool5 [1,3] |
| 32 | Feature-gains 6 | Feature-gains 6 | Feature-gains 6 | Feature-gains 6 | Feature-gains 6 | Feature-gains 6 | Feature-gains 6 | Feature-gains 6 | Feature-gains 6 | Feature-gains 6 |
| 33 | Lnorm6 | Lnorm6 | Lnorm6 | Lnorm6 | Lnorm6 | Lnorm6 | Lnorm6 | Lnorm6 | Lnorm6 | Lnorm6 |
| 34 | conv6 [6, 6, 512] | conv6 [5, 4, 512] | conv6 [6, 6, 512] | conv6 [5, 3, 512] | conv6 [6, 6, 512] | conv6 [6, 3, 512] | conv6 [4, 4, 512] | conv6 [5, 3, 512] | conv6 [4, 4, 512] | conv6 [3, 3, 512] |
| 35 | Relu6 | Relu6 | Relu6 | Relu6 | Relu6 | Relu6 | Relu6 | Relu6 | Relu6 | Relu6 |
| 36 | HannPool6 [2,4] | HannPool6 [1,2] | HannPool6 [1,3] | HannPool6 [1,3] | HannPool6 [1,2] | HannPool6 [1,2] | HannPool6 [1,2] | HannPool6 [1,3] | HannPool6 [1,2] | HannPool6 [1,2] |
| 37 | Feature-gains fc | Feature-gains 7 | Feature-gains 7 | Feature-gains 7 | Feature-gains 7 | Feature-gains 7 | Feature-gains 7 | Feature-gains 7 | Feature-gains 7 | Feature-gains 7 |
| 38 | flatten | Lnorm7 | Lnorm7 | Lnorm7 | Lnorm7 | Lnorm7 | Lnorm7 | Lnorm7 | Lnorm7 | Lnorm7 |
| 39 | fc [512] | conv7 [3, 5, 512] | conv7 [6, 6, 512] | conv7 [6, 4, 512] | conv7 [6, 4, 512] | conv7 [4, 4, 512] | conv7 [4, 4, 512] | conv7 [4, 6, 512] | conv7 [5, 3, 512] | conv7 [6, 6, 512] |
| 40 | Relufc | Relu7 | Relu7 | Relu7 | Relu7 | Relu7 | Relu7 | Relu7 | Relu7 | Relu7 |
| 41 | dropout | HannPool7 [1,3] | HannPool7 [1,2] | HannPool7 [1,2] | HannPool7 [1,3] | HannPool7 [1,1] | HannPool7 [1,3] | HannPool7 [1,3] | HannPool7 [1,3] | HannPool7 [1,3] |
| 42 | fc [800] | Feature-gains 8 | Feature-gains 8 | Feature-gains fc | Feature-gains 8 | Feature-gains 8 | Feature-gains 8 | Feature-gains 8 | Feature-gains 8 | Feature-gains 8 |
| 43 | | Lnorm8 | Lnorm8 | flatten | Lnorm8 | Lnorm8 | Lnorm8 | Lnorm8 | Lnorm8 | Lnorm8 |
| 44 | | conv8 [6, 4, 512] | conv8 [6, 4, 512] | fc [512] | conv8 [3, 4, 512] | conv8 [5, 5, 512] | conv8 [3, 3, 512] | conv8 [5, 6, 512] | conv8 [5, 6, 512] | conv8 [4, 4, 512] |
| 45 | | Relu8 | Relu8 | Relufc | Relu8 | Relu8 | Relu8 | Relu8 | Relu8 | Relu8 |
| 46 | | HannPool8 [1,1] | HannPool8 [1,2] | dropout | HannPool8 [1,3] | HannPool8 [1,3] | HannPool8 [1,2] | HannPool8 [1,2] | HannPool8 [1,2] | HannPool8 [1,2] |
| 47 | | Feature-gains fc | Feature-gains 9 | fc [800] | Feature-gains 9 | Feature-gains 9 | Feature-gains 9 | Feature-gains 9 | Feature-gains fc | Feature-gains 9 |
| 48 | | flatten | Lnorm9 | | Lnorm9 | Lnorm9 | Lnorm9 | Lnorm9 | flatten | Lnorm9 |
| 49 | | fc [512] | conv9 [6, 3, 512] | | conv9 [3, 3, 512] | conv9 [5, 4, 512] | conv9 [5, 6, 512] | conv9 [5, 6, 512] | fc [512] | conv9 [5, 5, 512] |
| 50 | | Relufc | Relu9 | | Relu9 | Relu9 | Relu9 | Relu9 | Relufc | Relu9 |
| 51 | | dropout | HannPool9 [1,3] | | HannPool9 [1,2] | HannPool9 [1,3] | HannPool9 [1,3] | HannPool9 [1,1] | dropout | HannPool9 [1,1] |
| 52 | | fc [800] | Feature-gains fc | | Feature-gains fc | Feature-gains fc | Feature-gains fc | Feature-gains fc | fc [800] | Feature-gains fc |
| 53 | | | flatten | | flatten | flatten | flatten | flatten | | flatten |
| 54 | | | fc [512] | | fc [512] | fc [512] | fc [512] | fc [512] | | fc [512] |
| 55 | | | Relufc | | Relufc | Relufc | Relufc | Relufc | | Relufc |
| 56 | | | dropout | | dropout | dropout | dropout | dropout | | dropout |
| 57 | | | fc [800] | | fc [800] | fc [800] | fc [800] | fc [800] | | fc [800] |
| 58 | | | | | | | | | | |

Grey bands indicate stages where feature-gain operations occurred. Legend: • *Lnorm*: layer normalization operation • conv [*h*, *w*, *k*]: convolutional layer with *h* = kernel height (frequency dimension), *w* = kernel width (time dimension), and *k* = number of kernels • Relu: rectified linear unit activation function • HannPool [$s_f$, $s_t$]: Hanning window weighted averaging pooling operation with stride $s_f$ in the frequency dimension and stride $s_t$ in the time dimension • flatten: reshape operation to map a multidimensional tensor to a vector • dropout: dropout regularization with 50% dropout rate • fc [*N*]: fully-connected layer with *N* units.

# Reporting Summary

## Statistics

For all statistical analyses, confirm that the following items are present in the figure legend, table legend, main text, or Methods section.

| n/a | Confirmed | |
|---|---|---|
| ☐ | ☒ | The exact sample size (*n*) for each experimental group/condition, given as a discrete number and unit of measurement |
| ☐ | ☒ | A statement on whether measurements were taken from distinct samples or whether the same sample was measured repeatedly |
| ☐ | ☒ | The statistical test(s) used AND whether they are one- or two-sided<br>*Only common tests should be described solely by name; describe more complex techniques in the Methods section.* |
| ☒ | ☐ | A description of all covariates tested |
| ☒ | ☐ | A description of any assumptions or corrections, such as tests of normality and adjustment for multiple comparisons |
| ☐ | ☒ | A full description of the statistical parameters including central tendency (e.g. means) or other basic estimates (e.g. regression coefficient) AND variation (e.g. standard deviation) or associated estimates of uncertainty (e.g. confidence intervals) |
| ☐ | ☒ | For null hypothesis testing, the test statistic (e.g. *F*, *t*, *r*) with confidence intervals, effect sizes, degrees of freedom and *P* value noted<br>*Give P values as exact values whenever suitable.* |
| ☒ | ☐ | For Bayesian analysis, information on the choice of priors and Markov chain Monte Carlo settings |
| ☒ | ☐ | For hierarchical and complex designs, identification of the appropriate level for tests and full reporting of outcomes |
| ☐ | ☒ | Estimates of effect sizes (e.g. Cohen's *d*, Pearson's *r*), indicating how they were calculated |

*Our web collection on statistics for biologists contains articles on many of the points above.*

## Software and code

Policy information about availability of computer code

| | |
|---|---|
| Data collection | Modeling: computational modeling in Python version 3.11.5, Pytorch version 2.1.1, and Pytorch Lighting version 2.1.1. The code associated with this study is available at the project repository: https://github.com/mcdermottLab/auditory_attention<br><br>Human experiments: experiments used standard publicly available code packages; jsPsych 7.3 for web browser experiments posted on the Prolific platform, and Python version 3.12 for in-lab experiments. |
| Data analysis | Code made use of Python 3.11.5 packages: Pandas 2.1.3, Numpy 1.26.0, Matplotlib 3.8.0, Statsmodels 0.14.2, Seaborn 0.13.2, Scipy 1.11.3, Pingouin 0.5.5. The code associated with this study is available at the project repository: https://github.com/mcdermottLab/auditory_attention  (Python environment to run code is included with the project repository). |

For manuscripts utilizing custom algorithms or software that are central to the research but not yet described in published literature, software must be made available to editors and reviewers. We strongly encourage code deposition in a community repository (e.g. GitHub). See the Nature Portfolio guidelines for submitting code & software for further information.

## Data

Policy information about availability of data

All manuscripts must include a data availability statement. This statement should provide the following information, where applicable:

- Accession codes, unique identifiers, or web links for publicly available datasets
- A description of any restrictions on data availability
- For clinical datasets or third party data, please ensure that the statement adheres to our policy

> All model and human data are available at https://osf.io/wjzvu

## Research involving human participants, their data, or biological material

Policy information about studies with human participants or human data. See also policy information about sex, gender (identity/presentation), and sexual orientation and race, ethnicity and racism.

| | |
|---|---|
| Reporting on sex and gender | Gender was self-reported by study participants. A gender-based analysis was not performed as we were interested in the comparison between humans and computational models and did not investigate individual differences in human behavior. |
| Reporting on race, ethnicity, or other socially relevant groupings | Race and ethnicity information was collected via voluntary self-report but was not considered in any analyses as we were interested in the comparison between humans and computational models and did not investigate individual differences in human behavior. |
| Population characteristics | See "Behavioural & social sciences study design" |
| Recruitment | Online participants were recruited on the Prolific platform with a geographic filter set to exclude individuals outside of the United States. Participants were invited to perform "word recognition in noise" studies. In-person participants were recruited from the Cambridge, MA area via a mailing list. |
| Ethics oversight | The study was approved by the Committee on the Use of Humans as Experimental Subjects at MIT. |

Note that full information on the approval of the study protocol must also be provided in the manuscript.

# Field-specific reporting

Please select the one below that is the best fit for your research. If you are not sure, read the appropriate sections before making your selection.

☐ Life sciences  ☒ Behavioural & social sciences  ☐ Ecological, evolutionary & environmental sciences

For a reference copy of the document with all sections, see nature.com/documents/nr-reporting-summary-flat.pdf

# Behavioural & social sciences study design

All studies must disclose on these points even when the disclosure is negative.

| | |
|---|---|
| Study description | This quantitative study measured human abilities to recognize words in different conditions. Human data (from either online or in-person participants) was averaged across participants and compared to model performance. |
| Research sample | For the diotic word recognition experiments, online participants were used for convenience. We screened for self-reported normal hearing and did not screen for age or self-reported gender. Based on our previous experience running online experiments, this sample was representative of typical online participant cohorts. For the in-person experiments (in which sounds were presented from a speaker array), the participants were drawn from the Cambridge community. The sample is representative of normal hearing humans experienced with listening experiments. <br><br>Demographic information: <br>Experiment 1: 195 participants (98 female, 92 male, 4 non-binary, 1 no-report) between ages 18 and 71 (median 33) years. <br>Experiment 1b: 84 participants (43 female, 38 male, 3 non-binary) between ages 20 and 40 (median 32) years. <br>Experiment 2: 90 participants (43 female, 47 male) between ages 19 and 64 (median 34.5) years. <br>Experiment 3: Reproduced from Saddler et al., 2024; 47 participants (24 female, 23 male) between ages 23 and 59 (median 39) years. <br>Experiment 4: Reproduced from Byrne et al., (2023); 18 participants (11 female, 6 male, 1 non-specified; aged 18-40 years). <br>Experiment 5: Reproduced from Freyman et al., (1999); Originally reported as "56 young college students". <br>Experiment 6: 33 participants (22 female, 10 male) between ages 18 and 40 years (median age = 23). <br>Experiment 7: 28 participants (16 female, 12 male) between ages 19 and 39 years (median age = 26). |
| Sampling strategy | We used convenience sampling because we sought to characterize normal hearing listeners. We cannot exclude the possibility that self-selection biases could have resulted in a somewhat non-representative sample (because participants knew they were volunteering for a hearing experiment). |

Target sample sizes were determined via power analyses. For Experiment 1, we ran a pilot experiment with 95 participants, calculated the split-half reliability of the confusion rates across conditions (as this seemed likely to be the least reliable measure from the experiment), and estimated that a sample size of about 200 participants would yield a split-half reliability of .9. We performed a similar analysis for Experiment 2 but using a subset of 3 conditions from the same pilot experiment, and estimated that a sample size of about 80 participants would yield a split-half reliability of .9. For Experiment 6, we ran a pilot experiment with 9 participants, and estimated confidence intervals about the threshold for the co-located condition in samples of different sizes. We extrapolated that a sample size of about 30 participants would yield a confidence interval of 1 dB with 90% power. For Experiment 7, we sought to be able to detect a difference in performance for the 10-degree offset conditions with targets at either 0 or 90 degrees. We assumed a large effect size (d=0.8), yielding a target sample size of 19 to achieve 90% power. Experiments 3-5 used human data from previously published experiments.

Data collection

Online participants completed experiments in a web browser via surveys designed in jsPsych that played audio and recorded typed responses. In-person participants were played sounds from an array of loudspeakers and typed responses into a keypad. No one was present besides the participant and the researcher. The researcher was not blinded to the experimental condition or study hypothesis.

Timing

Data for Experiments 1 - 3 was collected online between May 2023 and Aug 2024. Data for Experiment 1b was collected in Sept 2025. Data for Experiments 6 and 7 were collected between December 2023 and February 2025.

Data exclusions

309 online participants were excluded from analyses for failing to pass the headphone check or performing poorly on independent catch trials. These were pre-established criteria meant to exclude participants that did not comply with online experiment instructions.

Non-participation

No in-person participant dropped out / declined participation. An unknown number of online participants on the Prolific platform dropped out, primarily due to technical difficulties (e.g., loss of internet connection or audio failing to play in the web browser). Participants could quit the experiments at any time without providing a reason.

Randomization

Participants were not allocated into experimental groups.

# Reporting for specific materials, systems and methods

We require information from authors about some types of materials, experimental systems and methods used in many studies. Here, indicate whether each material, system or method listed is relevant to your study. If you are not sure if a list item applies to your research, read the appropriate section before selecting a response.

## Materials & experimental systems

| n/a | Involved in the study |
|-----|----------------------|
| ☒ | Antibodies |
| ☒ | Eukaryotic cell lines |
| ☒ | Palaeontology and archaeology |
| ☒ | Animals and other organisms |
| ☒ | Clinical data |
| ☒ | Dual use research of concern |
| ☒ | Plants |

## Methods

| n/a | Involved in the study |
|-----|----------------------|
| ☒ | ChIP-seq |
| ☒ | Flow cytometry |
| ☒ | MRI-based neuroimaging |

## Plants

Seed stocks

*Report on the source of all seed stocks or other plant material used. If applicable, state the seed stock centre and catalogue number. If plant specimens were collected from the field, describe the collection location, date and sampling procedures.*

Novel plant genotypes

*Describe the methods by which all novel plant genotypes were produced. This includes those generated by transgenic approaches, gene editing, chemical/radiation-based mutagenesis and hybridization. For transgenic lines, describe the transformation method, the number of independent lines analyzed and the generation upon which experiments were performed. For gene-edited lines, describe the editor used, the endogenous sequence targeted for editing, the targeting guide RNA sequence (if applicable) and how the editor was applied.*

Authentication

*Describe any authentication procedures for each seed stock used or novel genotype generated. Describe any experiments used to assess the effect of a mutation and, where applicable, how potential secondary effects (e.g. second site T-DNA insertions, mosiacism, off-target gene editing) were examined.*

