## [Peer Review File · Nature Human Behaviour]

Optimized feature gains explain and predict successes and failures of human selective listening

Corresponding Author: Dr Josh McDermott

Version 0:

Decision Letter:

16th July 2025

Dear Dr McDermott,

Thank you for submitting your manuscript, entitled "Optimized feature gains explain and predict successes and failures of human selective listening".

I'm pleased to say that we've decided to send your manuscript out to peer review. However, before we can do so, we ask that you prepare the code used in the study and make it available in a form that the peer reviewers can access during the review process.

Therefore, I am inviting you to revise the manuscript to include a link to the code. Please let me know if you foresee any problem with this plan. Please also note that to preserve reviewer anonymity, the code should not require users to login to access it.

Link Redacted

We look forward to seeing the revised manuscript and thank you for the opportunity to review your work. Please do not hesitate to contact me if you have any questions or would like to discuss these revisions further.

Sincerely,

Nature Human Behaviour

Version 1:

Decision Letter:

5th September 2025

Dear Dr McDermott,

Thank you once again for your manuscript, entitled "Optimized feature gains explain and predict successes and failures of human selective listening," and for your patience during the peer review process.

Your manuscript has now been evaluated by 3 reviewers, whose comments are included at the end of this letter. Although

the reviewers find your work to be of interest, they also raise some important concerns. We are very interested in the possibility of publishing your study in Nature Human Behaviour, but would like to consider your response to these concerns in the form of a revised manuscript before we make a decision on publication.

In sum, we invite you to revise your manuscript taking into account all reviewer and editor comments. We are committed to providing a fair and constructive peer-review process. Do not hesitate to contact us if there are specific requests from the reviewers that you believe are technically impossible or unlikely to yield a meaningful outcome.

We hope to receive your revised manuscript within two months. I would be grateful if you could contact us as soon as possible if you foresee difficulties with meeting this target resubmission date.

- Include a "Response to the editors and reviewers" document detailing, point-by-point, how you addressed each editor and referee comment. If no action was taken to address a point, you must provide a compelling argument. When formatting this document, please respond to each reviewer comment individually, including the full text of the reviewer comment verbatim followed by your response to the individual point. This response will be used by the editors to evaluate your revision and sent back to the reviewers along with the revised manuscript.

- Highlight all changes made to your manuscript or provide us with a version that tracks changes.

- **EXTENDED DATA FIGURES**

Link Redacted

We look forward to seeing the revised manuscript and thank you for the opportunity to review your work. Please do not hesitate to contact me if you have any questions or would like to discuss these revisions further.

Sincerely,

██████████

████████████████████
████████████████████

Nature Human Behaviour

REVIEWER COMMENTS:

Reviewer #1 (Remarks to the Author):

This study tested how integrating a biologically inspired model of attentional gain into a DNN explains human perceptual behavior in speech processing tasks. The model replicated several previously reported behavioral speech-in-noise findings and predicted two new ones, which the authors then validated in new human studies.

This is a creative integration of ideas from neurophysiology into advanced computational models of auditory processing. The study is quite thorough, testing performance of the model over a wide range of naturalistic conditions. It makes a fairly convincing argument that gain-based attention models can explain several natural speech processing behaviors. While this work is important and likely to have impact, it is difficult to get a sense of how much the gain-based model actually contributes to the human-like performance that is reported. It would be helpful if the authors could provide a clearer quantification of this point.

Major concern.

The several results in Figs. 2-4 illustrate striking parallels between human and machine performance, but these aren't entirely shocking, given that both humans and DNNs are generally considered able to perform close to the limit of information available in noisy speech conditions. The critical analysis for this study is in Fig. 6, which quantifies the relative performance of different attention models. What's important is not absolute performance but rather that the gain model makes similar errors to humans. A few clarifications would help:

1. How does the absolute performance of the models compare? (In particular, in the comparison of the feature-gain and baseline CNNs.) Does it matter?
2. Are there any training effects? E.g., if the training set size is cut in half, would the relative performance of the different attention models change? One can easily imagine that the different complexity / architecture of the models could impact their relative performance for a fixed training set size. It is asking a lot to re-run everything, but it would help if the authors could provide a convincing case that the results are not an artifact of differing model complexity.
3. A final nitpick: In Fig. 5c, the cloud of points for the baseline CNN clearly falls further from the dashed line than for the feature-gain model. However, the points appear more tightly clustered. A metric like mutual information might show tighter correspondence between this model and human performance than the feature-gain model, which produces more of a cloud, at least in the high-performance domain. Can the authors show that an MI analysis demonstrates a better match for the feature-gain model? Or argue that their MSE/correlation metrics are more appropriate?

Lesser concerns.

L83. Typo: "it's" should be "its".

L100. It would be helpful to mention that you apply an HRTF to the inputs here.

L123. "feature gains happened immediately prior to each convolutional 'block'". Is this detail of the model architecture important to the results? It's quite reasonable, given the neurophysiology literature, but it would be compelling if the configuration of operations impacts performance. Again, it may be a lot to re-train many models, but any details on its impact would be helpful.

Fig. 1b. Confusing: displaying two cartoon model implies that there were two CNNs. After puzzling through the text, it became clear that the cue and mixture are actually processed by the same CNN. It might help to somehow tweak the figure or otherwise emphasize that both stimuli were presented to a single model.

Fig. 2. (And later figures). It's hard to distinguish the dots that define the data points from the dashed lines. Can the dots be bigger?

L202. It would be quite compelling if the lapses occurred for the same words/conditions. Is there any evidence that the model matched human performance on a stimulus-by-stimulus level?

L519. "... relatively late selection" Can the authors say anything about the level of representation in the model at which the separation emerges? Does model activity in this layer correlate with pitch? Phoneme identity? Word identity? This is interesting because it could establish more concrete parallels (or hypotheses) about the level of representation at which attentional effects emerge.

L532. "... predictions of effects..." Can the relative performance in these new conditions be attributed to acoustics (e.g., is separation in the HRTF-transformed stimulus less for elevation differences than azimuth difference)? Or do they suggest some differences in neural processing?

L554. The authors should comment on how biologically plausible their gain model is. Where would the gain signal be stored? Tonic feedback? Short-term synaptic weight changes?

L764. Are separate theta_1-3 values learned for each input channel in each layer? Or a single set of three values for each layer?

Reviewer #1 (Remarks on code availability):

I didn't have resources available to run the code, but I was able to download and inspect it. The documentation is reasonable.

Reviewer #2 (Remarks to the Author):

This article reports on a novel neural net model that incorporates feature-based weighting tasked with identifying a target voice in the presence of a distractor or multiple distractors. The model was tested in simulated experiments, and the results were compared to behavioral data from 7 different experiments with human participants (in some cases drawn from previously published data), each probing a specific aspect of selective attention and speech recognition performance in cocktail-party like environments. Prominent features probed were speech level, pitch, and spatial location. Model performance was shown to be remarkably good despite no optimization targeted at the speech cues themselves. It's hard to comment on any weaknesses in the manuscript, as the sheer number of validation experiments, thorough presentation of methods and results, and impact on our understanding of how auditory selective attention could be operating biologically. The only quibble I have is related to Experiment 4 on the precedence effect. It's unclear to me where in the model the illusory spatial "percept" of the model would come from. I may have missed something in the methods, but does the auditory front end (binaural cochleagram) account for this possibility or does the model figure out something qualitatively different? The comment that "the model presumably learns representations of sound location that are adapted to presence of reflections" is not easily understood in the context of the stage this would be possible. It also opens the door for what other illusions it would be capable of learning, and I think this could be expanded on in the discussion. Aside from that, I want to commend the authors on the manuscript and the contribution to the field.

Reviewer #2 (Remarks on code availability):

The python code is well documented and well organized for sufficient reproduction of the paper's methodology.

Reviewer #3 (Remarks to the Author):

This manuscript by Griffith et al presents an ambitious and technically impressive attempt to instantiate a long-standing neurophysiological motif of attention, namely multiplicative feature gains, in a task-optimized model of selective listening. At first glance, it is striking that the resulting models replicate many psychophysical hallmarks of human auditory attention, including both its successes and its characteristic failures. Using this model, the authors also generate testable predictions subsequently borne out in human data.

ql found the work very strong in scope and execution. Still, two limitations should temper the claims. First, the reliance on a highly artificial cueing regime makes it unclear how the framework would extend to sustained or less constrained attention. Second, the interpretation of human-like errors as evidence of normative optimality seems somewhat circular, as the model is built to exploit precisely those cues. Finally, the link to neurobiology remains abstract: while the emergence of "late selection" is consistent with cortical data, the feedforward CNN with sigmoidal gains is still far removed from neural circuitry. Overall, however, this is a very valuable contribution. It was a pleasure to review this. The study provides a rare, computationally grounded account of auditory selective attention that connects neural motifs, behavioral performance, and normative explanation. I am supportive of publication after some tempering of the broader claims about optimality and biological plausibility. See my detailed comments below.

###MAJOR

– Cueing: To my understanding, the entire framework hinges on an explicit two-second cue of the target voice/location. This is far removed from everyday listening. Importantly, it raises questions about ecological validity and generalizability with a paper at this level should certainly not brush over. Some discussion of how the model could be extended to sustained or dynamic attention (without such clear cues) are needed.

Closely related, all behavioral comparisons are with relatively (not to say, highly) controlled lab tasks. How the framework would fare in more ecological listening settings (continuous speech, multiple moving sources, no explicit cue) has remained less clear to me, and – in all fairness – has not been this paper's focus. However, I feel this limitation needs to be fairly discussed.

– I wondered about the biological plausibility, or rather the degree to which these results of are themselves evidence for a human-like process quite a bit. The claim that the model "explains" human late selection should be tempered, in my mind. The emergence of late selection in the model could simply reflect the architecture's representational bottlenecks, i.e. where discriminative features happen to reside after training. This is not necessarily indicative of a mechanistic parallel to human auditory cortex and attention systems. Moreover, the feedforward CNN with cue-dependent multiplicative gains is still very far from known auditory circuits: it lacks recurrent connectivity, neuromodulatory influences, or any dynamics of attentional control. The result is still interesting — it shows that a simple gain mechanism suffices to generate late-selection-like behavior — but I would be cautious to frame this as evidence for how late selection is implemented in the brain.

– Also, I think the conclusion that human-like errors are "inevitable" and thus normative is not entirely warranted. Alternative explanations (suboptimal heuristics, lapses, executive control failures) should be considered more explicitly.

– Framing: Much of the modeling cleverly unifies known elements (feature gains, normalization, task optimization) rather than introducing a fundamentally new attentional mechanism. The manuscript could better acknowledge this and position the contribution as integration + prediction rather than conceptual breakthrough.

###MINOR/MISC:

-- When comparing to visual attention models, I feel the Discussion in places overstates the uniqueness of this approach; relevant work in vision is acknowledged, but a more balanced framing would help.

-- Executive control / effort: The deterministic, static gains leave no room for attentional effort or dynamic adaptation. This limitation could be discussed in relation to human attention variability.

-- Figure clarity: Some multi-panel figures are extremely dense (esp. Fig. 2, 3, 4). A schematic "map" of experimental

conditions and which ones were validated in humans vs. only simulated would aid readers.

-- The "early-only" and "late-only" controls are informative, but further analysis (e.g., why the early-only still partially works) would strengthen the conclusions.

-- Somewhat related, the phrase "late selection" is used normatively. However, readers outside the auditory attention field may expect a clearer link back to the classic early/late selection debate.

###TYPO-LEVEL

-- "contrained" instead of "constrained" (p. 2).

-- "explicitely" should be "explicitly".

Reviewer #3 (Remarks on code availability):

I have not fully testdriven the code, but have perused the provided materials. Providing results files and example stimuli is commendable, as is the python code.

Version 2:

Decision Letter:

Our ref: NATHUMBEHAV-25073107B

7th November 2025

Dear Dr. McDermott,

Thank you for submitting your revised manuscript "Optimized feature gains explain and predict successes and failures of human selective listening" (NATHUMBEHAV-25073107B). It has now been seen by the original referees and their comments are below. As you can see, the reviewers find that the paper has improved in revision. We will therefore be happy in principle to publish it in Nature Human Behaviour, pending minor revisions to comply with our editorial and formatting guidelines.

Sincerely,

[REDACTED]

[REDACTED]

Nature Human Behaviour

Reviewer #1 (Remarks to the Author):

The authors have addressed all the reviewers' concerns quite thoroughly. The manuscript is now quite strong, as well as interesting, and I have no further critiques.

Reviewer #1 (Remarks on code availability):

Same as before. Looks organized, too big to evaluate.

Reviewer #3 (Remarks to the Author):

The authors should be congratulated for such a rich and deep set of data and results, which—in my view—they have managed to improve upon considerably through their attentive and very constructive revision.

I have no further concerns about this going into print.

Reviewer #3 (Remarks on code availability):

(Not this time round)

Griffith, Hess & McDermott - Response to Reviews

Please note that all line numbers quoted below refer to the pdf document that has tracked changes.

Reviewer #1 (Remarks to the Author):

This study tested how integrating a biologically inspired model of attentional gain into a DNN explains human perceptual behavior in speech processing tasks. The model replicated several previously reported behavioral speech-in-noise findings and predicted two new ones, which the authors then validated in new human studies.

This is a creative integration of ideas from neurophysiology into advanced computational models of auditory processing. The study is quite thorough, testing performance of the model over a wide range of naturalistic conditions. It makes a fairly convincing argument that gain-based attention models can explain several natural speech processing behaviors.

Thank you.

While this work is important and likely to have impact, it is difficult to get a sense of how much the gain-based model actually contributes to the human-like performance that is reported. It would be helpful if the authors could provide a clearer quantification of this point.

Major concern.

The several results in Figs. 2-4 illustrate striking parallels between human and machine performance, but these aren't entirely shocking, given that both humans and DNNs are generally considered able to perform close to the limit of information available in noisy speech conditions.

We note that although there are prior demonstrations of similar human and DNN performance in noisy speech conditions, it has not been previously possible to compare human and model performance when there are multiple concurrent talkers (because this requires a way for the model to select one of several talkers for a response). So while there is some precedent for the speech-in-noise results of Fig. 2g&h (indeed, these conditions were included to show that the model could also reproduce the human-model similarity exhibited by previous models), the speech-on-speech results of Fig. 2a-f are quite new. As a result, it was not at all obvious a priori whether the models would succeed at human levels in a setting requiring selection. We have tried to better point this out in the revised manuscript:

“Previous models of word recognition have replicated patterns of intelligibility across different types of noise. However, because they lacked the ability to use a

cue to selectively report one of several talkers, they were unable to perform at human levels in the presence of a single distractor talker, because without a cue, the task is ill-defined. As a result, it was not clear a priori that a model would be able to match human behavior in this regime.” (lines 180-184)

The critical analysis for this study is in Fig. 6, which quantifies the relative performance of different attention models. What’s important is not absolute performance but rather that the gain model makes similar errors to humans.

Similar errors are important, but it is also important for a model to match absolute performance - we seek models and theories that fully explain what people do, and that includes how well they do it. For this reason we used metrics sensitive to both the similarity of errors and absolute performance. As discussed and shown in more detail below, the feature-gain model best accounts for absolute as well as relative performance across conditions. We have revised the text to address this issue and to explain and justify the metrics used for human-model comparison:

“These two metrics are complementary: RMS error metric captures differences in both absolute and relative performance, whereas the correlation metric isolates differences in relative performance. A good model should match both absolute performance and relative performance differences between conditions, and should match humans as well or better than other models on all metrics.” (lines 508-512)

A few clarifications would help:

1. How does the absolute performance of the models compare? (In particular, in the comparison of the feature-gain and baseline CNNs.) Does it matter?

We have added scatter plots comparing the control models to the feature-gain model. These show that the control models produce worse recognition of the target talker, and more confusions (erroneous reports of the distractor talker).

This figure has been added as Supplementary Figure 14 and is referenced in the text:

“See Supplementary Fig. 14 for explicit comparison of model performance.” (line 548)

2. Are there any training effects? E.g., if the training set size is cut in half, would the relative performance of the different attention models change? One can easily imagine that the different complexity / architecture of the models could impact their relative performance for a fixed training set size. It is asking a lot to re-run everything, but it would help if the authors could provide a convincing case that the results are not an artifact of differing model complexity.

We note that the four model architectures are very similar in the total number of parameters. This is because there is only one sigmoid function per model stage (each with three parameters). The feature-gain model has only 21 more parameters than the early-only and late-only models (out of a total of 62.6 million). The baseline model actually has the most parameters, because there are four channels in the first convolutional stage. As a result, the differences in performance and human-similarity are very unlikely to be explained by the amount of training data.

We nonetheless attempted to train the feature-gain model and baseline model (as the baseline model differs most in number of parameters from the feature-gain model) on half as much training data. Cutting the training set in half turned out to make training much slower. We trained both models for about a week and a half, and both were improving, but it became clear that it would have taken a fair bit longer to reach convergence. Given that the difference in model complexity is in actuality quite modest, we concluded that the issue was not worth the substantial GPU time that would have been needed to train things to completion. Instead, we have clarified that the complexity of the four models is in fact quite similar:

“Because each model stage had a single shared sigmoid gain function defined by only three parameters, the early-only, late-only, and feature-gain models had nearly the same number of parameters (differing by 21 parameters out of 62.6 million). The baseline model had slightly more parameters (just over 69 million) due to the extra input channels, but the models were overall similar in complexity.” (lines 498-502)

We have also added more discussion of the importance of the architectural constraint of the gains, and how to think about this:

“We emphasize that the results do not preclude the (likely) possibility that a sufficiently large baseline model, trained on enough data, could reproduce the performance of the feature-gain model (potentially by learning to replicate the effect of the gains). We view the feature gains as analogous to other architectural

motifs (e.g. convolution) which in principle could also be learned by a less structured model. We also emphasize that the alternative models were fully optimized with the alternative architecture, and were free to learn an entirely different feature hierarchy that allowed them to maximize their performance given the constraint of gains in particular locations (or no explicit gains at all). It is thus perhaps not surprising that these models were able to perform well above chance. The success of the architectural constraint in replicating human behavior nonetheless provides support for its role in biological attention.” (lines 551-560)

3. A final nitpick: In Fig. 5c, the cloud of points for the baseline CNN clearly falls further from the dashed line than for the feature-gain model. However, the points appear more tightly clustered. A metric like mutual information might show tighter correspondence between this model and human performance than the feature-gain model, which produces more of a cloud, at least in the high-performance domain. Can the authors show that an MI analysis demonstrates a better match for the feature-gain model? Or argue that their MSE/correlation metrics are more appropriate?

We have added an analysis using mutual information, which shows similar trends to the other metrics we used. This analysis has been added as a supplementary figure. As described above, we have also added more motivation and discussion of the metrics we used, which each capture different aspects of absolute and relative similarity. A good model should match humans well according to all metrics, and it is reassuring that the feature-gain model better fits human results according to all the metrics we used.

This new analysis is mentioned in the main text:

“We also used a mutual information metric (Supplementary Fig. 12), which yielded similar conclusions.” (lines 512-513)

And is described in the methods:

“To calculate mutual information, we generated histograms of human and model performance. To ensure robustness to histogram bin size, mutual information was computed using bin counts of 5, 10, 15, and 20 bins. Error bars on the mutual information for each bin size were obtained via bootstrap (1000 samples).” (lines 1725-1728)

Lesser concerns.

L83. Typo: “it’s” should be “its”.

Corrected.

L100. It would be helpful to mention that you apply an HRTF to the inputs here.

Done. The revised text reads:

“During training, audio was spatially rendered at locations within simulated reverberant rooms using head-related transfer functions to reproduce the spatial cues available to humans.” (lines 95-97)

L123. “feature gains happened immediately prior to each convolutional ‘block’”. Is this detail of the model architecture important to the results? It’s quite reasonable, given the neurophysiology literature, but it would be compelling if the configuration of operations impacts performance. Again, it may be a lot to re-train many models, but any details on its impact would be helpful.

To address this issue, we trained an additional model in which the gains were applied after the normalization operation rather than before (i.e., between the normalization and convolution operations). This alternative ordering produced worse performance, and a worse match to human behavior. We have added this alternative model to Fig. 6, and describe the model and results in the text:

“To test the importance of the application of gains followed by normalization, we also trained a feature-gain model in which this ordering was reversed, with gains applied after normalization (“gain-after-normalization”).” (lines 493-495)

“Each alternate architecture showed significantly lower human-model similarity (root-mean-squared error: feature-gain vs baseline CNN $p=0.002$, difference of means = -0.070 ; feature-gain vs gain-after-normalization $p=0.002$, difference of means = -0.078 ; feature-gain model vs early-only $p=0.002$, difference of means = -0.040 ; feature-gain model vs late-only $p=0.002$, difference of means = -0.125 ; Pearson’s r^2 : feature-gain vs baseline CNN $p=0.002$, difference of means = 0.093 ; feature-gain vs gain-after-normalization $p=0.002$, difference of means = 0.007 ;

feature-gain model vs early-only $p=0.002$, difference of means = 0.036; feature-gain model vs late-only $p=0.002$, difference of means = 0.507.” (lines 538-5445)

Here is the new Figure 6 that includes the results for the new model:

Fig. 1b. Confusing: displaying two cartoon model implies that there were two CNNs. After puzzling through the text, it became clear that the cue and mixture are actually processed by the same CNN. It might help to somehow tweak the figure or otherwise emphasize that both stimuli were presented to a single model.

We have re-arranged the figure and made it more explicit that the same CNN processes the cue and mixture:

Fig. 2. (And later figures). It's hard to distinguish the dots that define the data points from the dashed lines. Can the dots be bigger?

We have made the dots bigger in the revised figures. Here is an example from Figure 2:

L202. It would be quite compelling if the lapses occurred for the same words/conditions. Is there any evidence that the model matched human performance on a stimulus-by-stimulus level?

We ran an analysis to measure the correlation between human and model performance for individual stimuli. For each tested SNR, we computed the correlation between human and model performance on individual 2-talker stimuli (i.e., with a target talker and a single distractor talker). We then averaged this correlation across the five tested SNRs. We also estimated a noise ceiling for how high this correlation might be expected to be by measuring the correlation between different groups of human participants. The human-model correlation was calculated separately for each of the two splits of human participants used for the human-human correlation and then averaged. Error bars on each correlation were obtained by bootstrapping over stimuli. We also calculated a null distribution for the correlation via the human-model correlation with permuted trials.

The human-human noise ceiling here is relatively low because the experiments were not designed to produce reliable results for individual stimuli, with each stimulus being presented only a few times on average across all human participants. But the human-human correlation is nonetheless greater than chance, and the human-model correlation is on par with the human-human correlation (note the large error bars). This analysis suggests that the model indeed captures some of the stimulus-specific variation in performance. We note that the differences between models are not statistically significant in this analysis, and that the experiment was not designed to enable this analysis to be especially well powered.

Trial-level correlation in performance between splits of human participants, and between those human participants and the model. Error bars plot 1 SEM.

Given that the experiments were not designed to test trial-level similarity, our preference is to leave this analysis out of the paper. The issue merits a dedicated experiment, but is not critical to the paper’s conclusions, and so seems best left for a future project.

L519. "... relatively late selection" Can the authors say anything about the level of representation in the model at which the separation emerges? Does model activity in this layer correlate with pitch? Phoneme identity? Word identity? This is interesting because it could establish more concrete parallels (or hypotheses) about the level of representation at which attentional effects emerge.

We addressed this issue with two new analyses. In the first, we extended the stage of selection analysis that was in the original manuscript. The original analysis had been restricted to co-located target and distractor signals. We added the analogous analyses for target and distractor signals with small (10 degree) and large (90 degree spatial separation), in order to show how the stage of selection would be affected when spatial cues are also used for selection. These show more pronounced differences between the target-mixture and distractor-mixture correlations, as expected given that task performance was better for these configurations. However, the stage at which the target- and distractor-mixture correlations diverge remained largely consistent. There is some modest separation evident at early stages for the largest spatial separation, but this is also present in the model with random weights, and is plausibly explained by the SNR difference that is induced in each ear in this setting.

Overall, this extended analysis substantiates the results of the original analysis.

To probe how the stage of selection might relate to the tuning to different types of features, we ran a second analysis. In this analysis we measured model activations for a set of speech excerpts rendered at different spatial locations, and analyzed the proportion of units at each model stage that were selective for location, pitch, talker, and the middle word of the excerpt. Specifically, we used a linear model to measure the variance of each unit's response that was explained by each of these features.

This analysis revealed that tuning to location and talker – the two dimensions used for selection in the task – was prevalent in early model stages, with tuning for words being weak in early stages and stronger later. Tuning to pitch (specifically, the mean f_0 of the voiced segments of the speech excerpt), which is one of many features defining the talker's voice, was evident throughout the model. Location tuning was less prevalent in deep model stages, whereas talker tuning persisted, potentially because talker cues are extracted over a larger number of hierarchical stages.

Overall, this tuning is consistent with the stage-of-selection analysis, in that tuning to the dimensions used for selection is present prior to the stages where selection is evident, such that feature gains could be used there to help select the target.

This analysis has been added as Supplementary Fig. 11, and is described in the main text:

“To examine how selection relates to feature tuning, we measured model activations for a set of speech excerpts rendered at different spatial locations, and analyzed the proportion of units at each model stage that were selective for location, pitch, talker, and the middle word of the excerpt (Supplementary Fig. 11). This analysis revealed that tuning to location and talker – the two dimensions used for selection in the task – was prevalent in early model stages, with tuning for words being weak in early stages and stronger later. Overall, this tuning is consistent with the results of Fig. 5, in that tuning to the dimensions used for selection is present prior to the stages where selection is evident, such that feature gains could be used there to help select the target.” (lines 479-486)

Supplementary Fig. 11 | Feature tuning across model stages. We measured model activations for a set of speech excerpts rendered at different spatial locations, and analyzed the variance of each unit’s response that was explained by location, pitch, talker, and the middle word of the excerpt. The graph plots the mean of this quantity at each model stage. The two dimensions that are useful for selection (location and talker) account for considerable response variance in early model stages. Tuning to words is weak in early stages and increases towards the end of the model. Tuning to pitch (specifically, the mean f0 of the voiced segments of the speech excerpt), which is one of many features defining the talker’s voice, is evident throughout the model. Location tuning was less prevalent in deep model stages, whereas talker tuning persisted, potentially because talker cues are extracted over a larger number of hierarchical stages.

L532. “... predictions of effects...” Can the relative performance in these new conditions be attributed to acoustics (e.g., is separation in the HRTF-transformed stimulus less for elevation differences than azimuth difference)? Or do they suggest some differences in neural processing?

We believe that both effects, in different ways, reflect the acoustic spatial cues available for attentional mechanisms to use for selection. We have added an analysis to clarify this issue.

We quantified acoustic spatial cues by calculating the difference between the binaural cochlear representations of the same speech excerpt presented at different spatial positions. Specifically, we generated representational dissimilarity matrices for a) pairs of single target speech excerpts rendered at different positions, as a function of their spatial separation in either elevation or azimuth, and b) pairs of single target speech excerpts and the mixture of that target speech excerpt with a distractor excerpt. In the second case the distractor was always presented at the midline. The rationale was to test 1) whether the presence of the distractor disrupted spatial cues of the target and 2) whether there was evidence that spatial cues were less precise in the periphery compared to the midline.

We note that this analysis is merely intended to complement the model/human behavioral results by providing some intuition for the acoustic factors that constrain performance. We emphasize that the analysis does not reveal anything about whether the location-related differences in the cochlear representation could actually be used for attentional selection (for this you need a model of behavior, which is the point of the paper). But it nonetheless is useful in providing some intuitive understanding.

The RDM analysis shows, as expected, that there are spatial cues to both elevation and azimuth for single speech excerpts - in both cases there is a gradient of dissimilarity moving away from the diagonal. However, when a distractor talker is present, only the azimuth cues remain pronounced. The

gradient away from the diagonal is much weaker for elevation when a second talker is present. There is some residual variation with elevation, and given this analysis alone it would not be clear whether this variation might be useful for attentional selection. But in conjunction with the model and human behavioral results, this analysis suggests that the presence of a second talker has a fairly disruptive effect on elevation cues, and that is why they are not used for attentional selection (this interpretation is what we hypothesized in the original submission).

RDMs for Cochleagram Representations by Elevation

Single talker v Single talker
 $1 - r$ RDM

Single talker v Two talker
 $1 - r$ RDM

RDMs for Cochleagram Representations by Azimuth

Single talker v Single talker
 $1 - r$ RDM

Single talker v Two talker
 $1 - r$ RDM

The RDM for azimuth also shows a bowtie structure like that seen in the model behavior, confirming that the wider spotlight in the periphery is due to the non-uniform variation in spatial cues with position, as we had suggested in the original submission. This can be clearly seen in line plots of the dissimilarity vs. talker excerpt separation when the first excerpt is at 0 (in front) vs. at 90 degrees (all the way out to the side):

We have added these results as a supplementary figure (Supplementary Fig. 7), and have referenced the analysis in the text as helping to explain the results:

“It seemed plausible that monaural cues might be less robust to the presence of concurrent sources (see Supplementary Fig. 7).” (lines 385-387)

“This effect is plausibly due to the higher acuity of localization at the midline⁴⁶⁻⁴⁸, which in turn is thought to relate to the derivative of binaural cues with spatial position (see Supplementary Fig. 7).” (lines 405-407)

L554. The authors should comment on how biologically plausible their gain model is. Where would the gain signal be stored? Tonic feedback? Short-term synaptic weight changes?

We have added some speculation about this supported by references to relevant literature. We presume that the target of attention is stored in working memory, potentially in pre-frontal cortex (as is commonly proposed for visual attention), with gains implemented via some mix of excitation and inhibition (there are many such proposals in the computational neuroscience literature).

“The model results leave open the neurobiological implementation of the cue memory and feature gains, but are compatible with previous proposals for sources of feature-based attention in prefrontal cortex, and of gain changes implemented via combinations of excitation and inhibition.” (lines 630-633)

L764. Are separate theta_1-3 values learned for each input channel in each layer? Or a single set of three values for each layer?

A single set of three values is learned for each layer. This was discussed in the original submission, but we have made this more prominent in the revised manuscript:

“We used a single set of shared sigmoid parameters per model stage (we found this was sufficient to yield good performance).” (lines 124-125)

Reviewer #1 (Remarks on code availability):

I didn't have resources available to run the code, but I was able to download and inspect it. The documentation is reasonable.

Reviewer #2 (Remarks to the Author):

This article reports on a novel neural net model that incorporates feature-based weighting tasked with identifying a target voice in the presence of a distractor or multiple distractors. The model was tested in simulated experiments, and the results were compared to behavioral data from 7 different experiments with human participants (in some cases drawn from previously published data), each probing a specific aspect of selective attention and speech recognition performance in cocktail-party like environments. Prominent features probed were speech level, pitch, and spatial location. Model performance was shown to be remarkably good despite no optimization targeted at the speech cues themselves. It's hard to comment on any weaknesses in the manuscript, as the sheer number of validation experiments, thorough presentation of methods and results, and impact on our understanding of how auditory selective attention could be operating biologically.

Thank you.

The only quibble I have is related to Experiment 4 on the precedence effect. It's unclear to me where in the model the illusory spatial "percept" of the model would come from. I may have missed something in the methods, but does the auditory front end (binaural cochleagram) account for this possibility or does the model figure out something qualitatively different? The comment that "the model presumably learns representations of sound location that are adapted to presence of reflections" is not easily understood in the context of the stage this would be possible. It also opens the door for what other illusions it would be capable of learning, and I think this could be expanded on in the discussion. Aside from that, I want to commend the authors on the manuscript and the contribution to the field.

This effect is not due to the cochleagram. The model presumably learns to suppress echos in order to more robustly represent the true locations of sound sources. The effect has previously been seen in models of sound localization, so it is cool, but not shocking, to see it emerge in this setting. The results provide more evidence that localization has to be robust to reflections in order to be useful in real-world tasks, such that the model here learns to deal with reflections even when it doesn't have to explicitly localize sounds.

To clarify this issue we have added more extensive background explanation of the precedence effect to the paper:

In the caption to Figure 3:

"This effect is commonly thought to reflect a localization strategy adapted to cope with environmental reflections. Reflections typically arrive from directions distinct from the true direction of the source from a listener, but are delayed due to the increased path length compared to the sound that arrives directly from the source. The human auditory system appears to suppress localization cues from delayed

sound components that are likely to be due to reflections, presumably to achieve more robust localization in real-world conditions.” (lines 299-304)

And in the methods description to the experiment:

“The effect has been hypothesized to reflect a strategy for localizing accurately in the presence of reflections. Because reflections typically arrive from a direction that is different from the direction of the source from the listener, they provide erroneous localization cues. However, they are also delayed relative to the direct sound from the source (which has cues that are faithful to the source location), because the path length traversed by the reflection is longer than that for the direct sound. Suppressing location cues from delayed components of a sound could thus make localization more robust to reflections. Consistent with this explanation, models that are optimized for localization in environments with reflections exhibit the precedence effect, whereas those optimized in anechoic environments do not. Models plausibly learn to accentuate localization cues from early parts of a sound and suppress cues that are likely to originate from reflections, akin to effects seen in the early auditory system.” (lines 1471-1481)

We have also added a speculative explanation after the description of the results, to help the reader see how a model could learn to mediate the effect:

“As in models optimized for sound localization in realistic environments, the model presumably learns representations of sound location that are adapted to the presence of reflections (potentially by suppressing the localization cues of late-arriving sound akin to the biological auditory system), allowing it to benefit from the illusory spatial separation in this experimental setting.” (lines 337-341)

Reviewer #2 (Remarks on code availability):

The python code is well documented and well organized for sufficient reproduction of the paper's methodology.

Reviewer #3 (Remarks to the Author):

This manuscript by Griffith et al presents an ambitious and technically impressive attempt to instantiate a long-standing neurophysiological motif of attention, namely multiplicative feature gains, in a task-optimized model of selective listening. At first glance, it is striking that the resulting models replicate many psychophysical hallmarks of human auditory attention, including both its successes and its characteristic failures. Using this model, the authors also generate testable predictions subsequently borne out in human data.

I found the work very strong in scope and execution.

Thank you.

Still, two limitations should temper the claims. First, the reliance on a highly artificial cueing regime makes it unclear how the framework would extend to sustained or less constrained attention. Second, the interpretation of human-like errors as evidence of normative optimality seems somewhat circular, as the model is built to exploit precisely those cues. Finally, the link to neurobiology remains abstract: while the emergence of “late selection” is consistent with cortical data, the feedforward CNN with sigmoidal gains is still far removed from neural circuitry. Overall, however, this is a very valuable contribution. It was a pleasure to review this. The study provides a rare, computationally grounded account of auditory selective attention that connects neural motifs, behavioral performance, and normative explanation. I am supportive of publication after some tempering of the broader claims about optimality and biological plausibility. See my detailed comments below.

Thanks. We basically agree with your take on the limitations, and have tried to better acknowledge and discuss these in the revised text.

###MAJOR

– Cueing: To my understanding, the entire framework hinges on an explicit two-second cue of the target voice/location. This is far removed from everyday listening. Importantly, it raises questions about ecological validity and generalizability with a paper at this level should certainly not brush over. Some discussion of how the model could be extended to sustained or dynamic attention (without such clear cues) are needed.

Closely related, all behavioral comparisons are with relatively (not to say, highly) controlled lab tasks. How the framework would fare in more ecological listening settings (continuous speech, multiple moving sources, no explicit cue) has remained less clear to me, and – in all fairness – has not been this paper’s focus. However, I feel this limitation needs to be fairly discussed.

We take the cue-mixture paradigm to be a convenient way to both measure attentional selection behavior at scale, and to optimize a model for such behavior. We envision the paradigm as an example of one typical way in which attention is directed: the observer has a memory representation of what they want to select, and this memory shapes their subsequent attention. Sometimes the memory is not derived from a recent stimulus, but rather from a long-term memory, e.g. of a

familiar voice. It could also be derived from another sense, as if you see something at a location and then listen for something at that location. In each of these variants we envision that the same basic attentional mechanism is likely to be at work.

Of course, these settings and others may also involve something somewhat more sophisticated. For instance, sometimes the memory that guides attention is relatively abstract (e.g. you may know to listen to a woman's voice without knowing the specific timbre). In these cases it seems likely that the attentional "filter" is refined over time, with some initial gain settings that enable you to get started, but with the gains being updated over time based on the specific features of the attended source. Even when attention is based on a recent stimulus, if the stimulus is extended in time, attentional gains seem likely to be refined over time as the observer gains additional evidence about the attended source's features. Such attentional refinement likely also enables attentive tracking of stimuli whose features are not stationary (e.g. the moving source that you mention). We view these phenomena and the associated computational requirements as one of the most natural next steps for this research program.

We tried to describe some of these extensions in the original paper, but have revised their description to more clearly note the limitations of the cue-mixture paradigm, noting that studying these situations will require some additional task settings, and a more complicated model architecture.

At the start of the Results section:

"The task captured key aspects of attentional selection in the wild by requiring selection based on the memory of a sound source, but its stereotyped form enabled training and behavioral benchmarking at a large scale." (lines 85-87)

In the Discussion, we removed the claim of "testing the effect of attentional mechanisms in a naturalistic setting and task", as it could be misinterpreted.

We also revised the first paragraph of the Limitations section of the Discussion in light of these issues:

"Our modeling framework used a stereotyped task setting in which there is a cue stimulus from which attentional gains can be derived, followed by a stimulus to which the attentional gains are applied. This setting facilitated large-scale training and testing of attention, but does not fully capture the variety of ways in which real-world attention arises. Some of the additional complexity found in real-world attention likely reflects flexible executive control of attention. For instance, humans can direct auditory attention to a verbally instructed location, or to a familiar voice that comes to mind even in the absence of a prior cue stimulus. We envision that the same basic attentional gain mechanisms could work in these

settings, but with the gains derived from an internal representation rather than the cue stimulus. Humans can also adjust the strength of attention using executive control, perhaps based on the perceived difficulty of a task, and potentially relating to the feeling of effort. Extending the modeling framework to allow the strength of attentional gains to vary could help to understand effort in computational terms.” (lines 667-678)

In addition, to show that the present model has some degree of generality, we ran a new experiment in which we varied the duration of the cue, simulating settings in which one gets only a “glimpse” of a source before needing to direct attention to it. This experiment shows that both humans and the feature-gain model are robust to having substantially shorter cues (500 ms instead of the 2s duration in the main experiments):

We have added this result as an additional supplementary figure, and now describe it in the text. In the Methods section:

“Experiment 1b: effect of cue duration (Supplementary figure 3)

It seemed possible that the fixed duration of the cue signal in our main task could limit the ecological validity of our results. To assess whether human and model behavior were robust to variation in the cue duration, we ran an additional experiment using the materials and procedure of Experiment 1, but with cues that varied in duration. The same set of cue, target, and single-distractor excerpts used in Experiment 1 were used, but cue signals were center cropped to be either 0.5-,

1-, or 2-seconds in duration. All target-distractor mixtures were presented at 0dB SNR.

Procedure

The procedure was identical to that of Experiment 1, with the following exceptions. Participants each completed 12 catch trials and 30 experimental trials (10 trials x 3 cue duration conditions). Each participant heard a random sample of 30 of the 488 target words, with the cue duration for each example randomly assigned for each participant. The 12 catch trials were randomly intermixed with the 30 experimental trials. 84 participants (43 female, 38 male, 3 non-binary) met our inclusion criteria (scoring 91% on the catch trials). Participant ages were between 20 and 40 (median 32) years.

Model experiment

Models were tested on all combinations of the 976 target clips (both male and female examples of each of the 488 target words) with the 3 cue duration conditions, and both distractor sex conditions (5,856 total stimuli) used in the human experiments. As in Experiment 1, the mono audio clips were not run through a room simulator, and were instead presented to the model diotically.” (lines 1309-1331)

And in the Results section:

“Human and model performance was robust to variation in cue duration (Supplementary Figure 3), demonstrating some generality in the model’s behavior despite the stereotyped structure of the training examples.” (lines 187-190)

– I wondered about the biological plausibility, or rather the degree to which these results of are themselves evidence for a human-like process quite a bit. The claim that the model “explains” human late selection should be tempered, in my mind. The emergence of late selection in the model could simply reflect the architecture’s representational bottlenecks, i.e. where discriminative features happen to reside after training. This is not necessarily indicative of a mechanistic parallel to human auditory cortex and attention systems. Moreover, the feedforward CNN with cue-dependent multiplicative gains is still very far from known auditory circuits: it lacks recurrent connectivity, neuromodulatory influences, or any dynamics of attentional control. The result is still interesting — it shows that a simple gain mechanism suffices to generate late-selection-like behavior — but I would be cautious to frame this as evidence for how late selection is implemented in the brain.

We agree with your perspective on this, and have revised the text accordingly in a number of places, referring to a “normative perspective” rather than a “normative explanation”, and removing strong claims that the result definitively explains what is observed in humans:

“Lastly, inspection of the model representations showed evidence of late selection, providing a normative perspective on effects seen neurophysiologically.” (lines 591-592)

“Our results show how relatively late selection can emerge as an optimized solution for attentional selection of speech. This optimized solution could also depend on the system architecture or on other biological constraints we did not model. Our result thus does not provide a definitive explanation of the locus of selection found in the brain, but it illustrates a computational approach to understanding the issue.” (lines 646-651)

– Also, I think the conclusion that human-like errors are “inevitable” and thus normative is not entirely warranted. Alternative explanations (suboptimal heuristics, lapses, executive control failures) should be considered more explicitly.

These alternative factors surely contribute to human errors. Our sense is that in the experiments we ran, these factors are relatively minor contributors, such that the errors predominantly reflect inherent stimulus ambiguities that affect the model as much as humans. In other settings, fatigue and distraction will undoubtedly introduce sources of errors that the present model cannot replicate. But they also probably contribute to some of the human errors in our experiments (and could explain the slightly higher confusion rates in humans compared to the model).

We have acknowledged this issue in the revised text:

“We note that humans made slightly higher rates of confusions than the model, which could potentially also reflect lapses or failures of executive function, which undoubtedly contribute to human errors in some settings.” (lines 240-242)

“the similarity of human and model performance (in particular, the similar extent and pattern of selection errors) suggests that human selection failures may partly be an inevitable consequence of feature overlap between target and distractor talkers (some are surely also due to lapses, distraction and other factors we did not model).” (lines 656-659)

– Framing: Much of the modeling cleverly unifies known elements (feature gains, normalization, task optimization) rather than introducing a fundamentally new attentional mechanism. The manuscript could better acknowledge this and position the contribution as integration + prediction rather than conceptual breakthrough.

This perspective is one that we share, and how we intended to portray the work. The ingredients here were well established, but they had not been combined into a

working system and compared to behavior. We have made this explicit at the start of the discussion section on prior work:

“Although the conceptual ingredients of our model have a long history in attention research, there has been little prior work incorporating them into working models of sensory systems.” (lines 595-596)

###MINOR/MISC:

-- When comparing to visual attention models, I feel the Discussion in places overstates the uniqueness of this approach; relevant work in vision is acknowledged, but a more balanced framing would help.

We went through the Discussion and made revisions in a few places that we thought you might be referring to. We have tried to strike a balance between acknowledging prior work in vision while also helping the reader to see what is unique to this modeling effort. We now describe the work as “building on” prior work in vision.

-- Executive control / effort: The deterministic, static gains leave no room for attentional effort or dynamic adaptation. This limitation could be discussed in relation to human attention variability.

We agree completely. In fact, we regard the introduction of dynamically adapting gains as one of the most exciting next steps in this line of work. We have clarified our discussion of this issue in the discussion section on limitations and future directions:

“Humans can also adjust the strength of attention using executive control, perhaps based on the perceived difficulty of a task, and potentially relating to the feeling of effort. Extending the modeling framework to allow the strength of attentional gains to vary could help to understand effort in computational terms.” (lines 675-678)

-- Figure clarity: Some multi-panel figures are extremely dense (esp. Fig. 2, 3, 4). A schematic “map” of experimental conditions and which ones were validated in humans vs. only simulated would aid readers.

All experiments except for the large-scale screen in Figure 5 were run in both humans and models. In the figure where the issue was most pronounced (Figure 2), we changed the labeling to make the human and model labels more prominent:

-- The “early-only” and “late-only” controls are informative, but further analysis (e.g., why the early-only still partially works) would strengthen the conclusions.

We think it is not so surprising, on reflection, that the alternative models perform above-chance to varying degrees. The key thing to note is that the entire model is

optimized with the alternative architectural constraint. As a result, the model is free to learn a completely different set of features from the feature-gain model in order to do the best it can with the architecture it is stuck with.

We have clarified this issue in the revised text:

“We also emphasize that the alternative models were fully optimized with the alternative architecture, and were free to learn an entirely different feature hierarchy that allowed them to maximize their performance given the constraint of gains in particular locations (or no explicit gains at all). It is thus perhaps not surprising that these models were able to perform well above chance.” (lines 555-559)

We have also moved a result that was previously easy to miss (in a supplementary figure that was referred to very briefly) into the main figure, as it shows that the alternative models are specifically impaired for conditions requiring spatial attention. Having this result featured more prominently will make it easier for readers to gain an intuition about the control models:

-- Somewhat related, the phrase “late selection” is used normatively. However, readers outside the auditory attention field may expect a clearer link back to the classic early/late selection debate.

We have revised the text to make the relation to the classic and more recent literature more clear:

“One of the main debates surrounding attention involves the stage at which attentional selection occurs. These debates originally concerned whether

attention acted before or after “semantic” processing, but with the discovery of attentional effects within sensory systems, interest shifted to differences in attentional modulation between stages of sensory hierarchies, and it is that setting in which we contrast “early” and “late” selection. .” (lines 642-646)

###TYPO-LEVEL

- “contrained” instead of “constrained” (p. 2).
- “explicitely” should be “explicitly”.

Oddly enough, we could not find either of these typos in the manuscript. We searched for both misspellings and found no occurrences.

Reviewer #3 (Remarks on code availability):

I have not fully testdriven the code, but have perused the provided materials. Providing results files and example stimuli is commendable, as is the python code.